# Rethinking Information-theoretic Generalization: Loss Entropy Induced PAC Bounds

**Yuxin Dong**[1], **Tieliang Gong**[1]*, **Hong Chen**[2], **Shujian Yu**[3] & **Chen Li**[1]
[1]Xi'an Jiaotong University, [2]Huazhong Agriculture University, [3]Vrije Universiteit Amsterdam
{yxdong9805,adidasgtl,yusj9011}@gmail.com, chenh@mail.hzau.edu.cn, cli@xjtu.edu.cn

## Abstract

Information-theoretic generalization analysis has achieved astonishing success in characterizing the generalization capabilities of noisy and iterative learning algorithms. However, current advancements are mostly restricted to average-case scenarios and necessitate the stringent bounded loss assumption, leaving a gap with regard to computationally tractable PAC generalization analysis, especially for long-tailed loss distributions. In this paper, we bridge this gap by introducing a novel class of PAC bounds through leveraging loss entropies. These bounds simplify the computation of key information metrics in previous PAC information-theoretic bounds to one-dimensional variables, thereby enhancing computational tractability. Moreover, our data-independent bounds provide novel insights into the generalization behavior of the minimum error entropy criterion, while our data-dependent bounds improve over previous results by alleviating the bounded loss assumption under both leave-one-out and supersample settings. Extensive numerical studies indicate strong correlations between the generalization error and the induced loss entropy, showing that the presented bounds adeptly capture the patterns of the true generalization gap under various learning scenarios.

## 1 Introduction

Employing information-theoretic measures to analyze the generalization properties of learning algorithms has recently attracted increasing attention (Xu & Raginsky, 2017; Bassily et al., 2018). The core concept involves quantifying the information stored in the model weights about the training dataset, serving as a natural indicator of overfitting. These bounds offer several advantages, as the information-theoretic measures are both dependent on the data distribution and the learning algorithm, while the underlying assumptions are considerably more lenient in comparison to contemporary techniques like uniform stability (Hardt et al., 2016; Bassily et al., 2020; Lei et al., 2021) and model compression (Arora et al., 2018; Zhou et al., 2018). Information-theoretic generalization analysis becomes a potent tool for characterizing the generalization capabilities of noisy and iterative learning algorithms (Negrea et al., 2019; Bu et al., 2020; Steinke & Zakynthinou, 2020; Wang & Mao, 2021; Haghifam et al., 2020; Neu et al., 2021; Raginsky et al., 2021).

Nonetheless, these generalization bounds are plagued by overestimation, as neural networks can memorize substantial training data while still generalizing well (Nasr et al., 2019). Furthermore, their information-theoretic foundation in Shannon's entropy renders them notoriously hard to compute due to the challenge of high-dimensional entropy estimation (Dong et al., 2023). A significant advancement was made in (Steinke & Zakynthinou, 2020) by adopting a setting where the training set is sampled from a larger supersample, and subsequently bounding the generalization error through the mutual information between the selection of the training set and various aspects e.g. predictions for the supersample (Harutyunyan et al., 2021), the loss pairs (Hellström & Durisi, 2022), or the loss difference (Wang & Mao, 2023). Remarkably, these bounds not only exhibit computational tractability due to the lower dimensionality of the associated random variables but also provide quantitatively tighter estimates, particularly for large neural networks.

While recent endeavors on computationally tractable generalization bounds have yielded promising outcomes, these bounds are mostly restricted to average-case scenarios, leaving a gap in terms of

---

*Corresponding author.

computationally tractable high-probability generalization bounds. Furthermore, these bounds rely on stringent bounded assumptions regarding the loss function, which can be limiting when dealing with long-tailed loss distributions e.g. cross-entropy. In this paper, we demonstrate that analyzing the properties of one-dimensional variables is sufficient for characterizing the generalization capabilities of learning algorithms. The generalization bounds we present address the computational challenges of prior high-probability bounds and mitigate the bounded assumption through our proposed thresholding strategy. Specifically, our contributions are summarized as follows:

- We demonstrate that the recently established bounds by Kawaguchi et al. (2023) can be further enhanced by substituting the mutual information between input and representation with loss entropy. This substitution leads to the derivation of data-independent generalization bounds, which employ both unconditional (Theorem 1) and conditional (Theorem 2) information measures. Notably, these bounds offer novel insights into the Minimum Error Entropy (MEE) criterion (Principe, 2010): To the best of our knowledge, we introduce the first information-theoretic generalization bound for MEE by showing that the generalization error scales with $\tilde{O}(\sqrt{H(E^w)/n})$, where $E^w$ represents the prediction error, and $n$ denotes the size of the training set.

- We simplify previous data-dependent generalization bounds relying on high-dimensional information measures with loss entropies. Our refined bounds exclusively involve one-dimensional variables: the losses or loss differences. These bounds are not only easier to evaluate but also alleviate the stringent bounded assumption regarding the loss function, under both the leave-one-out (Theorem 3) and supersample (Theorem 4) settings. Furthermore, we introduce novel fast-rate bounds for a generalized version of the weighted generalization error (Theorem 5), improving the convergence rate from $1/\sqrt{n}$ to $1/n$ when the empirical risk approaches zero. This advancement is further augmented by our thresholding strategy (Theorem 6), which effectively addresses the key limitations of previous fast-rate bounds when tackling long-tailed loss distributions.

- We substantiate our theoretical results with comprehensive numerical experiments on synthetic and real datasets, which exhibit distinct correlations between the generalization error and our proposed loss entropy metrics. Furthermore, the data-dependent bounds are capable of characterizing the generalization gap patterns across varied deep-learning tasks. Significantly, our square-root and fast-rate bounds consistently improve over the previously tightest high-probability bound (Hellström & Durisi, 2022) under the supersample setting.

## 2 PRELIMINARIES

We denote random variables by capitalized letters $(X)$, their specific realizations by lower-case letters $(x)$, and the corresponding spaces by calligraphic letters $(\mathcal{X})$. Shannon's entropy of random variable $X$ is denoted by $H(X)$, while Rényi's $\alpha$-order entropy of $X$ is denoted by $H_\alpha(X)$. The mutual information between random variables $X$ and $Y$ is denoted as $I(X;Y)$, and their conditional mutual information given $Z$ is denoted as $I(X;Y|Z)$. We further write $X|_{Y=y}$ (or $X|_y$) to represent $X$ conditioned on $Y = y$. We follow the same setting of (Kawaguchi et al., 2023) and assume that all loss variables involved in our theoretical analysis are discrete and have finite cardinality. This assumption also aligns well with continuous loss functions, as we discussed in Appendix G.3.

**Generalization Error**
Let $\mathcal{Z} = \mathcal{X} \times \mathcal{Y}$ denote the instance space of interest, where $\mathcal{X}$ and $\mathcal{Y}$ represent the input and label spaces, respectively. We define $\mathcal{W}$ as the hypotheses space, and $\ell : \mathcal{W} \times \mathcal{Z} \mapsto \mathcal{L} \in \mathbb{R}^+$ as the loss function. The training dataset $S = \{Z_i\}_{i=1}^n \in \mathcal{Z}^n$ is constructed by i.i.d sampling from the unknown data-generating distribution $\mu$. Given a learning algorithm $\mathcal{A}$ that takes $S$ as input and provides a hypothesis $W \in \mathcal{W}$, our main objective is to investigate the generalization behavior of $W$. Specifically, we denote $L^w = \ell(w, Z)$ for $Z \sim \mu$, and $L_i^w = \ell(w, Z_i)$ for $i \in [1, n]$. The generalization error $\Delta(w, S) = L(w) - L_S(w)$ is defined as the difference between the population risk $L(w) = \mathbb{E}_Z[L^w]$ and the empirical risk $L_S(w) = \frac{1}{n}\sum_{i=1}^n L_i^w$. Additionally, we introduce $b^w = \sup_{z \in \mathcal{Z}} \ell(w, z)$ to represent the maximum attainable loss, and $B^{w,S} = \sup_{i \in [1,n]} L_i^w$ to indicate the maximum samplewise loss given the hypothesis $w \in \mathcal{W}$ and the dataset $S$.

**Leave-One-Out Setting**
The leave-one-out (LOO) setting was recently introduced by Rammal et al. (2022) for generalization analysis. Let $\tilde{S}_l = \{Z_i\}_{i=1}^{n+1} \in \mathcal{Z}^{n+1}$ be a dataset containing $n + 1$ i.i.d samples. We denote $U \sim$

$\text{Unif}([1, n+1])$ as a uniform random variable, representing the single test sample chosen from $\tilde{S}_l$. The training $S_l$ and test $\bar{S}_l$ datasets are then constructed as $S_l = \tilde{S}_l \setminus Z_U$ and $\bar{S}_l = \{Z_U\}$. We define $R^w = \{L_i^w\}_{i=1}^{n+1} \in \mathcal{R}^w$ as the set of all individual samplewise losses and measure the generalization ability of the hypothesis $W$ by the LOO validation error $\Delta(W, \tilde{S}_l, U) = L_{\bar{S}_l}(W) - L_{S_l}(W)$.

**Supersample Setting**

The supersample framework was initially explored by Steinke & Zakynthinou (2020) for generalization analysis. Let $\tilde{S}_s = \{Z_{i,0}, Z_{i,1}\}_{i=1}^n \in \mathcal{Z}^{n \times 2}$ be a dataset constructed by i.i.d sampling $n \times 2$ samples. We denote $\tilde{U} = \{\tilde{U}_i\}_{i=1}^n \sim \text{Unif}(\{0,1\}^n)$ as $n$ random $\{0,1\}$ variables used to separate training and test samples, where $\tilde{U}_i = 0$ indicates that $Z_{i,0}$ is used for training and $Z_{i,1}$ for testing. The training and test datasets are then constructed as $S_s = \{Z_{i,\tilde{U}_i}\}_{i=1}^n$ and $\bar{S}_s = \{Z_{i,1-\tilde{U}_i}\}_{i=1}^n$, respectively. We define $\tilde{R}^w = \{L_{i,0}^w, L_{i,1}^w\}_{i=1}^n \in \tilde{\mathcal{R}}^w$ as the set of all individual samplewise losses, and $\tilde{R}_\Delta^w = \{\Delta L_i^w\}_{i=1}^n \in \tilde{\mathcal{R}}_\Delta^w$ as the set of loss differences, where $\Delta L_i^w = L_{i,1}^w - L_{i,0}^w$. The generalization ability is measured by the validation error $\Delta(W, \tilde{S}_s, \tilde{U}) = L_{\bar{S}_s}(W) - L_{S_s}(W)$.

## 3 MAIN THEOREMS

In this section, we investigate high-probability bounds to establish a connection between loss entropy and the generalization error. We begin by enhancing existing upper bounds presented in (Kawaguchi et al., 2023), considering scenarios where the model remains fixed and independent of the training dataset. Subsequently, we introduce novel data-dependent generalization bounds under both the LOO and supersample settings. Lastly, we put forth fast-rate bounds concerning a generalized version of the weighted validation error (Zhivotovskiy & Hanneke, 2018; Yang et al., 2019).

### 3.1 DATA-INDEPENDENT BOUNDS

The first part of our analysis demonstrates that minimizing the loss entropy effectively enhances the generalization ability of deep learning models, under the assumption that the network $w$ remains fixed and independent of the training dataset $S$. Precisely, given arbitrary $w \in \mathcal{W}$, random sample $Z \sim \mu$ and let $L^w = \ell(w, Z)$, we establish the following upper bounds:

**Theorem 1.** *For any $\gamma > 0$ and $\delta > 0$, with probability at least $1 - \delta$ over the draw of $S$:*

$$\Delta(w, S) \leq C_1^w \sqrt{\frac{H(L^w) + C_2^w}{n}} + \frac{C_3^w}{\sqrt{n}}, \quad \text{where} \begin{cases} C_1^w = 2b^w\sqrt{2} \\ C_2^w = c^w\sqrt{\frac{m\log(\sqrt{n}/\gamma)}{2}} + \log(2/\delta) \\ C_3^w = \gamma b^w + \frac{B^{w,S}\sqrt{\gamma}}{n^{1/4}}\sqrt{2\log(2/\delta)} \end{cases}$$

**Theorem 2.** *For any $\gamma > 0$ and $\delta > 0$, with probability at least $1 - \delta$ over the draw of $S$:*

$$\Delta(w, S) \leq \tilde{C}_1^w \sqrt{\frac{H(L^w|Y) + \tilde{C}_2^w}{n}} + \frac{\tilde{C}_3^w}{\sqrt{n}}, \quad \text{where} \begin{cases} \tilde{C}_1^w = 2b^w\sqrt{2|\mathcal{Y}|} \\ \tilde{C}_2^w = c^w\sqrt{\frac{m\log(\sqrt{n}/\gamma)}{2}} + \log(2|\mathcal{Y}|/\delta) \\ \tilde{C}_3^w = \gamma b^w + \frac{B^{w,S}\sqrt{\gamma|\mathcal{Y}|}}{n^{1/4}}\sqrt{2\log(2|\mathcal{Y}|/\delta)} \end{cases}$$

In this context, $m$ and $c^w$ represent the dimension and the sensitivity of the nuisance variables respectively, which are the source of randomness in $\mu$. It is evident that $C_1^w, C_2^w, C_3^w = \tilde{O}(1)$ as $n \to \infty$ (likewise for $\tilde{C}_1^w, \tilde{C}_2^w, \tilde{C}_3^w$), exhibiting a convergence rate of $1/\sqrt{n}$. It is worth noting that benefitted from the adoption of final layer outputs (i.e. losses), our bounds are applicable to arbitrary deterministic models and loss functions that can be expressed as $f : \mathcal{Z} \mapsto \mathbb{R}^+$, without requirements on the presence of intermediate representation layers as stipulated in (Kawaguchi et al., 2023).

A comparison between Theorem 1 and 2 highlights a trade-off concerning the complexity of $|\mathcal{Y}|$ (cardinality of the label space): On the one hand, we have $H(L^w|Y) \leq H(L^w)$, as conditioning always reduces entropy. On the other hand, Theorem 2 scales linearly with $\sqrt{|\mathcal{Y}|}$, resulting in potentially ineffective bounds when $|\mathcal{Y}|$ is large, e.g. regression tasks. When $I(L^w; Y)$ is large, signifying prominent performance variation across classes, it is advisable to adopt Theorem 2 for tighter upper bounds. Otherwise, the upper bound in Theorem 1 is preferred.

A prominent difference between Theorem 2 and previous data-independent bounds established in (Kawaguchi et al., 2023) lies in the replacement of $I(X; T^w|Y)$ with $H(L^w|Y)$, where $T^w$ represents any intermediate representation generated by the encoder of the model. By recognizing the Markov chain relationship $X \to T^w \to L^w$ conditioned on the label $Y$ and leveraging the data-processing inequality (DPI), we rigorously prove that the mutual information $I(L^w; X|Y) = H(L^w|Y) - H(L^w|X,Y)$ serves as a tighter bound over the objective $I(X; T^w|Y)$ of the information bottleneck principle. When the model $w$ is deterministic, which is commonly the case for modern deep learning models, we have $H(L^w|X,Y) = 0$, implying the equivalence between $I(L^w; X|Y)$ and $H(L^w|Y)$. Moreover, it is known that the mutual information $I(X; T^w|Y)$ can be infinite in certain special cases when $X$ can be deterministically recovered from $T^w$ (Amjad & Geiger, 2019). Conversely, the loss entropies $H(L^w)$ and $H(L^w|Y)$ are always finite.

Another improvement brought by our bounds is related to the factor $C_1^w$ which originally involves $\sum_{t \in \mathcal{T}_\epsilon^w} \mathbb{P}^{1/2}(T^w = t)$, where $\mathcal{T}_\epsilon^w \approx \mathcal{T}^w$ represents the typical subset of the representation space $\mathcal{T}^w$. This quantity exhibits exponential scaling with $H_{1/2}(T^w) > H(T^w)$, necessitating an additional assumption about the probability decay rate of $T^w$ to bound $C_1^w$ and maintain the convergence rate of $1/\sqrt{n}$. By adopting leave-one-out analysis (see Appendix B and C) when establishing concentration bounds for multinomial distributions, we overcome this limitation and obtain constant $C_1^w$ factors that are independent of $L^w$, without any further assumptions.

It is noteworthy that our data-independent bounds provide new insights into understanding the generalization of the MEE criterion. Typically, for regression or binary classification tasks, the prediction error is evaluated by $E^w = Y - f(w, X)$ given the sample $Z \sim \mu$, where $f$ represents the encoder. The final loss is then deterministically computed given the loss function $\ell$ as $L^w = \ell(E^w)$, e.g. the MSE loss $\ell(x) = x^2$. Then we have $H(L^w) \le H(E^w)$ due to the data-processing inequality, which directly indicates that optimizing $H(E^w)$ minimizes the upper bound in Theorem 1, verifying that MEE enhances the generalization capability of deep-learning models.

Lastly, our results address an important computability issue of previous upper bounds presented by Kawaguchi et al. (2023): accurately calculating the value of $I(X; T^w|Y)$ is exceedingly challenging due to the high dimensionality of both $X$ and $T^w$, particularly for modern large neural networks. This poses additional obstacles when attempting to utilize these bounds to assess the generalization ability of deep learning models in practical scenarios. In contrast, our upper bounds only involve $L^w$ and $Y$, which are one-dimensional for most traditional learning tasks, making the estimation of both $H(L^w)$ and $H(L^w|Y)$ feasible and efficient.

While one may notice that the model $w$ is not always independent of the training dataset $S$ in real applications, theorem 1 and 2 still offer practical utility in several real-world learning tasks, such as pre-training, where one aims to assess the generalization ability of pre-trained models on a specific dataset. Another relevant task is model evaluation in the validation dataset, for which our bounds provide valuable information on the magnitude and the likelihood of the average validation error deviating from the population risk.

## 3.2 DATA-DEPENDENT BOUNDS

In the following section, we extend our analysis to encompass data-dependent scenarios, where the model $W$ becomes a random variable correlated with the dataset $S$ during training. To achieve this, we explore both the LOO and supersample settings to analyze the generalization ability of deep learning models and establish upper bounds for the validation error under both cases.

**Theorem 3.** *For any $\lambda \in (0,1)$ and $\delta > 0$, with probability at least $1 - \delta$ over the draw of $W, \tilde{S}_l, U$:*

$$\Delta\left(W, \tilde{S}_l, U\right) \le C_1^W \sqrt{H_{1-\lambda}(R^W) + C_2^W}, \quad \text{where} \begin{cases} C_1^W = \sqrt{2} \Sigma_{R^W} \\ C_2^W = \frac{1}{\lambda} \log\left(\frac{1}{\delta}\right) + \log\left(\frac{2}{\delta}\right) \end{cases}$$

*by assuming that $\frac{n+1}{n}(L_U^W - \bar{L}^W)$ is $\Sigma_{R^W}$-subgaussian w.r.t $U$ for $\Sigma_{R^W} \in [0, B^{W, \tilde{S}_l}]$.*

**Theorem 4.** *For any $\lambda \in (0,1)$ and $\delta > 0$, with probability at least $1 - \delta$ over the draw of $W, \tilde{S}_s, \tilde{U}$:*

$$\Delta\left(W, \tilde{S}_s, \tilde{U}\right) \le \tilde{C}_1^W \sqrt{\frac{H_{1-\lambda}(\tilde{R}_\Delta^W) + \tilde{C}_2^W}{n}}, \quad \text{where} \begin{cases} \tilde{C}_1^W = \sqrt{\frac{2}{n} \sum_{i=1}^n \left(\Delta L_i^W\right)^2} \\ \tilde{C}_2^W = \frac{1}{\lambda} \log\left(\frac{1}{\delta}\right) + \log\left(\frac{2}{\delta}\right) \end{cases}$$

Importantly, it holds that $\lim_{n\to\infty} \Delta(W, \tilde{S}_s, \tilde{U}) = \Delta(W, S_s)$, indicating the supersample validation error approximates the generalization error when the dataset is sufficiently large. Similarly, we have $\tilde{C}_1^W, \tilde{C}_2^W = \tilde{O}(1)$ as $n \to \infty$. The parameter $\lambda$ implies a trade-off between Rényi's $\alpha$-order joint entropy of samplewise losses and the probability of generalization, as $H_\alpha(X)$ monotonically decreases with increasing $\alpha$. Notably, Theorem 4 scales proportionally with $1/\sqrt{n}$, whereas Theorem 3 does not. This is attributed to the fact that the test loss is only evaluated on a single sample $Z_U$ under the LOO setting, resulting in higher variance in the validation error. Such a phenomenon is also observed by Rammal et al. (2022); Haghifam et al. (2022).

The main advantage of Theorem 3 lies in its applicability under the interpolating regime, where the model achieves zero training loss. In this case, $H(R^W)$ simplifies to $H(L_U^W)$, which is the entropy of a one-dimensional random variable. This feature enables direct tractability of this upper bound when provided with i.i.d samples of the test loss $L_U^W$. Similarly, $H(\tilde{R}_\Delta^W)$ reduces to $H(\{L_{i,1-\tilde{U}_i}\}_{i=1}^n)$ under the interpolating regime, the joint entropy of all samplewise test losses. While this quantity is not directly tractable due to its high dimensionality, the subadditivity of Shannon's entropy could be utilized to establish an alternative upper bound by $H(\{L_{i,1-\tilde{U}_i}\}_{i=1}^n) \leq \sum_{i=1}^n H(L_{i,1-\tilde{U}_i})$. This subadditivity property also applies to Rényi's entropy when $\lambda \approx 0$ (see Appendix E.1), confirming the tractability of the upper bounds in both Theorem 3 and 4.

Theorem 4 further utilizes the loss difference between training and test loss values to derive strictly tighter bounds, which are denoted as $\Delta L_i^W = L_{i,1}^W - L_{i,0}^W$. This concept was first explored in (Wang & Mao, 2023), in which it was proven that $I(\Delta L_i^W, \tilde{U}_i) \leq I(L_{i,1}^W, L_{i,0}^W; \tilde{U}_i)$ by applying the DPI. Here we extend this conclusion to the loss entropy metrics by leveraging the concavity of Shannon's entropy. Specifically, we have that $\frac{1}{2}\big(H(\Delta L_i^W) + H(L_{i,0}^W)\big) \leq H(L_{i,1}^W)$ and $\frac{1}{2}\big(H(\Delta L_i^W) + H(L_{i,1}^W)\big) \leq H(L_{i,0}^W)$, which further yields the following reductions:

$$H(\Delta L_i^W) \leq \tfrac{1}{2}\big(H(L_{i,0}^W) + H(L_{i,1}^W)\big) \leq \max\big(H(L_{i,0}^W), H(L_{i,1}^W)\big) \leq H(L_{i,0}^W, L_{i,1}^W).$$

The most notable improvement of Theorem 3 and 4 compared to previous data-dependent bounds presented in (Negrea et al., 2019; Rammal et al., 2022; Steinke & Zakynthinou, 2020) is the replacement of $I(W; S)$, $I(W; U|\tilde{S}_l)$ and $I(W; \tilde{U}|\tilde{S}_s)$ with $H(R^W)$ and $H(\tilde{R}_\Delta^W)$. To see this, we take the LOO setting as an example and leverage the Markov chain relationship: $U \to S_l \to W \to R^W$ conditioned on $\tilde{S}_l$, which demonstrates that $I(R^W; U|\tilde{S}_l)$ is strictly smaller than $I(W; U|\tilde{S}_l)$ and $I(W; S_l|\tilde{S}_l)$ by applying the DPI. By utilizing the independence between $U$ and $\tilde{S}_l$, we have $I(R^W; U) \leq I(R^W; U) + I(U; \tilde{S}_l|R^W) = I(R^W; U|\tilde{S}_l) + I(U; \tilde{S}_l) = I(R^W; U|\tilde{S}_l)$. Similarly, the conditional independence between $\tilde{S}_l$ and $W$ given $S_l$ indicates $I(W; S_l|\tilde{S}_l) \leq I(W; S_l|\tilde{S}_l) + I(W; \tilde{S}_l) = I(W; \tilde{S}_l|S_l) + I(W; S_l) = I(W; S_l)$. When the training process $\mathcal{A} : S_l \mapsto W$ is deterministic (e.g., using full gradient descent or stochastic algorithms with a fixed seed), the randomness of $R^W$ is mainly induced by $U$, which implies $H(R^W|U) \approx 0$. With these in mind, we have

$$H(R^W) \approx I(R^W; U) \leq I(R^W; U|\tilde{S}_l) \leq I(W; U|\tilde{S}_l) \leq I(W; S_l|\tilde{S}_l) \leq I(W; S_l).$$

This verifies our claim that introducing the loss entropy $H(R^W)$ constitutes a significant improvement over $I(W; S)$, $I(W; U|\tilde{S}_l)$ and $I(W; \tilde{U}|\tilde{S}_s)$. The same conclusion can also apply to $H(\tilde{R}_\Delta^W)$. Moreover, these previous bounds encounter the same computational challenge in modern deep learning settings, which is even more severe than estimating $I(X; T^w|Y)$ due to the considerably higher dimensionalities of both $W$ and $S_l$ compared to $X$ or $T^w$. In contrast, our bounds could be efficiently approximated by estimating the entropy of one-dimensional random variables.

Theorems 3 and 4 are versatile and applicable to scenarios where the training process of the model $W$ involves the dataset $S$. This broadens the range of applications compared to Theorems 1 and 2, encompassing supervised learning, unsupervised learning, transfer learning, and other contexts. For both Theorems 3 and 4, the key quantities $H(R^W)$ or $H(\tilde{R}^W)$ can be further decomposed into $H(R^W) \leq H(R_S^W) + H(R_{\tilde{S}}^W)$, where $R_S^W$ and $R_{\tilde{S}}^W$ represent the collections of samplewise training and test losses, respectively. Intuitively, training algorithms aim to minimize the training loss, thereby reducing $H(R_S^W)$ since all training losses tend to approach zero for a well-fitted model. At the same time, $H(R_{\tilde{S}}^W)$ measures the extent of overfitting, which arises when the model provides incorrect answers to test samples. This illustrates a novel trade-off between training and test loss entropies to achieve the best generalization performance.

In addition to the presented main theorems, our proof technique also holds intrinsic research interest. Specifically, we observe the Markov chain $W \to R \to \Delta$ conditioned on the selection $U$, where $R$ acts as the "bottleneck" for information flow from $W$ to the evaluation error $\Delta$ (e.g. $R^W$ or $\tilde{R}_\Delta^W$). We proceed by investigating a typical subset of space $\mathcal{R}_\epsilon$ which satisfies $\mathbb{P}(R \notin \mathcal{R}_\epsilon) < \delta$ and $|\mathcal{R}_\epsilon| = O(e^{H(R)})$. By systematically enumerating each element $r \in \mathcal{R}_\epsilon$ within this specified subset, we effectively isolate $\Delta$ from $W$ and construct generalization bounds upon concentration inequalities that exclusively leverage the stochastic nature of $U$. By taking the union bound across every single element $r \in \mathcal{R}$, we establish a connection between sample complexity-based and information-theoretic analyses, ultimately leading to our conclusive results. While our paper primarily investigates the LOO and supersample settings, we underscore potential extensions of such a technique to accommodate alternative sampling scenarios, e.g. in cases of random selecting $n < m$ training samples from a supersample dataset comprising a total of $m$ samples.

### 3.2.1 Fast-Rate Bounds

We now delve into the weighted validation error denoted as $\Delta_C(W, \tilde{S}_s, \tilde{U}) = L_{\bar{S}_s}(W) - (1 + C)L_{S_s}(W)$, where $C$ is a selected positive constant. The introduction of this weighted error enables establishing fast-rate bounds that scale linearly with $1/n$ instead of the conventional $1/\sqrt{n}$ rates (Catoni, 2007; Hellström & Durisi, 2021b). While previous bounds typically adopt a universal value of $C$ for all training losses (Hellström & Durisi, 2022; Wang & Mao, 2023), we empirically observe that individual sample losses often exhibit a long-tailed distribution in well-trained deep-learning models: the majority of training samples have losses that cluster near zero, while a small number of samples consistently exhibit comparatively high losses even after the training process, significantly influencing the overall empirical risk. Motivated by this observation, we employ distinct values of $C_i$ for each individual training loss $L_{i,\tilde{U}_i}^W$ to derive strictly tighter bounds.

**Theorem 5.** *For any $\kappa \geq 0$, $\lambda, \gamma \in (0,1)$ and $\delta > 0$, if $\kappa \geq B^{W,\tilde{S}_s}$, then with probability at least $1 - \delta$ over the draw of $W, \tilde{S}_s, \tilde{U}$, the following inequality holds:*

$$\Delta\left(W, \tilde{S}_s, \tilde{U}\right) \leq \frac{1}{n}\sum_{i=1}^n C_i L_{i,\tilde{U}_i}^W + G_1^W \frac{H_{1-\lambda}(\tilde{R}^W) + G_2^W}{n}, \quad \text{where} \begin{cases} G_1^W = \frac{1}{\eta} = \frac{2\kappa}{\gamma \log 2} \\ G_2^W = \frac{1}{\lambda}\log\left(\frac{1}{\delta}\right) + \log\left(\frac{4}{\delta}\right) \\ C_i = -\frac{\log\left(2 - e^{2\eta \hat{L}_i^W}\right)}{2\eta \hat{L}_i^W} - 1 \end{cases}$$

*and $\hat{L}_i^W = \max(L_{i,0}^W, L_{i,1}^W)$ for any $i \in [1, n]$.*

In the interpolating regime where training losses approach zero, the weighted validation error simplifies to its original unweighted form. Therefore, a convergence rate of $1/n$ is achieved by letting $\gamma \to 1$ when the empirical risk approaches zero. This characteristic renders the fast-rate bounds particularly valuable when the empirical risk is small or even zero. Conversely, when positive training losses are prevalent, the adaptive modulation of weights $C_i$ w.r.t $\hat{L}_i^W$ endows the bounds with the versatility to accommodate various loss distributions. In contrast, if a universal constant $C$ is applied uniformly to all training losses, it must satisfy the condition that $C \geq \sup_{i \in [1,n]} C_i$, inevitably resulting in looser bounds when dealing with non-interpolating scenarios (see Appendix F.2).

Notably, the joint entropy $H(\tilde{R}^W)$ encompasses all samplewise losses, which facilitates direct computational tractability through the subadditivity property inherent in information entropy. In contrast, the previously established fast-rate high-probability generalization bound (Hellström & Durisi, 2022) employs the multivariate mutual information $I(\tilde{R}^W; \tilde{U}|\tilde{S}_s)$, which is unfortunately intractable due to the high-dimensional nature of the variables $\tilde{R}^W, \tilde{U},$ and $\tilde{S}_s$.

However, it is evident that the bound presented in Theorem 5 exhibits linear scaling with $B^{W,\tilde{S}_s}$, the largest samplewise loss. In scenarios involving long-tailed loss distributions (e.g. cross-entropy), this factor can be considerably larger than the subgaussian norm in Theorem 3 or the $L_2$ norm of loss differences in Theorem 4. Consequently, the improvement in the convergence rate may diminish in practical evaluations when compared to these previously established bounds.

In light of this observation, we further explore scenarios where $\kappa < B^{W,\tilde{S}_s}$. This naturally leads to the consideration of a thresholding strategy aimed at deriving tightened fast-rate generalization

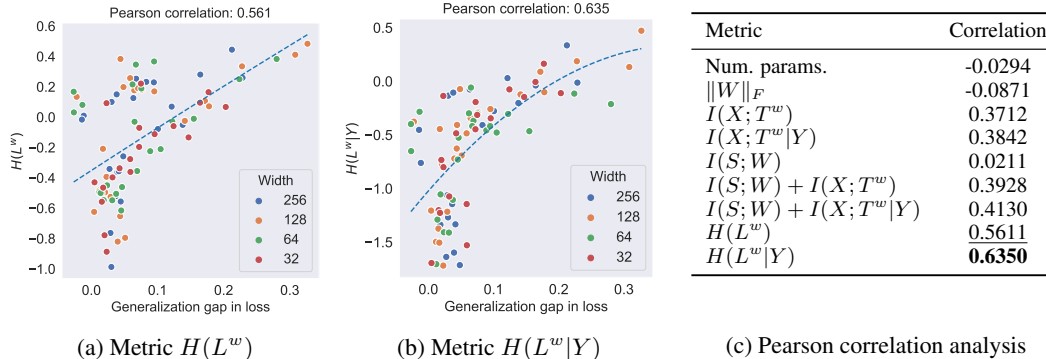

Figure 1: Pearson correlation analysis between the generalization error and different metrics for 5-layer MLP trained on synthetic 2D Gaussian datasets. (a), (b): The correlations of $H(L^w)$ and $H(L^w|Y)$ with the generalization gap, respectively. (c): Comparison of Pearson correlation coefficients for different metrics and the generalization gap.

bounds. With an arbitrary threshold $\kappa > 0$ and a samplewise loss $L$, it can be verified that $L = L^\kappa + L^{-\kappa}$, where $L^\kappa = \min(L, \kappa)$ and $L^{-\kappa} = \max(L - \kappa, 0)$. By employing the strategies outlined in Theorems 4 and 5, we can derive generalization bounds that simultaneously leverage the advantages of fast convergence rates and the modest $L_2$ scaling factor:

**Theorem 6.** *For any $\kappa > 0$, $\gamma, \lambda_1, \lambda_2 \in (0, 1)$ and $\delta > 0$, with probability at least $1 - \delta$ over the draw of $W, \tilde{S}_s, \tilde{U}$, the following inequality holds:*

$$\Delta\left(W, \tilde{S}_s, \tilde{U}\right) \leq \frac{1}{n} \sum_{i=1}^{n} C_i L_{i,\tilde{U}_i}^{W,\kappa} + \tilde{G}_1^W \frac{H_{1-\lambda_1}(\tilde{R}^{W,\kappa}) + \tilde{G}_2^W}{n} + \tilde{G}_3^W \sqrt{\frac{H_{1-\lambda_2}(\tilde{R}_\Delta^{W,-\kappa}) + \tilde{G}_4^W}{n}},$$

*where $\tilde{R}^{W,\kappa} = \{L_{i,0}^{W,\kappa}, L_{i,1}^{W,\kappa}\}_{i=1}^{n}$, $\tilde{R}_\Delta^{W,-\kappa} = \{\Delta L_i^{W,-\kappa}\}_{i=1}^{n}$, $\Delta L_i^{W,-\kappa} = L_{i,1}^{W,-\kappa} - L_{i,0}^{W,-\kappa}$ and*

$$\tilde{G}_1^W = \frac{1}{\eta} = \frac{2\kappa}{\gamma \log 2}, \quad \tilde{G}_2^W = \frac{1}{\lambda_1}\log\left(\frac{2}{\delta}\right) + \log\left(\frac{8}{\delta}\right), \quad C_i = -\frac{\log\left(2 - e^{2\eta \hat{L}_i^{W,\kappa}}\right)}{2\eta \hat{L}_i^{W,\kappa}} - 1,$$

$$\tilde{G}_3^W = \sqrt{\frac{2}{n} \sum_{i=1}^{n}\left(\Delta L_i^{W,-\kappa}\right)^2}, \quad \tilde{G}_4^W = \frac{1}{\lambda_2}\log\left(\frac{2}{\delta}\right) + \log\left(\frac{4}{\delta}\right).$$

Theorem 6 introduces a manual threshold $\kappa$ to partition each samplewise loss into $L = L^\kappa + L^{-\kappa}$. Combining with the techniques developed in Theorems 4 and 5, we subsequently establish upper bounds for these two components. Consequently, the previous factor $B^{W,\tilde{S}_s}$ is replaced by the manually set threshold $\kappa$ for the first component $L^\kappa$, and by the more moderate $L_2$ norm of loss differences for the second component $L^{-\kappa}$. Our experimental findings indicate that this bound is significantly tighter than both Theorems 4 and 5 (see Appendix F.2). It is worth noting that this thresholding technique can also be applied to the bounds presented in (Wang & Mao, 2023), yielding improved bounds for the expected generalization error when dealing with long-tailed loss distributions (see Appendix E.2). Furthermore, the joint entropy of all training losses, denoted as $H(\tilde{R}_S^W)$, serves as a tighter alternative to the terms related to loss variance and sharpness, which were utilized in (Wang & Mao, 2023) to derive fast-rate expected generalization bounds (see Appendix E.3).

## 4 NUMERICAL RESULTS

In this section, we conduct empirical comparisons between the generalization bounds established in this paper and the previous high-probability bounds proposed in (Kawaguchi et al., 2023; Hellström & Durisi, 2022). Our evaluation involves two sets of experiments: Firstly, we investigate data-independent bounds using synthetic 2D Gaussian datasets by employing a simple MLP network as the classifier, which follows the same learning settings as (Kawaguchi et al., 2023). Secondly, we evaluate data-dependent bounds by training more complex neural networks on real-world image classification datasets (4-layer CNN on MNIST (LeCun & Cortes, 2010) and ResNet-50 (He et al.,

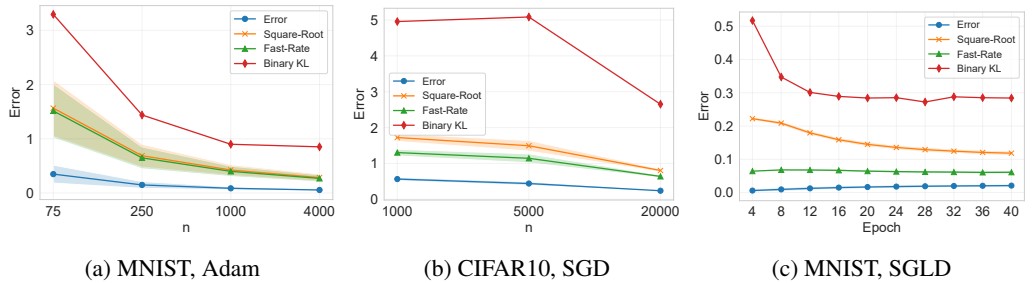

Figure 2: Comparison of the generalization gap in 3 different deep-learning scenarios, along with theoretical upper bounds including the square-root bound (Theorem 4), the fast-rate bound (Theorem 6) and the binary KL bound (Theorem 7 in (Hellström & Durisi, 2022)).

2016) on CIFAR10 (Krizhevsky et al., 2009)). These experiments follow the same deep learning settings as those in (Harutyunyan et al., 2021; Hellström & Durisi, 2022; Wang & Mao, 2023). In all of these experiments, we utilize the cross-entropy loss $\mathcal{L}_{CE}$ to quantify the generalization error and adopt the empirical risk minimization criterion to find the hypothesis $W$.

## 4.1 SYNTHETIC DATA

For the first classification task, we utilize synthetic 2D Gaussian datasets and employ a 5-layer MLP, comprising a variational encoder and a deterministic classifier. The experimental settings are consistent with those in (Kawaguchi et al., 2023), where we train a total of 216 different models, covering various architectures, weight decay rates, dataset draws, and random seeds. The optimization of the hypothesis is performed using the reparameterization trick (Kingma et al., 2015) in an end-to-end manner. A Monte-Carlo sampling-based estimator is employed to estimate the values of $I(X; T^w)$ and $I(X; T^w|Y)$, and $I(W; S)$ is computed based on the posterior distribution modeled by the SWAG method (Mandt et al., 2017; Maddox et al., 2019). The computations of $H(L^w)$ and $H(L^w|Y)$ are relatively straightforward, such that a simple kernel density estimator suffices.

Here, we use Gaussian kernels and determine the kernel width according to the rule-of-thumb criterion. To empirically evaluate the predictive power of different metrics on generalization, we follow the approach in previous works (Galloway et al., 2023; Kawaguchi et al., 2023) and adopt the Pearson correlation analysis. As depicted in Figure 1, both $H(L^w)$ and $H(L^w|Y)$ exhibit stronger correlations with the generalization error in comparison to other metrics, including the number of parameters, the F-norm of the hypothesis, information bottleneck $I(X; T^w)$ related metrics, and the input-output mutual information $I(W; S)$. This observation highlights the capabilities of loss entropy metrics in assessing the generalization ability of deep learning models. Moreover, the observation that $H(L^w|Y)$ outperforms $H(L^w)$ supports our analysis, demonstrating that Theorem 2 is more effective than Theorem 1 when the cardinality of the label space $|\mathcal{Y}|$ is finite.

## 4.2 REAL-WORLD LEARNING TASKS

In order to assess information-theoretic generalization bounds within the context of modern deep-learning tasks, we adhere to the experimental configurations established by Harutyunyan et al. (2021). Specifically, we consider training a 4-layer CNN on binarized MNIST data, which is restricted to comprise only digits 4 and 9. Additionally, we engage in fine-tuning a pre-trained ResNet-50 model on the CIFAR10 dataset. A comprehensive illustration of the training algorithm, network architecture, and experimental setup can be found in Appendix F.

As of our current knowledge, the most stringent fast-rate high-probability information-theoretic generalization bound available in the literature is the binary KL bound (Theorem 7 in (Hellström & Durisi, 2022)) employing $I(\tilde{R}^W; \tilde{U}|\tilde{S}_s)$. Note that this quantity is computationally intractable due to its high dimensionality, and here we adopt a lower-bound approximation: $\sum_{i=1}^{n} I(L_{i,0}^W, L_{i,1}^W; \tilde{U}_i)$ (see Appendix F.2 for the proof). In addition to this baseline (Binary KL), our primary comparison centers around the square-root generalization bound in Theorem 4 (Square-Root) and the fast-rate generalization bound in Theorem 6 (Fast-Rate).

The final outcomes are presented in Figure 2. As can be seen, these upper bounds effectively capture the patterns of the generalization gap. Across the three distinct learning scenarios, both our square-root bound and fast-rate bound exhibit significant improvements over the lower-bound approximation of the binary KL bound. This observation substantiates our previous discussion that our bounds can adeptly accommodate long-tailed loss distributions, while the binary KL bound is influenced by the maximum attainable loss value. Moreover, the fast-rate bound consistently outperforms the square-root bound, particularly in more challenging learning scenarios (CIFAR10 and SGLD). This provides empirical validation for the efficacy of our thresholding strategy.

Furthermore, it is evident from Figure 2c that the binary KL bound is especially loose in the initial epochs. This behavior arises due to the adoption of a universal $C$: according to the expression of $C_i$ in Theorem 5 and 6, using distinct $C_i$ particularly improves the bound when the majority of loss values are close to the threshold $\kappa$, which happens at the early stages of the training process. In contrast, our loss entropy-based generalization bounds overcome this limitation, consistently offering accurate predictions of the actual generalization error.

## 5 RELATED WORKS

This work is intricately linked to extensive literature concerning **information-theoretic generalization bounds**. The seminal contributions by Russo & Zou (2016); Xu & Raginsky (2017); Russo & Zou (2019) initially motivated the characterization of generalization properties through mutual information between training samples and model parameters. This approach has demonstrated its efficacy in dissecting the behavior of noisy and iterative learning algorithms, exemplified by its application in SGLD (Negrea et al., 2019; Wang et al., 2021) and SGD (Neu et al., 2021; Wang & Mao, 2021; Dong et al., 2023). Furthermore, it serves as a scaffold for subsequent enhancements by encompassing techniques such as conditioning (Hafez-Kolahi et al., 2020), the chaining strategy (Asadi et al., 2018; Zhou et al., 2022; Clerico et al., 2022), the random subsets or individual techniques (Bu et al., 2020; Rodríguez-Gálvez et al., 2021), and conditional information measures (Steinke & Zakynthinou, 2020; Haghifam et al., 2020).

Remarkably, Harutyunyan et al. (2021) introduced a novel approach to establish generalization bounds by leveraging conditional mutual information (CMI) between the model's output and super-sample variables. This technique treats the neural network as a "black box", resulting in a substantial reduction in the dimensionality of random variables used in constructing generalization bounds, rendering them directly tractable. Building upon this foundation, Hellström & Durisi (2022); Wang & Mao (2023) further extended the approach by incorporating evaluated losses and loss differences to derive tighter bounds. Another notable exploration in the CMI framework is the leave-one-out setting, as recently investigated by Haghifam et al. (2022); Rammal et al. (2022). This variant significantly reduces the required number of samples from $n \times 2$ to just $n + 1$.

The **information bottleneck** (IB) principle (Tishby et al., 2000) and the **minimum error entropy** (MEE) criterion (Erdogmus & Principe, 2000) play essential roles in designing supervised learning algorithms under the information-theoretic learning framework (Principe, 2010). The IB objective is developed based on the concept of minimal sufficient statistics and is widely adopted as a regularization technique (Alemi et al., 2016; Kolchinsky & Tracey, 2017; Kolchinsky et al., 2019). Notably, Kawaguchi et al. (2023) provides the first information-theoretic generalization bound for the IB principle, revealing that the generalization error scales roughly as $\tilde{O}(\sqrt{I(X;T^w|Y)/n})$. The MEE criterion is recognized for its robustness against outliers or covariate shifts, attributed to its capacity to capture high-order statistics of error distributions, and thus find applications in a variety of learning tasks (Hu et al., 2013; Shen & Li, 2015; Guo et al., 2020).

## 6 CONCLUSION

In this paper, we introduce a series of high-probability information-theoretic generalization bounds based on loss entropy. We demonstrate that these bounds are superior in tightness compared to previous counterparts established in (Kawaguchi et al., 2023; Hellström & Durisi, 2022), both theoretically and empirically. Our numerical experiments on both synthetic and real-world datasets substantiate that our bounds are consistently predictive of the true generalization gap across various deep-learning scenarios, validating the efficacy of our theoretical analysis.

ACKNOWLEDGMENTS

This work was supported by National Key Research and Development Program of China(No. 2021ZD0110700), National Natural Science Foundation of China (62106191, 12071166, 62192781,61721002), Innovation Research Team of Ministry of Education (IRT_17R86), Project of China Knowledge Centre for Engineering Science and Technology and Project of Chinese academy of engineering "The Online and Offline Mixed Educational Service System for 'The Belt and Road' Training in MOOC China".

**Reproducibility Statement**. To ensure reproducibility, we include complete proofs of our theoretical results in Appendix B, C and D, detailed explanations of our experimental settings in Appendix F, and source codes at `https://github.com/Yuxin-Dong/Loss-Entropy`.

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

# Appendix

## Table of Contents

## A  PREREQUISITE DEFINITIONS AND LEMMAS

Unless otherwise noted, we use $\log$ to denote the logarithmic function with base $e$.

**Definition 1.** *(Subgaussian) A random variable $X$ is $\sigma$-subgaussian if for any $\rho \in \mathbb{R}$, $\mathbb{E}[\exp(\rho(X - \mathbb{E}[X]))] \leq \exp(\rho^2 \sigma^2 / 2)$.*

**Definition 2.** *(Kullback-Leibler Divergence) Let $P$ and $Q$ be probability measures on the same space $\mathcal{X}$, the KL divergence from $P$ to $Q$ is defined as $\mathrm{KL}(P \parallel Q) \triangleq \int_{\mathcal{X}} p(x) \log(p(x)/q(x)) \, \mathrm{d}x$.*

**Definition 3.** *(Mutual Information) Let $(X, Y)$ be a pair of random variables with values over the space $\mathcal{X} \times \mathcal{Y}$. Let their joint distribution be $P_{X,Y}$ and the marginal distributions be $P_X$ and $P_Y$ respectively, the mutual information between $X$ and $Y$ is defined as $I(X; Y) = \mathrm{KL}(P_{X,Y} \parallel P_X P_Y)$.*

**Lemma 1.** *(Lemma 3 in Kawaguchi et al., 2022) Let the vector $X = (X_1, \cdots, X_n)$ follow the multinomial distribution with parameters $m$ and $p = (p_1, \cdots, p_n)$. Let $\bar{a}_1, \cdots, \bar{a}_n \geq 0$ be fixed such that $\sum_{i=1}^{n} \bar{a}_i p_i \neq 0$. Then for any $\epsilon > 0$,*

$$\mathbb{P}\left(\sum_{i=1}^{n} \bar{a}_i \left(p_i - \frac{X_i}{m}\right) > \epsilon\right) \leq \exp\left(-\frac{m\epsilon^2}{\beta}\right),$$

*where $\beta = 2 \sum_{i=1}^{n} \bar{a}_i^2 p_i$.*

**Lemma 2.** *(Lemma 1 in Xu & Raginsky, 2017) Let $(X, Y)$ be a pair of random variables with joint distribution $P_{X,Y}$ and let $\bar{Y}$ be an independent copy of $Y$. If $f(x, y)$ is a measurable function such that $E_{X,Y}[f(X, Y)]$ exists and $f(X, \bar{Y})$ is $\sigma$-subgaussian, then*

$$\left|\mathbb{E}_{X,Y}[f(X, Y)] - \mathbb{E}_{X,\bar{Y}}[f(X, \bar{Y})]\right| \leq \sqrt{2\sigma^2 I(X; Y)}.$$

**Lemma 3.** *(Donsker-Varadhan formula) Let $P$ and $Q$ be probability measures defined on the same measurable space, where $P$ is absolutely continuous with respect to $Q$. Then*

$$\mathrm{KL}(P \parallel Q) = \sup_{X}\{\mathbb{E}_P[X] - \log \mathbb{E}_Q[e^X]\},$$

*where $X$ is any random variable such that $e^X$ is $Q$-integrable and $\mathbb{E}_P[X]$ exists.*

## B  PROOF OF THEOREM 1

We define the data generating process as $Z = (\theta(Y, V), Y)$, where $Y$ is the randomly generated label, $\theta$ is some hidden deterministic function and $V = \{V_i\}_{i=1}^{m} \in \mathcal{V} \subset \mathbb{R}^m$ are i.i.d nuisance variables. For any $y \in \mathcal{Y}$, define the sensitivity $c_{i,y}^w$ of the model w.r.t the nuisance variable $V_i$:

$$
\begin{aligned}
c_{i,y}^w = \sup_{v_1, \cdots, v_i, \hat{v}_i, \cdots, v_m} \Big| &\log(p_l \circ \ell)(w, \theta_y(v_1, \cdots, v_i, \cdots, v_m), y) \\
&- \log(p_l \circ \ell)(w, \theta_y(v_1, \cdots, \hat{v}_i, \cdots, v_m), y)\Big|,
\end{aligned}
$$

where $p_l(l) = \mathbb{P}(L^w = l)$ and $\theta_y(v) = \theta(y, v)$. We then define the global sensitivity of $w$ as:

$$c_y^w = \sup_{i \in [1,m]} c_{i,y}^w, \quad \text{and} \quad c^w = \mathbb{E}_Y[c_Y^w].$$

Given the hypothesis $w \in \mathcal{W}$, we denote the set of all possible loss values by

$$\mathcal{L}^w = \{\ell(w, \theta_y(v), y) : v \in \mathcal{V}, y \in \mathcal{Y}\}.$$

For any $\gamma > 0$, we then define the typical subset of $\mathcal{L}^w$ by

$$\mathcal{L}_\gamma^w = \left\{l \in \mathcal{L}^w : -\log p_l(l) - H(L^w) \leq c^w \sqrt{\frac{m \log(\sqrt{n}/\gamma)}{2}}\right\}. \tag{1}$$

Note that $\mathcal{L}_\gamma^w$ is deterministic given the fixed hypothesis $w \in \mathcal{W}$.

### B.1  PROPERTIES OF THE TYPICAL SUBSET

**Lemma 4.** *For any $\gamma > 0$, we have*

$$\mathbb{P}\left(L^w \notin \mathcal{L}_\gamma^w\right) \leq \frac{\gamma}{\sqrt{n}},$$

*and*

$$\left|\mathcal{L}_\gamma^w\right| \leq \exp\left(H(L^w) + c^w \sqrt{\frac{m \log(\sqrt{n}/\gamma)}{2}}\right).$$

*Proof.* Consider the following function

$$f(y, v) = -\log p_l(h_y(v)), \quad \text{where} \quad h_y(v) = \ell(w, \theta_y(v)).$$

Let $p_y(y) = \mathbb{P}(Y = y)$, $p_v(v) = \mathbb{P}(V = v)$ and $h_y^{-1}(l) = \{v \in \mathcal{V} : h_y(v) = l\}$, we have

$$
\begin{aligned}
\mathbb{E}_{Y,V}[f(Y,V)] &= -\sum_{y \in \mathcal{Y}} p_y(y) \sum_{v \in \mathcal{V}} p_v(v) \log p_l(h_y(v)) \\
&= -\sum_{y \in \mathcal{Y}} p_y(y) \sum_{l \in \mathcal{L}^w} \sum_{v \in h_y^{-1}(l)} p_v(v) \log p_l(h_y(v)) \\
&= -\sum_{l \in \mathcal{L}^w} \left( \sum_{y \in \mathcal{Y}} p_y(y) \sum_{v \in h_y^{-1}(l)} p_v(v) \right) \log p_l(l) \\
&= -\sum_{l \in \mathcal{L}^w} p_l(l) \log p_l(l) \\
&= H(L^w).
\end{aligned}
$$

Therefore, by applying McDiarmid's inequality on $f(V) = -\log p_l(L^w)$, we have

$$
\mathbb{P}(-\log p_l(L^w) - H(L^w) \geq \epsilon) \leq \exp\left(-\frac{2\epsilon^2}{m(c^w)^2}\right). \tag{2}
$$

Take $\delta$ as the RHS of (2), we have

$$
\epsilon = c^w \sqrt{\frac{m \log(1/\delta)}{2}}. \tag{3}
$$

Combining with (1), we select $\delta = \gamma/\sqrt{n}$ and

$$
\mathbb{P}(L^w \notin \mathcal{L}_\gamma^w) \leq \delta = \frac{\gamma}{\sqrt{n}}.
$$

We now consider the size of the typical subset. For any $l \in \mathcal{L}_\gamma^w$, we have

$$
\begin{aligned}
-\log p_l(l) - H(L^w) &\leq \epsilon \\
-\log p_l(l) &\leq H(L^w) + \epsilon \\
-H(L^w) - \epsilon &\leq \log p_l(l) \\
\exp(-H(L^w) - \epsilon) &\leq p_l(l).
\end{aligned}
$$

This implies that

$$
1 \geq \mathbb{P}(L^w \in \mathcal{L}_\gamma^w) = \sum_{l \in \mathcal{L}_\gamma^w} p_l(l) \geq \sum_{l \in \mathcal{L}_\gamma^w} \exp(-H(L^w) - \epsilon) = |\mathcal{L}_\gamma^w| \exp(-H(L^w) - \epsilon).
$$

Combining with (3), we finally get

$$
|\mathcal{L}_\gamma^w| \leq \exp\left( H(L^w) + c^w \sqrt{\frac{m \log(\sqrt{n}/\gamma)}{2}} \right).
$$

$\square$

## B.2 DECOMPOSITION OF THE GENERALIZATION ERROR

For convenience, we define $t = |\mathcal{L}_\gamma^w|$, the elements of the typical subset as $\mathcal{L}_\gamma^w = \{a_1, \cdots, a_t\}$, and

$$
\begin{aligned}
\mathcal{I} &= \{i \in [1, n] : L_i^w \notin \mathcal{L}_\gamma^w\}, \\
\mathcal{I}_k &= \{i \in [1, n] : L_i^w = a_k\}.
\end{aligned}
$$

Note that $\mathcal{I}$ and $\mathcal{I}_k$ are random variables that are dependent on the training dataset $S$. We then use these notations to decompose the generalization error:

**Lemma 5.** *The generalization error $\Delta(w, S)$ satisfies*

$$
\Delta(w, S) = \mathbb{P}(L^w \notin \mathcal{L}_\gamma^w) \left( \mathbb{E}_{L^w}[L^w | L^w \notin \mathcal{L}_\gamma^w] - \frac{1}{|\mathcal{I}|} \sum_{i \in \mathcal{I}} L_i^w \right)
$$

$$+ \frac{1}{|\mathcal{I}|} \left( \mathbb{P}(L^w \notin \mathcal{L}_\gamma^w) - \frac{|\mathcal{I}|}{n} \right) \sum_{i \in \mathcal{I}} L_i^w$$

$$+ \sum_{k=1}^t \left( \mathbb{P}(L^w = a_k) - \frac{|\mathcal{I}_k|}{n} \right) a_k.$$

*Proof.* By noticing that $\mathcal{I} \cup \mathcal{I}_1 \cup \cdots \cup \mathcal{I}_t = [1, n]$, the population risk can then be decomposed as:

$$\mathbb{E}_{L^w}[L^w] = \mathbb{P}(L^w \notin \mathcal{L}_\gamma^w) \mathbb{E}_{L^w}[L^w | L^w \notin \mathcal{L}_\gamma^w] + \sum_{k=1}^t \mathbb{P}(L^w = a_k) \mathbb{E}_{L^w}[L^w | L^w = a_k]$$

$$= \mathbb{P}(L^w \notin \mathcal{L}_\gamma^w) \mathbb{E}_{L^w}[L^w | L^w \notin \mathcal{L}_\gamma^w] + \sum_{k=1}^t \mathbb{P}(L^w = a_k) a_k. \tag{4}$$

Similarly, we can decompose the empirical risk as:

$$\frac{1}{n} \sum_{i=1}^n L_i^w = \frac{1}{n} \left( \sum_{i \in \mathcal{I}} L_i^w + \sum_{k=1}^t \sum_{i \in \mathcal{I}_k} L_i^w \right)$$

$$= \frac{1}{n} \sum_{i \in \mathcal{I}} L_i^w + \sum_{k=1}^t \frac{1}{n} \sum_{i \in \mathcal{I}_k} a_k$$

$$= \frac{1}{n} \sum_{i \in \mathcal{I}} L_i^w + \sum_{k=1}^t \frac{|\mathcal{I}_k|}{n} a_k. \tag{5}$$

Substitute (4) and (5) into $\Delta(w, S)$, we can get

$$\Delta(w, S) = L(w) - L_S(w) = \mathbb{E}_{L^w}[L^w] - \frac{1}{n} \sum_{i=1}^n L_i^w$$

$$= \mathbb{P}(L^w \notin \mathcal{L}_\gamma^w) \mathbb{E}_{L^w}[L^w | L^w \notin \mathcal{L}_\gamma^w] - \mathbb{P}(L^w \notin \mathcal{L}_\gamma^w) \frac{1}{|\mathcal{I}|} \sum_{i \in \mathcal{I}} L_i^w$$

$$+ \mathbb{P}(L^w \notin \mathcal{L}_\gamma^w) \frac{1}{|\mathcal{I}|} \sum_{i \in \mathcal{I}} L_i^w - \frac{1}{n} \sum_{i \in \mathcal{I}} L_i^w$$

$$+ \sum_{k=1}^t \mathbb{P}(L^w = a_k) a_k - \sum_{k=1}^t \frac{|\mathcal{I}_k|}{n} a_k$$

$$= \mathbb{P}(L^w \notin \mathcal{L}_\gamma^w) \left( \mathbb{E}_{L^w}[L^w | L^w \notin \mathcal{L}_\gamma^w] - \frac{1}{|\mathcal{I}|} \sum_{i \in \mathcal{I}} L_i^w \right)$$

$$+ \frac{1}{|\mathcal{I}|} \left( \mathbb{P}(L^w \notin \mathcal{L}_\gamma^w) - \frac{|\mathcal{I}|}{n} \right) \sum_{i \in \mathcal{I}} L_i^w$$

$$+ \sum_{k=1}^t \left( \mathbb{P}(L^w = a_k) - \frac{|\mathcal{I}_k|}{n} \right) a_k.$$

$$\square$$

To simplify the notations, we denote the decomposition above as:

$$\Delta(w, S) = A(w, S) + B(w, S) + C(w, S),$$

where

$$A(w, S) = \mathbb{P}(L^w \notin \mathcal{L}_\gamma^w) \left( \mathbb{E}_{L^w}[L^w | L^w \notin \mathcal{L}_\gamma^w] - \frac{1}{|\mathcal{I}|} \sum_{i \in \mathcal{I}} L_i^w \right),$$

$$B(w, S) = \frac{1}{|\mathcal{I}|}\left(\mathbb{P}\left(L^w \notin \mathcal{L}_\gamma^w\right) - \frac{|\mathcal{I}|}{n}\right)\sum_{i\in\mathcal{I}} L_i^w,$$

$$C(w, S) = \sum_{k=1}^{t}\left(\mathbb{P}(L^w = a_k) - \frac{|\mathcal{I}_k|}{n}\right)a_k.$$

## B.3 Bounding each Term in the Decomposition

**Lemma 6.** *For any $\gamma > 0$, $A(w, S)$ satisfies:*

$$A(w, S) \leq \frac{\gamma b^w}{\sqrt{n}}.$$

*Proof.* From Lemma 4, we have

$$\mathbb{P}\left(L_w \notin \mathcal{L}_\gamma^w\right) \leq \frac{\gamma}{\sqrt{n}}.$$

Since $L_i^w \geq 0$ for any $i \in [1, n]$, we can prove that

$$\begin{aligned}
A(w, S) &= \mathbb{P}\left(L^w \notin \mathcal{L}_\gamma^w\right)\left(\mathbb{E}_{L^w}\left[L^w | L^w \notin \mathcal{L}_\gamma^w\right] - \frac{1}{|\mathcal{I}|}\sum_{i\in\mathcal{I}}L_i^w\right) \\
&\leq \mathbb{P}\left(L^w \notin \mathcal{L}_\gamma^w\right)\mathbb{E}_{L^w}\left[L^w | L^w \notin \mathcal{L}_\gamma^w\right] \\
&\leq \frac{\gamma}{\sqrt{n}}\mathbb{E}_{L^w}\left[L^w | L^w \notin \mathcal{L}_\gamma^w\right] \\
&\leq \frac{\gamma b^w}{\sqrt{n}}.
\end{aligned}$$

$\square$

**Lemma 7.** *For any $\gamma > 0$ and $\delta > 0$, with probability at least $1 - \delta$, $B(w, S)$ and $C(w, S)$ satisfies:*

$$B(w, S) \leq \frac{\sqrt{\mathbb{P}\left(L^w \notin \mathcal{L}_\gamma^w\right)}\sum_{i\in\mathcal{I}}L_i^w}{|\mathcal{I}|}\sqrt{\frac{2\log(2/\delta)}{n}},$$

$$C(w, S) \leq 2b^w\sqrt{\frac{2(H(L^w) + C_4^w) + 2\log(2/\delta)}{n}},$$

*where*

$$C_4^w = c^w\sqrt{\frac{m\log(\sqrt{n}/\gamma)}{2}}.$$

*Proof.* Denote $q_k = \mathbb{P}(L^w = a_k)$ for $k \in [1, t]$ and $q = \mathbb{P}\left(L^w \notin \mathcal{L}_\gamma^w\right)$. Let

$$C_k(w, S) = \sum_{i=1}^{t}\left(q_i - \frac{|\mathcal{I}_i|}{n}\right)a_i - \left(q_k - \frac{|\mathcal{I}_k|}{n}\right)a_k.$$

We then apply Lemma 1 with

$$\begin{aligned}
n &= t + 1, & X &= (|\mathcal{I}_1|, \cdots, |\mathcal{I}_t|, |\mathcal{I}|), & p &= (q_1, \cdots, q_t, q), \\
m &= n, & \bar{a}_k &= 0, \ \bar{a}_{t+1} = 0, \ \text{and} \ \bar{a}_i = a_i \ \text{for any} \ i \neq k.
\end{aligned}$$

When there exists $i \in [1, t] \setminus k$ such that $q_i a_i > 0$, we have $\sum_{i=1}^{t}\bar{a}_i q_i + \bar{a}_{t+1}q \neq 0$ and the precondition of Lemma 1 is satisfied. Then for any $\epsilon > 0$,

$$\mathbb{P}(C_k(w, S) > \epsilon) \leq \exp\left(-\frac{n\epsilon^2}{2\left(\sum_{i=1}^{t}q_i a_i^2 - q_k a_k^2\right)}\right), \tag{6}$$

for any $k \in [1, t]$. Similarly, by setting $\bar{a}_{t+1} = 1$ and $\bar{a}_i = 0$ for any $i \in [1, n]$, we have

$$\mathbb{P}\left(q - \frac{|\mathcal{I}|}{n} > \epsilon\right) \leq \exp\left(-\frac{n\epsilon^2}{2q}\right). \tag{7}$$

Take $\delta$ as the RHS of (6) and (7) respectively, we then get

$$\mathbb{P}\left(C_k(w, S) > \sqrt{\sum_{i=1}^{t} q_i a_i^2 - q_k a_k^2}\sqrt{\frac{2\log(1/\delta)}{n}}\right) \leq \delta, \tag{8}$$

for any $k \in [1, t]$ and

$$\mathbb{P}\left(q - \frac{|\mathcal{I}|}{n} > \sqrt{\frac{2q\log(1/\delta)}{n}}\right) \leq \delta. \tag{9}$$

Otherwise if $q_i a_i = 0$ for all $i \neq k$ or $q = 0$, then $C_k(w, S) = 0$ or $q - |\mathcal{I}|/n = 0$ and (8), (9) are natrually satisfied. Therefore, the inequalitites (8), (9) hold for arbitrary $(q_1 a_1, \cdots, q_t a_t, q)$.

By substituting (9) into $B(w, S)$, we have that for any $\delta > 0$, with probability at least $1 - \delta$,

$$\begin{aligned} B(w, S) &= \frac{1}{|\mathcal{I}|}\left(\mathbb{P}\left(L^w \notin \mathcal{L}_\gamma^w\right) - \frac{|\mathcal{I}|}{n}\right)\sum_{i \in \mathcal{I}} L_i^w \\ &\leq \frac{\sqrt{\mathbb{P}\left(L^w \notin \mathcal{L}_\gamma^w\right)}\sum_{i \in \mathcal{I}} L_i^w}{|\mathcal{I}|}\sqrt{\frac{2\log(1/\delta)}{n}}. \end{aligned} \tag{10}$$

Similarly, from (8), we have that for any $\delta > 0$ and $k \in [1, t]$, with probability at least $1 - \delta$,

$$\begin{aligned} C_k(w, S) &\leq \sqrt{\sum_{i=1}^{t} \mathbb{P}(L^w = a_i)a_i^2 - \mathbb{P}(L^w = a_k)a_k^2}\sqrt{\frac{2\log(1/\delta)}{n}} \\ &\leq b^w\sqrt{\sum_{i=1}^{t} \mathbb{P}(L^w = a_i) - \mathbb{P}(L^w = a_k)}\sqrt{\frac{2\log(1/\delta)}{n}} \\ &= b^w\sqrt{\mathbb{P}\left(L^w \in \mathcal{L}_\gamma^w \bigcap L^w \neq a_k\right)}\sqrt{\frac{2\log(1/\delta)}{n}} \\ &\leq b^w\sqrt{\frac{2\log(1/\delta)}{n}}. \end{aligned}$$

Taking the union bound over every $k \in [1, t]$, we have that for any $\delta > 0$, with probability at least $1 - \delta$, the following inequalities hold:

$$\begin{cases} C_1(w, S) \leq b^w\sqrt{\frac{2\log(t/\delta)}{n}}, \\ \qquad\qquad \vdots \\ C_t(w, S) \leq b^w\sqrt{\frac{2\log(t/\delta)}{n}}. \end{cases} \tag{11}$$

By substituting (11) into $C(w, S)$, we have that for any $\delta > 0$, with probability at least $1 - \delta$,

$$\begin{aligned} C(w, S) &= \sum_{k=1}^{t}\left(\mathbb{P}(L^w = a_k) - \frac{|\mathcal{I}_k|}{n}\right)a_k \\ &= \frac{1}{t-1}\sum_{k=1}^{t} C_k(w, S) \\ &\leq \frac{1}{t-1}\sum_{k=1}^{t} b^w\sqrt{\frac{2\log(t/\delta)}{n}} \end{aligned}$$

$$= \frac{t}{t-1} b^w \sqrt{\frac{2 \log(t/\delta)}{n}}.$$

For the extreme case where $t = 1$, we can similarly derive that for any $\delta > 0$, with probability at least $1 - \delta$,

$$C(w, S) = \left( \mathbb{P}(L^w = a_1) - \frac{|\mathcal{I}_1|}{n} \right) a_1$$

$$\leq a_1 \sqrt{\mathbb{P}(L^w = a_1)} \sqrt{\frac{2 \log(1/\delta)}{n}}$$

$$\leq b^w \sqrt{\frac{2 \log(t/\delta)}{n}}.$$

Therefore, for arbitrary $t \geq 1$, we have

$$C(w, S) \leq 2 b^w \sqrt{\frac{2 \log(t/\delta)}{n}}. \tag{12}$$

From Lemma 4, we know that

$$t = \left| \mathcal{L}_\gamma^w \right| \leq \exp \left( H(L^w) + c^w \sqrt{\frac{m \log(\sqrt{n}/\gamma)}{2}} \right).$$

Substituting into (12), we have

$$C(w, S) \leq 2 b^w \sqrt{\frac{2 \log(t) + 2 \log(1/\delta)}{n}}$$

$$\leq 2 b^w \sqrt{\frac{2 \left( H(L^w) + c^w \sqrt{\frac{m \log(\sqrt{n}/\gamma)}{2}} \right) + 2 \log(1/\delta)}{n}}. \tag{13}$$

Finally, by taking the union bound over (10) and (13), we have that for any $\delta > 0$, with probability at least $1 - \delta$, the following inequalities hold:

$$B(w, S) \leq \frac{\sqrt{\mathbb{P}\left( L^w \notin \mathcal{L}_\gamma^w \right)} \sum_{i \in \mathcal{I}} L_i^w}{|\mathcal{I}|} \sqrt{\frac{2 \log(2/\delta)}{n}},$$

$$C(w, S) \leq 2 b^w \sqrt{\frac{2(H(L^w) + C_4^w) + 2 \log(2/\delta)}{n}}.$$

$\square$

## B.4 COMPLETING THE PROOF

**Theorem 1** (Restate). *For any $\gamma > 0$ and $\delta > 0$, with probability at least $1 - \delta$, the following inequality holds:*

$$\Delta(w, S) \leq C_1^w \sqrt{\frac{H(L^w) + C_2^w}{n}} + \frac{C_3^w}{\sqrt{n}},$$

*where*

$$C_1^w = 2 b^w \sqrt{2},$$

$$C_2^w = c^w \sqrt{\frac{m \log(\sqrt{n}/\gamma)}{2}} + \log(2/\delta),$$

$$C_3^w = \gamma b^w + B^{w,S} \frac{\sqrt{\gamma}}{n^{1/4}} \sqrt{2 \log(2/\delta)}.$$

*Proof.* From Lemma 6, we know that for any $\gamma > 0$,

$$A(w, S) \leq \frac{\gamma b^w}{\sqrt{n}}. \tag{14}$$

Recall that in Lemma 4, we proved that

$$\mathbb{P}\big(L^w \notin \mathcal{L}_\gamma^w\big) \leq \frac{\gamma}{\sqrt{n}}.$$

By applying Lemma 7, we have that for any $\gamma > 0$ and $\delta > 0$, with probability at least $1 - \delta$,

$$
\begin{aligned}
B(w, S) &\leq \frac{\sqrt{\mathbb{P}\big(L^w \notin \mathcal{L}_\gamma^w\big) \sum_{i \in \mathcal{I}} L_i^w}}{|\mathcal{I}|} \sqrt{\frac{2\log(2/\delta)}{n}} \\
&\leq \frac{\sqrt{\gamma}}{n^{1/4}} \frac{1}{|\mathcal{I}|} \sum_{i \in \mathcal{I}} B^{w,S} \sqrt{\frac{2\log(2/\delta)}{n}} \\
&= B^{w,S} \frac{\sqrt{\gamma}}{n^{1/4}} \sqrt{\frac{2\log(2/\delta)}{n}},
\end{aligned}
\tag{15}
$$

and

$$
\begin{aligned}
C(w, S) &\leq 2b^w \sqrt{\frac{2(H(L^w) + C_4^w) + 2\log(2/\delta)}{n}} \\
&= C_1^w \sqrt{\frac{H(L^w) + C_2^w}{n}}.
\end{aligned}
\tag{16}
$$

Recall that in Lemma 5 we proved

$$\Delta(w, S) = A(w, S) + B(w, S) + C(w, S). \tag{17}$$

By substituting (14), (15) and (16) into (17), we finally get

$$
\begin{aligned}
\Delta(w, S) &\leq \frac{\gamma b^w}{\sqrt{n}} + B^{w,S} \frac{\sqrt{\gamma}}{n^{1/4}} \sqrt{\frac{2\log(2/\delta)}{n}} + C_1^w \sqrt{\frac{H(L^w) + C_2^w}{n}} \\
&= C_1^w \sqrt{\frac{H(L^w) + C_2^w}{n}} + \frac{1}{\sqrt{n}} \left( \gamma b^w + B^{w,S} \frac{\sqrt{\gamma}}{n^{1/4}} \sqrt{2\log(2/\delta)} \right) \\
&= C_1^w \sqrt{\frac{H(L^w) + C_2^w}{n}} + \frac{C_3^w}{\sqrt{n}}.
\end{aligned}
$$

$\square$

## C  PROOF OF THEOREM 2

Similarly, given the hypothesis $w \in \mathcal{W}$, we denote the set of all possible loss values per class by

$$\mathcal{L}_y^w = \{\ell(w, \theta_y(v), y) : v \in \mathcal{V}\}.$$

Let $L_y^w = L^w|_y$. For any $\gamma > 0$, define the typical subset of $\mathcal{L}_y^w$ by

$$\mathcal{L}_{\gamma,y}^w = \left\{ l \in \mathcal{L}_y^w : -\log p_{l|y}(l) - H(L_y^w) \leq c_y^w \sqrt{\frac{m\log(\sqrt{n}/\gamma)}{2}} \right\}, \tag{18}$$

where $p_{l|y}(l) = \mathbb{P}(L^w = l | Y = y) = \mathbb{P}\big(L_y^w = l\big)$.

### C.1  PROPERTIES OF THE CONDITIONAL TYPICAL SUBSET

We first show that $\mathcal{L}_{\gamma,y}^w$ shares exactly the same properties as the unconditional one:

**Lemma 8.** *For any $\gamma > 0$, we have*

$$\mathbb{P}\big(L_y^w \notin \mathcal{L}_{\gamma,y}^w\big) \leq \frac{\gamma}{\sqrt{n}},$$

*and*

$$|\mathcal{L}_{\gamma,y}^w| \leq \exp\left( H(L_y^w) + c_y^w \sqrt{\frac{m\log(\sqrt{n}/\gamma)}{2}} \right).$$

*Proof.* Consider the function $f_y(v) = -\log p_{l|y}(h_y(v))$, we have

$$
\begin{aligned}
\mathbb{E}_V[f_y(V)] &= -\sum_{v \in \mathcal{V}} p_v(v) \log p_{l|y}(h_y(v)) \\
&= -\sum_{l \in \mathcal{L}_y^w} \sum_{v \in h_y^{-1}(l)} p_v(v) \log p_{l|y}(h_y(v)) \\
&= -\sum_{l \in \mathcal{L}_y^w} \left( \sum_{v \in h_y^{-1}(l)} p_v(v) \right) \log p_{l|y}(l) \\
&= -\sum_{l \in \mathcal{L}_y^w} p_{l|y}(l) \log p_{l|y}(l) \\
&= H(L_y^w).
\end{aligned}
$$

Therefore, by applying McDiarmid's inequality on $f(V) = -\log p_{l|y}(L_y^w)$, we have

$$
\mathbb{P}\big(-\log p_{l|y}(L_y^w) - H(L_y^w) \geq \epsilon\big) \leq \exp\left(-\frac{2\epsilon^2}{m(c_y^w)^2}\right). \tag{19}
$$

Take $\delta$ as the RHS of (19), we have

$$
\epsilon = c_y^w \sqrt{\frac{m \log(1/\delta)}{2}}.
$$

Combining with (18), we select $\delta = \gamma/\sqrt{n}$ and

$$
\mathbb{P}\big(L_y^w \notin \mathcal{L}_{\gamma,y}^w\big) \leq \delta = \frac{\gamma}{\sqrt{n}}.
$$

Similar to the proof of Lemma 4, we can prove that

$$
\big|\mathcal{L}_{\gamma,y}^w\big| \exp\big(-H(L_y^w) - \epsilon\big) \leq 1,
$$

which further implies

$$
\big|\mathcal{L}_{\gamma,y}^w\big| \leq \exp\left(H(L_y^w) + c_y^w \sqrt{\frac{m \log(1/\delta)}{2}}\right).
$$

$\square$

## C.2 Decomposition of the Generalization Error

We further define $t_y = \big|\mathcal{L}_{\gamma,y}^w\big|$, the elements of the typical subset as $\mathcal{L}_{\gamma,y}^w = \{a_1^y, \cdots, a_{t_y}^y\}$, and

$$
\begin{aligned}
\mathcal{I}^y &= \big\{i \in [1,n] : L_i^w \notin \mathcal{L}_{\gamma,y}^w, Y_i = y\big\}, \\
\mathcal{I}_k^y &= \{i \in [1,n] : L_i^w = a_k^y, Y_i = y\}.
\end{aligned}
$$

We then use these notations to decompose the generalization error.

**Lemma 9.** *The generalization error $\Delta(w,S) = \tilde{A}(w,S) + \tilde{B}(w,S) + \tilde{C}(w,S)$, where*

$$
\tilde{A}(w,S) = \sum_{y \in \mathcal{Y}} \mathbb{P}\big(Y = y, L^w \notin \mathcal{L}_{\gamma,y}^w\big) \left( \mathbb{E}_{L^w}\big[L^w | Y = y, L^w \notin \mathcal{L}_{\gamma,y}^w\big] - \frac{1}{|\mathcal{I}^y|} \sum_{i \in I^y} L_i^w \right),
$$

$$
\tilde{B}(w,S) = \sum_{y \in \mathcal{Y}} \frac{1}{|\mathcal{I}^y|} \left( \mathbb{P}\big(Y = y, L^w \notin \mathcal{L}_{\gamma,y}^w\big) - \frac{|\mathcal{I}^y|}{n} \right) \sum_{i \in \mathcal{I}^y} L_i^w,
$$

$$
\tilde{C}(w,S) = \sum_{y \in \mathcal{Y}} \sum_{k=1}^{t_y} \left( \mathbb{P}\big(Y = y, L^w = a_k^y\big) - \frac{|\mathcal{I}_k^y|}{n} \right) a_k^y.
$$

*Proof.* The population risk conditioned on $Y = y$ can be decomposed as:

$$\mathbb{E}_{L_y^w}[L_y^w] = \mathbb{P}(L_y^w \notin \mathcal{L}_{\gamma,y}^w)\mathbb{E}_{L_y^w}[L_y^w|L_y^w \notin \mathcal{L}_{\gamma,y}^w] + \sum_{k=1}^{t_y} \mathbb{P}(L_y^w = a_k^y)\mathbb{E}_{L_y^w}[L_y^w|L_y^w = a_k^y]$$

$$= \mathbb{P}(L_y^w \notin \mathcal{L}_{\gamma,y}^w)\mathbb{E}_{L_y^w}[L_y^w|L_y^w \notin \mathcal{L}_{\gamma,y}^w] + \sum_{k=1}^{t_y} \mathbb{P}(L_y^w = a_k^y)a_k^y.$$

Summarising over $y \in \mathcal{Y}$, we have

$$\mathbb{E}_{L^w}[L^w] = \sum_{y\in\mathcal{Y}} \mathbb{P}(Y = y, L^w \notin \mathcal{L}_{\gamma,y}^w)\mathbb{E}_{L^w}[L^w|Y = y, L^w \notin \mathcal{L}_{\gamma,y}^w]$$

$$+ \sum_{y\in\mathcal{Y}}\sum_{k=1}^{t_y} \mathbb{P}(Y = y, L^w = a_k^y)a_k^y.$$

Similarly, we can decompose the empirical risk as:

$$\frac{1}{n}\sum_{i=1}^{n} L_i^w = \frac{1}{n}\sum_{y\in\mathcal{Y}}\left(\sum_{i\in\mathcal{I}^y} L_i^w + \sum_{k=1}^{t_y}\sum_{i\in\mathcal{I}_k^y} L_i^w\right)$$

$$= \frac{1}{n}\sum_{y\in\mathcal{Y}}\sum_{i\in\mathcal{I}^y} L_i^w + \sum_{y\in\mathcal{Y}}\sum_{k=1}^{t_y}\frac{1}{n}\sum_{i\in\mathcal{I}_k^y} a_k^y$$

$$= \frac{1}{n}\sum_{y\in\mathcal{Y}}\sum_{i\in\mathcal{I}^y} L_i^w + \sum_{y\in\mathcal{Y}}\sum_{k=1}^{t_y}\frac{|\mathcal{I}_k^y|}{n} a_k^y.$$

Substituting the inequalities above into $\Delta(w, S)$, we have

$$\Delta(w, S) = \sum_{y\in\mathcal{Y}} \mathbb{P}(Y = y, L^w \notin \mathcal{L}_{\gamma,y}^w)\mathbb{E}_{L^w}[L^w|Y = y, L^w \notin \mathcal{L}_{\gamma,y}^w]$$

$$- \sum_{y\in\mathcal{Y}} \mathbb{P}(Y = y, L^w \notin \mathcal{L}_{\gamma,y}^w)\left(\frac{1}{|\mathcal{I}^y|}\sum_{i\in I^y} L_i^w\right)$$

$$+ \sum_{y\in\mathcal{Y}} \mathbb{P}(Y = y, L^w \notin \mathcal{L}_{\gamma,y}^w)\left(\frac{1}{|\mathcal{I}^y|}\sum_{i\in I^y} L_i^w\right) - \sum_{y\in\mathcal{Y}}\frac{|\mathcal{I}^y|}{n}\left(\frac{1}{|\mathcal{I}^y|}\sum_{i\in\mathcal{I}^y} L_i^w\right)$$

$$+ \sum_{y\in\mathcal{Y}}\sum_{k=1}^{t_y} \mathbb{P}(Y = y, L^w = a_k^y)a_k^y - \sum_{y\in\mathcal{Y}}\sum_{k=1}^{t_y}\frac{|\mathcal{I}_k^y|}{n} a_k^y$$

$$= \sum_{y\in\mathcal{Y}} \mathbb{P}(Y = y, L^w \notin \mathcal{L}_{\gamma,y}^w)\left(\mathbb{E}_{L^w}[L^w|Y = y, L^w \notin \mathcal{L}_{\gamma,y}^w] - \frac{1}{|\mathcal{I}^y|}\sum_{i\in I^y} L_i^w\right)$$

$$+ \sum_{y\in\mathcal{Y}}\frac{1}{|\mathcal{I}^y|}\left(\mathbb{P}(Y = y, L^w \notin \mathcal{L}_{\gamma,y}^w) - \frac{|\mathcal{I}^y|}{n}\right)\sum_{i\in\mathcal{I}^y} L_i^w$$

$$+ \sum_{y\in\mathcal{Y}}\sum_{k=1}^{t_y}\left(\mathbb{P}(Y = y, L^w = a_k^y) - \frac{|\mathcal{I}_k^y|}{n}\right)a_k^y.$$

$\square$

## C.3 BOUNDING EACH TERM IN THE DECOMPOSITION

**Lemma 10.** *For any $\gamma > 0$, $\tilde{A}(w, S)$ satisfies:*

$$\tilde{A}(w, S) \leq \frac{\gamma b^w}{\sqrt{n}}.$$

*Additionally, for any $\delta > 0$, with probability at least $1 - \delta$, the following inequalities hold:*

$$\tilde{B}(w, S) \leq \sum_{y \in \mathcal{Y}} \frac{\sqrt{\mathbb{P}(Y = y, L^w \notin \mathcal{L}_{\gamma, y}^w)} \sum_{i \in \mathcal{I}^y} L_i^w}{|\mathcal{I}^y|} \sqrt{\frac{2 \log(2|\mathcal{Y}|/\delta)}{n}}$$

$$\tilde{C}(w, S) \leq 2b^w \sqrt{|\mathcal{Y}|} \sqrt{\frac{2(H(L^w|Y) + C_4^w) + 2 \log(2|\mathcal{Y}|/\delta)}{n}}.$$

*Proof.* From Lemma 8, we have

$$\mathbb{P}(L_y^w \notin \mathcal{L}_{\gamma, y}^w) \leq \frac{\gamma}{\sqrt{n}}.$$

By the fact that $L_i^w \geq 0$ for any $i \in [1, n]$, we can prove that

$$\tilde{A}(w, S) = \sum_{y \in \mathcal{Y}} \mathbb{P}(Y = y, L^w \notin \mathcal{L}_{\gamma, y}^w) \left( \mathbb{E}_{L^w}[L^w | Y = y, L^w \notin \mathcal{L}_{\gamma, y}^w] - \frac{1}{|\mathcal{I}^y|} \sum_{i \in I^y} L_i^w \right)$$

$$\leq \sum_{y \in \mathcal{Y}} \mathbb{P}(Y = y) \mathbb{P}(L_y^w \notin \mathcal{L}_{\gamma, y}^w) \mathbb{E}_{L^w}[L^w | Y = y, L^w \notin \mathcal{L}_{\gamma, y}^w]$$

$$\leq \sum_{y \in \mathcal{Y}} \mathbb{P}(Y = y) \frac{\gamma}{\sqrt{n}} b^w = \frac{\gamma b^w}{\sqrt{n}}.$$

Denote $q_k^y = \mathbb{P}(Y = y, L^w = a_k^y)$ for $k \in [1, t_y]$, $q^y = \mathbb{P}(Y = y, L^w \notin \mathcal{L}_{\gamma, y}^w)$ and let

$$\tilde{C}_k^y(w, S) = \sum_{i=1}^{t_y} \left( q_i^y - \frac{|\mathcal{I}_i^y|}{n} \right) a_i^y - \left( q_k^y - \frac{|\mathcal{I}_k^y|}{n} \right) a_k^y.$$

We apply Lemma 1 with

$$n = t_y + 1, \qquad X = \left( |\mathcal{I}_1^y|, \cdots, \left| \mathcal{I}_{t_y}^y \right|, |\mathcal{I}^y| \right), \qquad p = \left( q_1^k, \cdots, q_{t_y}^k, q^k \right),$$

$$m = n, \qquad \bar{a}_k = 0, \ \bar{a}_{t_y+1} = 0, \ \text{and} \ \bar{a}_i = a_i^y \ \text{for any} \ i \neq k.$$

Then for any $\epsilon > 0$, we have

$$\mathbb{P}\left( \tilde{C}_k^y(w, S) > \epsilon \right) \leq \exp\left( -\frac{n\epsilon^2}{2 \left( \sum_{i=1}^{t_y} q_i^y (a_i^y)^2 - q_k^y (a_k^y)^2 \right)} \right), \tag{20}$$

for any $k \in [1, t_y]$. Similarly, we can get

$$\mathbb{P}\left( q^y - \frac{|\mathcal{I}^y|}{n} > \epsilon \right) \leq \exp\left( -\frac{n\epsilon^2}{2q^y} \right). \tag{21}$$

Take $\delta$ as the RHS of (20) and (21) respectively, we have

$$\mathbb{P}\left( \tilde{C}_k^y(w, S) > \sqrt{\sum_{i=1}^{t_y} q_i^y (a_i^y)^2 - q_k^y (a_k^y)^2} \sqrt{\frac{2 \log(1/\delta)}{n}} \right) \leq \delta,$$

for any $k \in [1, t_y]$ and

$$q^y - \frac{|\mathcal{I}^y|}{n} \leq \sqrt{\frac{2q^y \log(1/\delta)}{n}}.$$

Take the union bound over $y \in \mathcal{Y}$, we have that for any $\delta > 0$, with probability at least $1 - \delta$, the following inequality holds for all $y \in \mathcal{Y}$ simultaneously:

$$\tilde{C}_k^y(w, S) \leq \sqrt{\sum_{i=1}^{t_y} q_i^y (a_i^y)^2 - q_k^y (a_k^y)^2} \sqrt{\frac{2 \log(|\mathcal{Y}|/\delta)}{n}}, \tag{22}$$

for any $k \in [1, t_y]$ and

$$\mathbb{P}\left( q^y - \frac{|\mathcal{I}^y|}{n} > \sqrt{\frac{2q^y \log(|\mathcal{Y}|/\delta)}{n}} \right) \leq \delta. \tag{23}$$

Substitute (23) into $\tilde{B}(w, S)$, we have that for any $\delta > 0$, with probability at least $1 - \delta$,

$$\tilde{B}(w, S) = \sum_{y \in \mathcal{Y}} \frac{1}{|\mathcal{I}^y|} \left( \mathbb{P}\big( Y = y, L^w \notin \mathcal{L}_{\gamma, y}^w \big) - \frac{|\mathcal{I}^y|}{n} \right) \sum_{i \in \mathcal{I}^y} L_i^w$$

$$\leq \sum_{y \in \mathcal{Y}} \frac{\sqrt{\mathbb{P}\big( Y = y, L^w \notin \mathcal{L}_{\gamma, y}^w \big)} \sum_{i \in \mathcal{I}^y} L_i^w}{|\mathcal{I}^y|} \sqrt{\frac{2 \log(|\mathcal{Y}|/\delta)}{n}}. \tag{24}$$

Similarly, from (22), we have that for any $\delta > 0$ and $k \in [1, t_y]$, with probability at least $1 - \delta$,

$$\tilde{C}_k^y(w, S) \leq \sqrt{\sum_{i=1}^{t_y} q_i^y (a_i^y)^2 - q_k^y (a_k^y)^2} \sqrt{\frac{2 \log(|\mathcal{Y}|/\delta)}{n}}$$

$$\leq b^w \sqrt{\sum_{i=1}^{t_y} q_i^y - q_k^y} \sqrt{\frac{2 \log(|\mathcal{Y}|/\delta)}{n}}$$

$$= b^w \sqrt{\mathbb{P}\Big( Y = y \bigcap L^w \in \mathcal{L}_{\gamma, y}^w \bigcap L^w \neq a_k^y \Big)} \sqrt{\frac{2 \log(|\mathcal{Y}|/\delta)}{n}}$$

$$\leq b^w \sqrt{\mathbb{P}(Y = y)} \sqrt{\frac{2 \log(|\mathcal{Y}|/\delta)}{n}}.$$

Taking the union bound over $k$, we have that for any $\delta > 0$, with probability at least $1 - \delta$, the following holds for all $k \in [1, t]$ and $y \in \mathcal{Y}$ simultaneously:

$$\tilde{C}_k^y(w, S) \leq b^w \sqrt{\mathbb{P}(Y = y)} \sqrt{\frac{2 \log(t_y |\mathcal{Y}|/\delta)}{n}}. \tag{25}$$

Substitute (25) into $\tilde{C}(w, S)$, we have that for any $\delta > 0$, with probability at least $1 - \delta$,

$$\tilde{C}(w, S) = \sum_{y \in \mathcal{Y}} \sum_{k=1}^{t_y} \left( \mathbb{P}(Y = y, L^w = a_k^y) - \frac{|\mathcal{I}_k^y|}{n} \right) a_k^y$$

$$= \sum_{y \in \mathcal{Y}} \frac{1}{t_y - 1} \sum_{k=1}^{t_y} \tilde{C}_k^y(w, S)$$

$$\leq \sum_{y \in \mathcal{Y}} \frac{1}{t_y - 1} \sum_{k=1}^{t_y} b^w \sqrt{\mathbb{P}(Y = y)} \sqrt{\frac{2 \log(t_y |\mathcal{Y}|/\delta)}{n}}$$

$$= \sum_{y \in \mathcal{Y}} b^w \sqrt{\mathbb{P}(Y = y)} \frac{t_y}{t_y - 1} \sqrt{\frac{2 \log(t_y |\mathcal{Y}|/\delta)}{n}}$$

$$\leq 2 \sum_{y \in \mathcal{Y}} b^w \sqrt{\mathbb{P}(Y = y)} \sqrt{\frac{2 \log(t_y |\mathcal{Y}|/\delta)}{n}}.$$

Recall Lemma 8 which proves that

$$t_y = \big| \mathcal{L}_{\gamma, y}^w \big| \leq \exp\left( H(L_y^w) + c_y^w \sqrt{\frac{m \log(\sqrt{n}/\gamma)}{2}} \right).$$

By applying Jensen's inequality, we have

$$\tilde{C}(w, S) \leq 2 b^w \sum_{y \in \mathcal{Y}} \sqrt{\mathbb{P}(Y = y)} \sqrt{\frac{2 \log(t_y) + 2 \log(|\mathcal{Y}|/\delta)}{n}}$$

$$\leq 2b^w \sum_{y\in\mathcal{Y}} \sqrt{\mathbb{P}(Y=y)} \sqrt{\frac{2\left(H(L_y^w)+c_y^w\sqrt{\frac{m\log(\sqrt{n}/\gamma)}{2}}\right)+2\log(|\mathcal{Y}|/\delta)}{n}}$$

$$\leq 2b^w \sqrt{|\mathcal{Y}|} \sqrt{\sum_{y\in\mathcal{Y}} \mathbb{P}(Y=y)\frac{2\left(H(L_y^w)+c_y^w\sqrt{\frac{m\log(\sqrt{n}/\gamma)}{2}}\right)+2\log(|\mathcal{Y}|/\delta)}{n}}$$

$$= 2b^w \sqrt{|\mathcal{Y}|} \sqrt{\frac{2\left(H(L^w|Y)+c^w\sqrt{\frac{m\log(\sqrt{n}/\gamma)}{2}}\right)+2\log(|\mathcal{Y}|/\delta)}{n}} \tag{26}$$

Finally, taking the union bound over (24) and (26), we have that for any $\delta > 0$, with probability at least $1 - \delta$, the following inequalities hold:

$$\tilde{B}(w,S) \leq \sum_{y\in\mathcal{Y}} \frac{\sqrt{\mathbb{P}(Y=y, L^w\notin\mathcal{L}_{\gamma,y}^w)}\sum_{i\in\mathcal{I}^y} L_i^w}{|\mathcal{I}^y|}\sqrt{\frac{2\log(2|\mathcal{Y}|/\delta)}{n}}$$

$$\tilde{C}(w,S) \leq 2b^w\sqrt{|\mathcal{Y}|}\sqrt{\frac{2(H(L^w|Y)+C_4^w)+2\log(2|\mathcal{Y}|/\delta)}{n}}.$$

$\square$

## C.4 Completing the Proof

**Theorem 2** (Restate). *For any $\gamma > 0$ and $\delta > 0$, with probability at least $1 - \delta$, the following inequality holds:*

$$\Delta(w,S) \leq \tilde{C}_1^w \sqrt{\frac{H(L^w|Y)+\tilde{C}_2^w}{n}}+\frac{\tilde{C}_3^w}{\sqrt{n}},$$

*where*

$$\tilde{C}_1^w = 2b^w\sqrt{2|\mathcal{Y}|},$$

$$\tilde{C}_2^w = c^w\sqrt{\frac{m\log(\sqrt{n}/\gamma)}{2}}+\log(2|\mathcal{Y}|/\delta),$$

$$\tilde{C}_3^w = \gamma b^w + B^{w,S}\frac{\sqrt{\gamma|\mathcal{Y}|}}{n^{1/4}}\sqrt{2\log(2|\mathcal{Y}|/\delta)}.$$

*Proof.* Recall from Lemma 8 that

$$\mathbb{P}\left(\mathcal{L}_y^w\notin\mathcal{L}_{\gamma,y}^w\right)\leq\frac{\gamma}{\sqrt{n}}.$$

Combining with Lemma 10, we have that for any $\gamma > 0$ and $\delta > 0$, with probability at least $1 - \delta$,

$$\tilde{B}(w,S) \leq \sum_{y\in\mathcal{Y}} \frac{\sqrt{\mathbb{P}(Y=y, L^w\notin\mathcal{L}_{\gamma,y}^w)}\sum_{i\in\mathcal{I}^y} L_i^w}{|\mathcal{I}^y|}\sqrt{\frac{2\log(2|\mathcal{Y}|/\delta)}{n}}$$

$$= \sum_{y\in\mathcal{Y}} \sqrt{\mathbb{P}(Y=y)}\sqrt{\mathbb{P}(L_y^w\notin\mathcal{L}_{\gamma,y}^w)}\frac{1}{|\mathcal{I}^y|}\left(\sum_{i\in\mathcal{I}^y} L_i^w\right)\sqrt{\frac{2\log(2|\mathcal{Y}|/\delta)}{n}}$$

$$\leq \frac{\sqrt{\gamma}}{n^{1/4}}\frac{1}{|\mathcal{I}^y|}\left(\sum_{i\in\mathcal{I}^y} B^{w,S}\right)\sqrt{|\mathcal{Y}|}\sqrt{\sum_{y\in\mathcal{Y}}\mathbb{P}(Y=y)}\sqrt{\frac{2\log(2|\mathcal{Y}|/\delta)}{n}}$$

$$\leq B^{w,S}\frac{\sqrt{\gamma|\mathcal{Y}|}}{n^{1/4}}\sqrt{\frac{2\log(2|\mathcal{Y}|/\delta)}{n}}.$$

and

$$\tilde{C}(w, S) \le 2b^w \sqrt{|\mathcal{Y}|} \sqrt{\frac{2(H(L^w|Y) + C_4^w) + 2\log(2|\mathcal{Y}|/\delta)}{n}}$$

$$= \tilde{C}_1^w \sqrt{\frac{H(L^w|Y) + \tilde{C}_2^w}{n}}.$$

By applying Lemma 9, we can get

$$\Delta(w, S) = \tilde{A}(w, S) + \tilde{B}(w, S) + \tilde{C}(w, S)$$

$$\le \frac{\gamma b^w}{\sqrt{n}} + B^{w,S} \frac{\sqrt{\gamma|\mathcal{Y}|}}{n^{1/4}} \sqrt{\frac{2\log(2|\mathcal{Y}|/\delta)}{n}} + \tilde{C}_1^w \sqrt{\frac{H(L^w|Y) + \tilde{C}_2^w}{n}}$$

$$\le \tilde{C}_1^w \sqrt{\frac{H(L^w|Y) + \tilde{C}_2^w}{n}} + \frac{1}{\sqrt{n}} \left( \gamma b^w + B^{w,S} \frac{\sqrt{\gamma|\mathcal{Y}|}}{n^{1/4}} \sqrt{2\log(2|\mathcal{Y}|/\delta)} \right)$$

$$= \tilde{C}_1^w \sqrt{\frac{H(L^w|Y) + \tilde{C}_2^w}{n}} + \frac{\tilde{C}_3^w}{\sqrt{n}}.$$

$\square$

# D   PROOF OF DATA-DEPENDENT BOUNDS

Let $R \in \mathcal{R}$ be any discrete random variable. Denote $p_r(r) = \mathbb{P}(R = r)$. For any $\lambda > 0$, define

$$C_\lambda = \frac{1}{e^{\lambda H(R)}} \sum_{r \in \mathcal{R}} p_r^{1-\lambda}(r).$$

For any $\epsilon > 0$, we define the typical subset

$$\mathcal{R}_\epsilon = \{r \in \mathcal{R} : -\log p_r(r) - H(R) \le \epsilon\}.$$

## D.1   PROPERTIES OF THE TYPICAL SUBSET

**Lemma 11.** *For any $\lambda > 0$, by taking $\epsilon = \frac{1}{\lambda}\log(C_\lambda/\delta)$, we have*

$$\mathbb{P}(R \notin \mathcal{R}_\epsilon) \le \delta,$$

*and*

$$|\mathcal{R}_\epsilon| \le \exp\left( H_{1-\lambda}(R) + \frac{1}{\lambda}\log\left(\frac{1}{\delta}\right) \right).$$

*Proof.* From the definition of the typical subset and by applying Markov's inequality, we have

$$\mathbb{P}(R \notin \mathcal{R}_\epsilon) = \mathbb{P}(-\log p_r(R) \ge H(R) + \epsilon)$$

$$= \mathbb{P}(-\lambda \log p_r(R) \ge \lambda H(R) + \lambda\epsilon)$$

$$= \mathbb{P}(p_r^{-\lambda}(R) \ge \exp(\lambda H(R) + \lambda\epsilon))$$

$$\le \frac{\mathbb{E}_R[p_r^{-\lambda}(R)]}{\exp(\lambda H(R) + \lambda\epsilon)} \tag{27}$$

$$= \frac{\sum_{r \in \mathcal{R}} p_r^{1-\lambda}(r)}{\exp(\lambda H(R) + \lambda\epsilon)} = \frac{C_\lambda}{e^{\lambda\epsilon}}. \tag{28}$$

Now we compute the size of $\mathcal{R}_\epsilon$:

$$-\log p_r(R) - H(R) \le \epsilon$$

$$-\log p_r(R) \le H(R) + \epsilon$$

$$-H(R) - \epsilon \le \log p_r(R)$$

$$\exp(-H(R) - \epsilon) \le p_r(R).$$

This further implies that

$$1 \ge \mathbb{P}(R \in \mathcal{R}_\epsilon) = \sum_{r \in \mathcal{R}_\epsilon} p_r(r) \ge \sum_{r \in \mathcal{R}_\epsilon} \exp(-H(R) - \epsilon) = |\mathcal{R}_\epsilon| \exp(-H(R) - \epsilon).$$

By taking $C_\lambda/e^{\lambda \epsilon} = \delta$ in (28), i.e. $\epsilon = \frac{1}{\lambda} \log(C_\lambda/\delta)$, we have

$$|\mathcal{R}_\epsilon| \le \exp\left( H(R) + \frac{1}{\lambda} \log\left(\frac{C_\lambda}{\delta}\right) \right)$$

$$= \exp\left( H(R) + \frac{1}{\lambda} \log\left(\frac{1}{\delta}\right) + \frac{1}{\lambda} \log\left( \frac{1}{e^{\lambda H(R)}} \sum_{r \in \mathcal{R}} p_r^{1-\lambda}(r) \right) \right)$$

$$= \exp\left( H(R) + \frac{1}{\lambda} \log\left(\frac{1}{\delta}\right) - \frac{1}{\lambda} \lambda H(R) + \frac{1}{\lambda} \log \sum_{r \in \mathcal{R}} p_r^{1-\lambda}(r) \right)$$

$$= \exp\left( H_{1-\lambda}(R) + \frac{1}{\lambda} \log\left(\frac{1}{\delta}\right) \right).$$

$\square$

## D.2 COMPLETING THE PROOF

**Theorem 3** (Restate). *For any $\lambda \in (0,1)$ and $\delta > 0$, with probability at least $1 - \delta$, the following inequality holds:*

$$\Delta\left(W, \tilde{S}_l, U\right) \le C_1^W \sqrt{H_{1-\lambda}(R^W) + C_2^W},$$

*where*

$$C_1^W = \sqrt{2} \Sigma_{R^W},$$

$$C_2^W = \frac{1}{\lambda} \log\left(\frac{1}{\delta}\right) + \log\left(\frac{2}{\delta}\right),$$

*by assuming that $\frac{n+1}{n}(L_U^W - \bar{L}^W)$ is $\Sigma_{R^W}$-subgaussian w.r.t $U$ for $\Sigma_{R^W} \in [0, B^{W, \tilde{S}_l}]$.*

*Proof.* Assume that $R^W = r$ for some $r = \{l_i\}_{i=1}^{n+1} \in \mathcal{R}^W$, we then have

$$\Delta\left(W, \tilde{S}_l, U\right) = l_U - \frac{1}{n} \sum_{i \neq U} l_i$$

$$= l_U - \frac{1}{n} \sum_{i=1}^{n+1} l_i + \frac{1}{n} l_U$$

$$= \frac{n+1}{n} l_U - \frac{n+1}{n} \bar{l}$$

$$= \frac{n+1}{n} (l_U - \bar{l}),$$

where $\bar{l} = \frac{1}{n+1} \sum_{i=1}^{n+1} l_i$. It is easy to verify that $\mathbb{E}_U\left[\Delta\left(W, \tilde{S}_l, U\right)\right] = 0$. When $l_U = b^{W, \tilde{S}_l} = \sup_{i \in [1, n+1]} l_i$ and $l_i = 0$ for any $i \neq U$, $\Delta\left(W, \tilde{S}_l, U\right)$ takes the maximum value of $b^{W, \tilde{S}_l}$. Similarly, one can prove that $\Delta\left(W, \tilde{S}_l, U\right) \ge -b^{W, \tilde{S}_l}$, which implies that $\Delta\left(W, \tilde{S}_l, U\right)$ is $b^{W, \tilde{S}_l}$-subgaussian. Assume that $R^W = r$ and let $\Delta\left(W, \tilde{S}_l, U\right)$ be $\sigma_r$-subgaussian w.r.t $U$, where $\sigma_r \in [0, b^{W, \tilde{S}_l}]$, then for any $t > 0$,

$$\mathbb{P}_U\left(\Delta\left(W, \tilde{S}_l, U\right) \ge t\right) \le \exp\left( -\frac{t^2}{2(\sigma_r)^2} \right).$$

That is, for any $\delta > 0$ and $r \in \mathcal{R}^W$, if $R^W = r$, then with probability at least $1 - \delta$,

$$\Delta\left(W, \tilde{S}_l, U\right) \leq \sigma_r \sqrt{2 \log(1/\delta)}. \tag{29}$$

From Lemma 11, we know that for any $\delta > 0$,

$$\mathbb{P}\left(R^W \notin \mathcal{R}_\epsilon^W\right) \leq \delta, \tag{30}$$

$$\left|\mathcal{R}_\epsilon^W\right| \leq \exp\left(H_{1-\lambda}(R^W) + \frac{1}{\lambda} \log\left(\frac{1}{\delta}\right)\right). \tag{31}$$

Taking the union bound of (29) over every $r \in \mathcal{R}_\epsilon^W$, we have that for any $\delta > 0$, with probability at least $1 - \delta$, the following inequality holds for all $r \in \mathcal{R}_\epsilon^W$ simultaneously if $R^W = r$:

$$\Delta\left(W, \tilde{S}_l, U\right) \leq \sigma_r \sqrt{2 \log(|\mathcal{R}_\epsilon^W|/\delta)}.$$

Therefore, if $R^W \in \mathcal{R}_\epsilon^W$, then we have that for any $\delta > 0$, with probability at least $1 - \delta$,

$$\Delta\left(W, \tilde{S}_l, U\right) \leq \Sigma_{R^W} \sqrt{2 \log(|\mathcal{R}_\epsilon^W|/\delta)}. \tag{32}$$

Again, take the union bound over (32) and (30), we have that for arbitrary $R^W \in \mathcal{R}^W$ and $\delta > 0$, with probability at least $1 - \delta$,

$$\Delta\left(W, \tilde{S}_l, U\right) \leq \Sigma_{R^W} \sqrt{2 \log(2|\mathcal{R}_\epsilon^W|/\delta)}. \tag{33}$$

By substituting (31) into the inequality above, we can get

$$\Delta\left(W, \tilde{S}_l, U\right) \leq \Sigma_{R^W} \sqrt{2\left(H_{1-\lambda}(R^W) + \frac{1}{\lambda} \log\left(\frac{1}{\delta}\right)\right) + 2 \log\left(\frac{2}{\delta}\right)}$$

$$= C_1^W \sqrt{H_{1-\lambda}(R^W) + C_2^W}.$$

$\square$

**Theorem 4** (Restate). *For any $\lambda \in (0, 1)$ and $\delta > 0$, with probability at least $1 - \delta$, the following inequality holds:*

$$\Delta\left(W, \tilde{S}_s, \tilde{U}\right) \leq \tilde{C}_1^W \sqrt{\frac{H_{1-\lambda}(\tilde{R}_\Delta^W) + \tilde{C}_2^W}{n}},$$

*where*

$$\tilde{C}_1^W = \sqrt{\frac{2}{n} \sum_{i=1}^n (\Delta L_i^W)^2},$$

$$\tilde{C}_2^W = \frac{1}{\lambda} \log\left(\frac{1}{\delta}\right) + \log\left(\frac{2}{\delta}\right).$$

*Proof.* Assume that $\tilde{R}_\Delta^W = r$ for some $r = \{\Delta l_i\}_{i=1}^n \in \tilde{\mathcal{R}}_\Delta^W$, we then have

$$\Delta\left(W, \tilde{S}_s, \tilde{U}\right) = L_{\bar{S}}(W) - L_S(W)$$

$$= \frac{1}{n} \sum_{i=1}^n L_{i,1-\tilde{U}}^W - L_{i,\tilde{U}}^W$$

$$= \frac{1}{n} \sum_{i=1}^n (-1)^{\tilde{U}_i} \Delta l_i.$$

Notice that $\mathbb{E}_{\tilde{U}_i}[(-1)^{\tilde{U}_i}] = 0$, by applying McDiarmid's inequality with $f(\tilde{U}) = \Delta\left(W, \tilde{S}_s, \tilde{U}\right)$, we have that when $\tilde{R}_\Delta^W = r$, for any $t > 0$:

$$\mathbb{P}_{\tilde{U}}\left(\Delta\left(W, \tilde{S}_s, \tilde{U}\right) \geq t\right) \leq \exp\left(-\frac{2t^2}{\sum_{i=1}^n (2\Delta l_i/n)^2}\right).$$

That is, for any $\delta > 0$ and $r \in \tilde{\mathcal{R}}_{\Delta,\epsilon}^W$, if $\tilde{R}_\Delta^W = r$, then with probability at least $1 - \delta$,

$$\Delta\left(W, \tilde{S}_s, \tilde{U}\right) \leq \sqrt{\frac{1}{n}\sum_{i=1}^{n}(\Delta l_i)^2}\sqrt{\frac{2\log(1/\delta)}{n}}.$$

From Lemma 11, we know that for any $\delta > 0$,

$$\mathbb{P}\left(\tilde{R}_\Delta^W \notin \tilde{\mathcal{R}}_{\Delta,\epsilon}^W\right) \leq \delta, \tag{34}$$

$$\left|\tilde{\mathcal{R}}_{\Delta,\epsilon}^W\right| \leq \exp\left(H_{1-\lambda}(\tilde{R}_\Delta^W) + \frac{1}{\lambda}\log\left(\frac{1}{\delta}\right)\right). \tag{35}$$

Taking the union bound over every $r \in \tilde{\mathcal{R}}_{\Delta,\epsilon}^W$, we have that for any $\delta > 0$, with probability at least $1 - \delta$, the following bounds holds for all $r \in \tilde{R}_{\Delta,\epsilon}^W$ simultaneously if $\tilde{R}_\Delta^W = r$:

$$\Delta\left(W, \tilde{S}_s, \tilde{U}\right) \leq \sqrt{\frac{1}{n}\sum_{i=1}^{n}(\Delta l_i)^2}\sqrt{\frac{2\log\left(\left|\tilde{R}_{\Delta,\epsilon}^W\right|/\delta\right)}{n}}.$$

Therefore, if $\tilde{R}_\Delta^W \in \tilde{\mathcal{R}}_{\Delta,\epsilon}^W$, we have that for any $\delta > 0$, with probability at least $1 - \delta$,

$$\Delta\left(W, \tilde{S}_s, \tilde{U}\right) \leq \sqrt{\frac{1}{n}\sum_{i=1}^{n}(\Delta L_i^W)^2}\sqrt{\frac{2\log\left(\left|\tilde{\mathcal{R}}_{\Delta,\epsilon}^W\right|/\delta\right)}{n}}. \tag{36}$$

Again, take the union bound over (36) and (34), we have that for arbitrary $\tilde{R}_\Delta^W \in \tilde{\mathcal{R}}_\Delta^W$ and $\delta > 0$, with probability at least $1 - \delta$,

$$\Delta\left(W, \tilde{S}_s, \tilde{U}\right) \leq \sqrt{\frac{1}{n}\sum_{i=1}^{n}(\Delta L_i^W)^2}\sqrt{\frac{2\log\left(2\left|\tilde{\mathcal{R}}_{\Delta,\epsilon}^W\right|/\delta\right)}{n}}. \tag{37}$$

By substituting (35) into the inequality above, we have

$$\log\left(2\left|\tilde{\mathcal{R}}_{\Delta,\epsilon}^W\right|/\delta\right) = \log\left(\left|\tilde{\mathcal{R}}_{\Delta,\epsilon}^W\right|\right) + \log\left(\frac{2}{\delta}\right)$$

$$\leq H_{1-\lambda}(\tilde{R}_\Delta^W) + \frac{1}{\lambda}\log\left(\frac{1}{\delta}\right) + \log\left(\frac{2}{\delta}\right)$$

$$= H_{1-\lambda}(\tilde{R}_\Delta^W) + \tilde{C}_2^W.$$

Substituting this into (37), we have that for any $\delta > 0$, with probability at least $1 - \delta$,

$$\Delta\left(W, \tilde{S}_s, \tilde{U}\right) \leq \tilde{C}_1^W\sqrt{\frac{H_{1-\lambda}(\tilde{R}_\Delta^W) + \tilde{C}_2^W}{n}}.$$

$\square$

## D.3 FAST-RATE GENERALIZATION BOUNDS

We then prove our fast-rate generalization bounds for the validation error under the supersample setting. To overcome the loose $B^{W,\tilde{S}_s}$ factor in previous fast-rate bounds, we adopt a thresholding strategy based on a simple observation: for any $\kappa > 0$, the error could be decomposed by:

$$L = \min(\kappa, L) + \max(L - \kappa, 0).$$

For convenience, we denote $L^\kappa = \min(\kappa, L)$ and $L^{-\kappa} = \max(L - \kappa, 0)$, then $L = L^\kappa + L^{-\kappa}$.

**Lemma 12.** *For any $\kappa > 0$, $\lambda \in (0,1)$ and $\delta > 0$, with probability at least $1 - \delta$, the following inequality holds:*

$$\frac{1}{n} \sum_{i=1}^{n} L_{i,1-\tilde{U}_i}^{W,\kappa} - (1 + C_i) L_{i,\tilde{U}_i}^{W,\kappa} \leq \frac{H_{1-\lambda}(\tilde{R}^{W,\kappa}) + \frac{1}{\lambda}\log(1/\delta) + \log(4/\delta)}{n\eta},$$

*where $\tilde{R}^{W,\kappa} = \{L_{i,0}^{W,\kappa}, L_{i,1}^{W,\kappa}\}_{i=1}^{n}$ and*

$$\eta \in \left(0, \frac{\log 2}{2\kappa}\right), \quad C_i = -\frac{\log\left(2 - e^{2\eta\hat{L}_i^{W,\kappa}}\right)}{2\eta\hat{L}_i^{W,\kappa}} - 1, \quad \hat{L}_i^{W,\kappa} = \max\left(L_{i,0}^{W,\kappa}, L_{i,1}^{W,\kappa}\right),$$

*for any $i \in [1, n]$.*

*Proof.* Assume that $\tilde{R}^{W,\kappa} = r$ for some $r = \{l_{i,0}, l_{i,1}\}_{i=1}^{n} \in \tilde{\mathcal{R}}^{W,\kappa}$, the values of $C_i, i \in [1,n]$ are then deterministic given $r$. We then have

$$\mathbb{P}\left(\frac{1}{n}\sum_{i=1}^{n} l_{i,1-\tilde{U}_i} - (1 + C_i)l_{i,\tilde{U}_i} \geq t\right)$$

$$= \mathbb{P}\left(\frac{1}{n}\sum_{i=1}^{n}\left(1 + \frac{C_i}{2}\right)\left(l_{i,1-\tilde{U}_i} - l_{i,\tilde{U}_i}\right) - \frac{C_i}{2}l_{i,1-\tilde{U}_i} - \frac{C_i}{2}l_{i,\tilde{U}_i} \geq t\right)$$

$$= \mathbb{P}\left(\frac{1}{2n}\sum_{i=1}^{n}\left((-1)^{\tilde{U}_i}(2 + C_i)l_{i,1} - C_i l_{i,1}\right)\right.$$

$$\left. + \frac{1}{2n}\sum_{i=1}^{n}\left(-(-1)^{\tilde{U}_i}(2 + C_i)l_{i,0} - C_i l_{i,0}\right) \geq t\right)$$

$$\leq \mathbb{P}\left(\sup_{I \in \{0,1\}}\left\{\frac{1}{2n}\sum_{i=1}^{n}\left((-1)^{\tilde{U}_i}(2 + C_i) - C_i\right)l_{i,I}\right.\right.$$

$$\left.\left. + \frac{1}{2n}\sum_{i=1}^{n}\left(-(-1)^{\tilde{U}_i}(2 + C_i) - C_i\right)l_{i,1-I}\right\} \geq t\right)$$

$$\leq \mathbb{P}\left(\sup_{I \in \{0,1\}}\left\{\frac{1}{2n}\sum_{i=1}^{n}\left((-1)^{\tilde{U}_i}(2 + C_i) - C_i\right)l_{i,I}\right\}\right.$$

$$\left. + \sup_{I \in \{0,1\}}\left\{\frac{1}{2n}\sum_{i=1}^{n}\left(-(-1)^{\tilde{U}_i}(2 + C_i) - C_i\right)l_{i,1-I}\right\} \geq t\right)$$

$$\leq \inf_{\gamma \in (0,1)} \mathbb{P}\left(\sup_{I \in \{0,1\}}\left\{\frac{1}{2n}\sum_{i=1}^{n}\left((-1)^{\tilde{U}_i}(2 + C_i) - C_i\right)l_{i,I}\right\} \geq \gamma t\right)$$

$$+ \mathbb{P}\left(\sup_{I \in \{0,1\}}\left\{\frac{1}{2n}\sum_{i=1}^{n}\left(-(-1)^{\tilde{U}_i}(2 + C_i) - C_i\right)l_{i,1-I}\right\} \geq (1 - \gamma)t\right).$$

Observe that the two events above share the same marginal distribution, this implies that

$$\mathbb{P}\left(\frac{1}{n}\sum_{i=1}^{n} l_{i,1-\tilde{U}_i} - (1 + C_i)l_{i,\tilde{U}_i} \geq t\right)$$

$$\leq \inf_{\gamma \in (0,1)} \mathbb{P}\left(\sup_{I \in \{0,1\}}\left\{\frac{1}{2n}\sum_{i=1}^{n}\left((-1)^{\tilde{U}_i}(2 + C_i) - C_i\right)l_{i,I}\right\} \geq \gamma t\right)$$

$$+ \mathbb{P}\left(\sup_{I \in \{0,1\}}\left\{\frac{1}{2n}\sum_{i=1}^{n}\left((-1)^{\tilde{U}_i}(2 + C_i) - C_i\right)l_{i,I}\right\} \geq (1 - \gamma)t\right). \tag{38}$$

Since the values of $C_i, i \in [1, n]$ are determined by $\tilde{R}^{W,\kappa} = r$, they are independent of $\tilde{U}$. For any $I \in \{0, 1\}$, $t > 0$ and $\eta > 0$, by applying Markov's inequality, we then have

$$\mathbb{P}\left(\frac{1}{2n}\sum_{i=1}^{n}\left((-1)^{\tilde{U}_i}(2 + C_i) - C_i\right)l_{i,I} \ge t\right)$$

$$= \mathbb{P}\left(\exp\left(\eta\sum_{i=1}^{n}\left((-1)^{\tilde{U}_i}(2 + C_i) - C_i\right)l_{i,I}\right) \ge e^{2\eta nt}\right)$$

$$\le e^{-2\eta nt}\mathbb{E}_{\tilde{U}}\left[\exp\left(\eta\sum_{i=1}^{n}\left((-1)^{\tilde{U}_i}(2 + C_i) - C_i\right)l_{i,I}\right)\right]$$

$$= e^{-2\eta nt}\prod_{i=1}^{n}\mathbb{E}_{\tilde{U}_i}\left[\exp\left(\eta\left((-1)^{\tilde{U}_i}(2 + C_i) - C_i\right)l_{i,I}\right)\right]$$

$$= e^{-2\eta nt}\prod_{i=1}^{n}\frac{e^{-2\eta l_{i,I}(1+C_i)} + e^{2\eta l_{i,I}}}{2}.$$

We intend to carefully select the values of $\eta$ and $C_i$, such that $e^{-2\eta l_{i,I}(1+C_i)} + e^{2\eta l_{i,I}} \le 2$ is satisfied for any $i \in [1, n]$ and $I \in \{0, 1\}$. Notice that $e^{2\eta l_{i,I}} \le 2$ implies $2\eta l_{i,I} \le \log 2 < 1$. Furthermore, since $e^{-2\eta l_{i,I}(1+C_i)} + e^{2\eta l_{i,I}}$ decreases monotonically with the increase of $C_i$, it is sufficient to select a large enough $C_i$ that satisfies:

$$\frac{2 - e^{2\eta l_{i,I}}}{e^{-2\eta l_{i,I}}} \ge e^{-2\eta l_{i,I}C_i}.$$

Solving the inequality above yields $C_i \ge -\log(2 - e^{2\eta l_{i,I}})/2\eta l_{i,I} - 1$. It is easy to verify that this lower bound increases monotonically with the increase of $l_{i,I}$. Therefore, if we choose the value of $C_i$ by

$$C_i \ge -\frac{\log(2 - e^{2\eta\max(l_{i,0}, l_{i,1})})}{2\eta\max(l_{i,0}, l_{i,1})} - 1,$$

we will have

$$\frac{e^{-2\eta l_{i,I}(1+C_i)} + e^{2\eta l_{i,I}}}{2} \le 1,$$

for any $i \in [1, n]$ and the following inequality can hold:

$$\mathbb{P}\left(\frac{1}{2n}\sum_{i=1}^{n}\left((-1)^{\tilde{U}_i}(2 + C_i) - C_i\right)l_{i,I} \ge t\right) \le e^{-2\eta nt}. \tag{39}$$

Take the union bound of (39) over $I \in \{0, 1\}$, we can get

$$\mathbb{P}\left(\sup_{I\in\{0,1\}}\left\{\frac{1}{2n}\sum_{i=1}^{n}\left((-1)^{\tilde{U}_i}(2 + C_i) - C_i\right)l_{i,I}\right\} \ge t\right) \le 2e^{-2\eta nt}. \tag{40}$$

Substituting (40) into (38), we have

$$\mathbb{P}\left(\frac{1}{n}\sum_{i=1}^{n}l_{i,1-\tilde{U}_i} - (1 + C_i)l_{i,\tilde{U}_i} \ge t\right)$$

$$\le \inf_{\gamma\in(0,1)} 2e^{-2\eta\gamma nt} + 2e^{-2\eta(1-\gamma)nt}$$

$$= 2e^{-\eta nt} + 2e^{-\eta nt} = 4e^{-\eta nt}. \tag{41}$$

By selecting $\delta$ as the RHS of (41), we have that for any $\delta > 0$, with probability at least $1 - \delta$,

$$\frac{1}{n}\sum_{i=1}^{n}l_{i,1-\tilde{U}_i} - (1 + C_i)l_{i,\tilde{U}_i} \le \frac{\log(4/\delta)}{n\eta}.$$

From Lemma 11, we know that for any $\delta > 0$,

$$\mathbb{P}\left(\tilde{R}^{W,\kappa} \notin \tilde{\mathcal{R}}_\epsilon^{W,\kappa}\right) \le \delta, \tag{42}$$

$$\left|\tilde{\mathcal{R}}_\epsilon^{W,\kappa}\right| \le \exp\left(H_{1-\lambda}(\tilde{R}^{W,\kappa}) + \frac{1}{\lambda}\log\left(\frac{1}{\delta}\right)\right). \tag{43}$$

Notice that the marginal distribution of $\tilde{R}^{W,\kappa}$ is symmetric due to the existence of $\tilde{U}$, i.e. if $\{l_{i,0}, l_{i,1}\}_{i=1}^n \in \tilde{\mathcal{R}}_\epsilon^{W,\kappa}$, then we also have $\{l_{i,1}, l_{i,0}\}_{i=1}^n \in \tilde{\mathcal{R}}_\epsilon^{W,\kappa}$. Since the upper bound above (40) holds for both $I \in \{0,1\}$, the equivalent size of $\tilde{\mathcal{R}}_\epsilon^{W,\kappa}$ can be divided by 2. Taking the union bound over every $r \in \tilde{\mathcal{R}}_\epsilon^{W,\kappa}$, we then have that for any $\delta > 0$, with probability at least $1-\delta$, the following bound holds for all $\tilde{R}^{W,\kappa} \in \tilde{\mathcal{R}}_\epsilon^{W,\kappa}$ simultaneously if $\tilde{R}^{W,\kappa} = r$:

$$\frac{1}{n}\sum_{i=1}^n L_{i,1-\tilde{U}_i}^{W,\kappa} - (1+C_i)L_{i,\tilde{U}_i}^{W,\kappa} \le \frac{\log\left(2\left|\tilde{\mathcal{R}}_\epsilon^{W,\kappa}\right|/\delta\right)}{n\eta}. \tag{44}$$

By substituting (43) into (44) and take the union bound with (42), we have that for any $\tilde{R}^{W,\kappa} \in \tilde{\mathcal{R}}^{W,\kappa}$ and $\delta > 0$, with probability at least $1-\delta$,

$$\frac{1}{n}\sum_{i=1}^n L_{i,1-\tilde{U}_i}^{W,\kappa} - (1+C_i)L_{i,\tilde{U}_i}^{W,\kappa} \le \frac{\log\left(\left|\tilde{\mathcal{R}}_\epsilon^{W,\kappa}\right|\right) + \log(4/\delta)}{n\eta}$$

$$\le \frac{H_{1-\lambda}(\tilde{R}^{W,\kappa}) + \frac{1}{\lambda}\log(1/\delta) + \log(4/\delta)}{n\eta}.$$

The only requirement on the selection of $C_i$ and $\eta$ is

$$C_i, \eta \in \left\{C_i \ge -\frac{\log\left(2 - e^{2\eta\hat{L}_i^{W,\kappa}}\right)}{2\eta\hat{L}_i^{W,\kappa}} - 1\right\}, \quad \text{for all } i \in [1,n].$$

Note that the condition above implies $e^{2\eta\hat{L}_i^{W,\kappa}} < 2$ for any $i \in [1,n]$, we therefore choose

$$\eta < \frac{\log 2}{2\kappa} \le \frac{\log 2}{2\sup_{i\in[1,n]}\left(\hat{L}_i^{W,\kappa}\right)}.$$

$\square$

**Theorem 5** (Restate). *For any $\kappa \ge 0$, $\lambda, \gamma \in (0,1)$ and $\delta > 0$, if $\kappa \ge B^{W,\tilde{S}_s}$, then with probability at least $1-\delta$, the following inequality holds:*

$$\Delta\left(W, \tilde{S}_s, \tilde{U}\right) \le \frac{1}{n}\sum_{i=1}^n C_i L_{i,\tilde{U}_i}^W + G_1^W \frac{H_{1-\lambda}(\tilde{R}^W) + G_2^W}{n},$$

*where*

$$G_1^W = \frac{1}{\eta} = \frac{2\kappa}{\gamma \log 2},$$

$$G_2^W = \frac{1}{\lambda}\log\left(\frac{1}{\delta}\right) + \log\left(\frac{4}{\delta}\right),$$

$$C_i = -\frac{\log\left(2 - e^{2\eta\hat{L}_i^W}\right)}{2\eta\hat{L}_i^W} - 1.$$

*Proof.* The proof directly follows from Lemma 12 by noticing that $L_{i,I}^{W,\kappa} = L_{i,I}^W$ for any $i \in [1,n]$ and $I \in \{0,1\}$ when $\kappa \ge B^{W,\tilde{S}_s}$. $\square$

**Theorem 6** (Restate). *For any $\kappa > 0$, $\gamma, \lambda_1, \lambda_2 \in (0,1)$ and $\delta > 0$, with probability at least $1-\delta$, the following inequality holds:*

$$\Delta\left(W, \tilde{S}_s, \tilde{U}\right) \le \frac{1}{n}\sum_{i=1}^n C_i L_{i,\tilde{U}_i}^{W,\kappa} + \tilde{G}_1^W \frac{H_{1-\lambda_1}(\tilde{R}^{W,\kappa}) + \tilde{G}_2^W}{n} + \tilde{G}_3^W\sqrt{\frac{H_{1-\lambda_2}(\tilde{R}_\Delta^{W,-\kappa}) + \tilde{G}_4^W}{n}},$$

where $\tilde{R}^{W,\kappa} = \{L_{i,0}^{W,\kappa}, L_{i,1}^{W,\kappa}\}_{i=1}^{n}$, $\tilde{R}_{\Delta}^{W,-\kappa} = \{\Delta L_{i}^{W,-\kappa}\}_{i=1}^{n}$, $\Delta L_{i}^{W,-\kappa} = L_{i,1}^{W,-\kappa} - L_{i,0}^{W,-\kappa}$ and

$$\tilde{G}_1^W = \frac{1}{\eta} = \frac{2\kappa}{\gamma \log 2},$$

$$\tilde{G}_2^W = \frac{1}{\lambda_1} \log(2/\delta) + \log(8/\delta),$$

$$\tilde{G}_3^W = \sqrt{\frac{2}{n} \sum_{i=1}^{n} \left(\Delta L_i^{W,-\kappa}\right)^2},$$

$$\tilde{G}_4^W = \frac{1}{\lambda_2} \log(2/\delta) + \log(4/\delta),$$

$$C_i = -\frac{\log\left(2 - e^{2\eta \hat{L}_i^{W,\kappa}}\right)}{2\eta \hat{L}_i^{W,\kappa}} - 1.$$

*Proof.* By the definition of the validation error, we have the following decomposition:

$$\Delta\left(W, \tilde{S}_s, \tilde{U}\right) = \frac{1}{n} \sum_{i=1}^{n} L_{i,1-\tilde{U}_i}^{W} - L_{i,\tilde{U}_i}^{W}$$

$$= \frac{1}{n} \sum_{i=1}^{n} L_{i,1-\tilde{U}_i}^{W,\kappa} - (1+C_i)L_{i,\tilde{U}_i}^{W,\kappa} + \frac{1}{n} \sum_{i=1}^{n} C_i L_{i,\tilde{U}_i}^{W,\kappa}$$

$$+ \frac{1}{n} \sum_{i=1}^{n} L_{i,1-\tilde{U}_i}^{W,-\kappa} - L_{i,\tilde{U}_i}^{W,-\kappa}. \tag{45}$$

By applying Lemma 12, we have that for any $\delta > 0$, with probability at least $1 - \delta$,

$$\frac{1}{n} \sum_{i=1}^{n} L_{i,1-\tilde{U}_i}^{W,\kappa} - (1+C_i)L_{i,\tilde{U}_i}^{W,\kappa} \le \frac{H_{1-\lambda_1}(\tilde{R}^{W,\kappa}) + \frac{1}{\lambda_1} \log(1/\delta) + \log(4/\delta)}{n\eta},$$

By taking $\eta = \frac{\gamma \log 2}{2\kappa} < \frac{\log 2}{2\kappa}$, we then have

$$\frac{1}{n} \sum_{i=1}^{n} L_{i,1-\tilde{U}_i}^{W,\kappa} - (1+C_i)L_{i,\tilde{U}_i}^{W,\kappa} \le \frac{2\kappa\left(H_{1-\lambda_1}(\tilde{R}^{W,\kappa}) + \frac{1}{\lambda_1} \log(1/\delta) + \log(4/\delta)\right)}{n\gamma \log 2}. \tag{46}$$

By reapplying the proof of Theorem 4 on $\tilde{R}^{W,-\kappa}$ instead of $\tilde{R}^{W}$, we have that for any $\delta > 0$, with probability at least $1 - \delta$,

$$\frac{1}{n} \sum_{i=1}^{n} L_{i,1-\tilde{U}_i}^{W,-\kappa} - L_{i,\tilde{U}_i}^{W,-\kappa} \le \sqrt{\frac{2}{n} \sum_{i=1}^{n} \left(\Delta L_i^{W,-\kappa}\right)^2} \sqrt{\frac{H_{1-\lambda_2}(\tilde{R}_{\Delta,d}^{W,-\kappa}) + \frac{1}{\lambda_2} \log(1/\delta) + \log(2/\delta)}{n}}. \tag{47}$$

By taking the union bound of (46), (47) and then substituting it into (45), we have

$$\Delta\left(W, \tilde{S}_s, \tilde{U}\right) \le \frac{1}{n} \sum_{i=1}^{n} C_i L_{i,\tilde{U}_i}^{W,\kappa} + \frac{2\kappa\left(H_{1-\lambda_1}(\tilde{R}^{W,\kappa}) + \frac{1}{\lambda_1} \log(2/\delta) + \log(8/\delta)\right)}{n\gamma \log 2}$$

$$+ \sqrt{\frac{2}{n} \sum_{i=1}^{n} \left(\Delta L_i^{W,-\kappa}\right)^2} \sqrt{\frac{H_{1-\lambda_2}(\tilde{R}_{\Delta}^{W,-\kappa}) + \frac{1}{\lambda_2} \log(2/\delta) + \log(4/\delta)}{n}}.$$

$\square$

# E ADDITIONAL THEORETICAL RESULTS

## E.1 SUBADDITIVITY OF RÉNYI'S ENTROPY

It is easy to show that Shannon's entropy enjoys the subadditivity property, i.e. given arbitrary random variables $X_1, \cdots, X_n$, we have

$$H(X_1, \cdots, X_n) = H(X_1) + H(X_2|X_1) + \cdots + H(X_n|X_1, \cdots, X_{n-1})$$
$$\leq H(X_1) + H(X_2) + \cdots + H(X_n).$$

If the subadditivity property also applies to Rényi's entropy, we will be able to upper bound the joint entropy of samplewise losses in our main theorems by the sum of entropies of each individual loss, which avoids estimating high-dimensional information-theoretic quantities and yields directly tractable upper bounds. We will show that for discrete random variables with finite support, the subadditivity property holds for Rényi's entropy when $\alpha \approx 1$. Note that these assumptions are naturally satisfied for digital computers using floating-point numbers.

**Lemma 13.** *Given arbitrary discrete random variables $X \in \mathcal{X}$ and $Y \in \mathcal{Y}$, assume that $|\mathcal{X}| = m < \infty$ and $|\mathcal{Y}| = n < \infty$. If $X$ and $Y$ are independent, then $H_\alpha(X, Y) = H_\alpha(X) + H_\alpha(Y)$. Otherwise for any $\epsilon \in (0, H(X) + H(Y) - H(X, Y))$, there exists $\delta > 0$ such that when $\alpha \in (1 - \delta, 1)$, we have $H_\alpha(X, Y) + \epsilon \leq H_\alpha(X) + H_\alpha(Y)$.*

*Proof.* Let $p_{ij} = \mathbb{P}(X = \mathcal{X}_i, Y = \mathcal{Y}_j)$, $p_i = \mathbb{P}(X = \mathcal{X}_i) = \sum_{j=1}^n p_{ij}$ and $q_j = \mathbb{P}(Y = \mathcal{Y}_j) = \sum_{i=1}^m p_{ij}$, then by the definition of Rényi's entropy, we have

$$H_\alpha(X, Y) = \frac{1}{1-\alpha} \log \sum_{i=1}^m \sum_{j=1}^n p_{ij}^\alpha,$$

$$H_\alpha(X) = \frac{1}{1-\alpha} \log \sum_{i=1}^m p_i^\alpha, \qquad H_\alpha(Y) = \frac{1}{1-\alpha} \log \sum_{j=1}^n q_j^\alpha.$$

When $X$ and $Y$ are independent, we have $p_{ij} = p_i q_j$ and thus

$$\sum_{i=1}^m \sum_{j=1}^n p_{ij}^\alpha = \left(\sum_{i=1}^m p_i^\alpha\right) \cdot \left(\sum_{j=1}^n q_j^\alpha\right),$$

which directly implies $H_\alpha(X, Y) = H_\alpha(X) + H_\alpha(Y)$.

Otherwise when $X$ and $Y$ are dependent, recall that $H(X, Y) < H(X) + H(Y)$, we then have

$$\lim_{\alpha \to 1} H_\alpha(X, Y) \leq \lim_{\alpha \to 1} H_\alpha(X) + H_\alpha(Y).$$

For any $\epsilon \in (0, H(X) + H(Y) - H(X, Y))$, let $\gamma = (H(X) + H(Y) - H(X, Y) - \epsilon)/2$. Then by the definition of limit, we know that there exist $\delta_1, \delta_2 > 0$ such that when $\alpha \in (1 - \delta_1, 1)$, we have $|H_\alpha(X, Y) - H(X, Y)| \leq \gamma$, when $\alpha \in (1 - \delta_2, 1)$, we have $|H_\alpha(X) + H_\alpha(Y) - H(X) - H(Y)| \leq \gamma$. Combining these two inequalities yields the desired result:

$$H_\alpha(X, Y) \leq \gamma + H(X, Y) = \gamma + H(X) + H(Y) - \epsilon - 2\gamma$$
$$\leq \gamma + \gamma + H_\alpha(X) + H_\alpha(Y) - \epsilon - 2\gamma = H_\alpha(X) + H_\alpha(Y) - \epsilon.$$

The proof is complete by taking $\delta = \min(\delta_1, \delta_2)$. □

When the number of samples $n$ is large enough, we have $\lim_{n \to \infty} \frac{1}{n} \log(1/\delta) = 0$ in our main theorems. These high-probability factors are thus significantly smaller than the joint loss entropy term $H_{1-\lambda}(R^W)$, and we can select $\lambda \approx 0$ to acquire computational tractable bounds as

$$H_{1-\lambda}(R^W) \leq H_{1-\lambda}(L_1) + H_{1-\lambda}(L_2) + \cdots + H_{1-\lambda}(L_n),$$

by applying Lemma 13 recursively.

### E.2 Tightening Previous Bounds by Thresholding

In this section, we demonstrate that the fast-rate bounds for the expected generalization error established in (Wang & Mao, 2023) can be further tightened by adopting our thresholding strategy. The original theorems are stated as follows:

**Theorem 7.** *(Theorem 3.2 in Wang & Mao, 2023) Assume that $\ell(\cdot, \cdot) \in [0, 1]$, then*

$$\left| \mathbb{E}_{W, \tilde{S}_s, \tilde{U}} \left[ \Delta \left( W, \tilde{S}_s, \tilde{U} \right) \right] \right| \leq \frac{1}{n} \sum_{i=1}^{n} \sqrt{2 I(\Delta L_i^W; \tilde{U}_i)}.$$

**Theorem 8.** *(Theorem 4.3 in Wang & Mao, 2023) Assume that $\ell(\cdot, \cdot) \in [0, 1]$, then for any $C_2 \in (0, \log 2/2)$,*

$$\mathbb{E}_{W, \tilde{S}_s, \tilde{U}} \left[ \Delta \left( W, \tilde{S}_s, \tilde{U} \right) \right] \leq C_1 \mathbb{E}_{W, S_s} [L_{S_s}(W)] + \sum_{i=1}^{n} \frac{I(L_{i,1}^W; \tilde{U}_i)}{C_2 n},$$

*where*

$$C_1 = -\frac{\log\left(2 - e^{2C_2}\right)}{2C_2} - 1.$$

As can be seen, these upper bounds only hold when $\ell(\cdot, \cdot) \in [0, 1]$. In the case of unbounded loss functions e.g. cross-entropy, these bounds should be multiplied by $b^{W, \tilde{S}_s}$ to stay hold, which is significantly looser when the marginal distributions of the losses are long-tailed. We first tighten Theorem 7 by introducing subgaussianity:

**Lemma 14.** *Assume that $L_{i,0}^W$ and $L_{i,1}^W$ are $\sigma$-subgaussian for any $i \in [1, n]$, then*

$$\left| \mathbb{E}_{W, \tilde{S}_s, \tilde{U}} \left[ \Delta \left( W, \tilde{S}_s, \tilde{U} \right) \right] \right| \leq \frac{2}{n} \sum_{i=1}^{n} \sqrt{2 \sigma^2 I(\Delta L_i^W; \tilde{U}_i)}.$$

*Proof.* From the definition of the validation error, we have

$$\left| \mathbb{E}_{W, \tilde{S}_s, \tilde{U}} \left[ \Delta \left( W, \tilde{S}_s, \tilde{U} \right) \right] \right| = \left| \mathbb{E}_{\tilde{S}_s, \tilde{U}, W} \left[ \frac{1}{n} \sum_{i=1}^{n} L_{i, 1-\tilde{U}_i}^W - L_{i, \tilde{U}_i}^W \right] \right|$$

$$= \frac{1}{n} \left| \mathbb{E}_{\tilde{S}_s, \tilde{U}, W} \left[ \sum_{i=1}^{n} (-1)^{\tilde{U}_i} \left( L_{i,1}^W - L_{i,0}^W \right) \right] \right|$$

$$\leq \frac{1}{n} \sum_{i=1}^{n} \left| \mathbb{E}_{\Delta L_i^W, \tilde{U}_i} \left[ (-1)^{\tilde{U}_i} \Delta L_i^W \right] \right|.$$

From the assumption that $L_{i,0}^W$ and $L_{i,1}^W$ are $\sigma$-subgaussian, we have that $\Delta L_i^W = L_{i,1}^W - L_{i,0}^W$ is $2\sigma$-subgaussian. Then by applying Lemma 2 with $f(\tilde{U}_i, \Delta L_i^W) = (-1)^{\tilde{U}_i} \Delta L_i^W$, we have

$$\left| \mathbb{E}_{\Delta L_i^W, \tilde{U}_i} \left[ (-1)^{\tilde{U}_i} \Delta L_i^W \right] - \mathbb{E}_{\Delta \bar{L}_i^W, \tilde{U}_i} \left[ (-1)^{\tilde{U}_i} \Delta \bar{L}_i^W \right] \right| \leq 2 \sqrt{2 \sigma^2 I(\Delta L_i^W; \tilde{U}_i)}.$$

Since $\Delta \bar{L}_i^W$ is independent of $\tilde{U}_i$, we have $\mathbb{E}_{\Delta \bar{L}_i^W, \tilde{U}_i} \left[ (-1)^{\tilde{U}_i} \Delta \bar{L}_i^W \right] = 0$ and

$$\left| \mathbb{E}_{W, \tilde{S}_s, \tilde{U}} \left[ \Delta \left( W, \tilde{S}_s, \tilde{U} \right) \right] \right| \leq \frac{1}{n} \sum_{i=1}^{n} \left| \mathbb{E}_{\Delta L_i^W, \tilde{U}_i} \left[ (-1)^{\tilde{U}_i} \Delta L_i^W \right] \right|$$

$$\leq \frac{2}{n} \sum_{i=1}^{n} \sqrt{2 \sigma^2 I(\Delta L_i^W; \tilde{U}_i)}.$$

$\square$

Note that this bound is at least as tight as Theorem 7, since the condition that $\ell(\cdot, \cdot) \in [0, 1]$ implies $\sigma \leq \frac{1}{2}$.

We further tighten Theorem 8 by introducing the thresholding strategy. For any $\kappa > 0$, the expected generalization error can be decomposed as:

$$
\begin{aligned}
&\mathbb{E}_{W, \tilde{S}_s, \tilde{U}}\left[\Delta\left(W, \tilde{S}_s, \tilde{U}\right)\right] \\
&= \mathbb{E}_{\tilde{S}_s, \tilde{U}, W}\left[\frac{1}{n} \sum_{i=1}^{n} L_{i,1-\tilde{U}_i}^{W} - L_{i,\tilde{U}_i}^{W}\right] \\
&= \mathbb{E}_{\tilde{S}_s, \tilde{U}, W}\left[\frac{1}{n} \sum_{i=1}^{n} L_{i,1-\tilde{U}_i}^{W,\kappa} - (1 + C_1) L_{i,\tilde{U}_i}^{W,\kappa} + L_{i,1-\tilde{U}_i}^{W,-\kappa} - L_{i,\tilde{U}_i}^{W,-\kappa} + C_1 L_{i,\tilde{U}_i}^{W,\kappa}\right] \\
&\leq \mathbb{E}_{\tilde{S}_s, \tilde{U}, W}\left[\frac{1}{n} \sum_{i=1}^{n} L_{i,1-\tilde{U}_i}^{W,\kappa} - (1 + C_1) L_{i,\tilde{U}_i}^{W,\kappa}\right] + \mathbb{E}_{\tilde{S}_s, \tilde{U}, W}\left[\frac{1}{n} \sum_{i=1}^{n} L_{i,1-\tilde{U}_i}^{W,-\kappa} - L_{i,\tilde{U}_i}^{W,-\kappa}\right] \\
&\quad + \mathbb{E}_{\tilde{S}_s, \tilde{U}, W}\left[\frac{1}{n} \sum_{i=1}^{n} C_1 L_{i,\tilde{U}_i}^{W,\kappa}\right]. \tag{48}
\end{aligned}
$$

The following theorem establishes an upper bound for the decomposition above by leveraging both square-root and fast-rate bounds:

**Theorem 9.** *For any $\kappa > 0$ and $C_2 \in (0, \frac{\log 2}{2\kappa})$, assume that $L_{i,0}^{W,\kappa}$ and $L_{i,1}^{W,\kappa}$ are $\sigma_\kappa$-subgaussian for all $i \in [1, n]$, then*

$$
\mathbb{E}_{W, \tilde{S}_s, \tilde{U}}\left[\Delta\left(W, \tilde{S}_s, \tilde{U}\right)\right] \leq \sum_{i=1}^{n} \frac{I(L_{i,1}^{W,\kappa}; \tilde{U}_i)}{C_2 n} + \frac{2}{n} \sum_{i=1}^{n} \sqrt{2 \sigma_\kappa^2 I(\Delta L_i^{W,-\kappa}; \tilde{U}_i)}
$$
$$
+ \frac{C_1}{n} \mathbb{E}_{W, S_s}\left[\sum_{i=1}^{n} L_{i,\tilde{U}_i}^{W,\kappa}\right],
$$

*where*

$$
C_1 = -\frac{\log\left(2 - e^{2 C_2 \kappa}\right)}{2 C_2 \kappa} - 1.
$$

*Proof.* We first decompose the first term in (48) as:

$$
\begin{aligned}
&\mathbb{E}_{W, \tilde{S}_s, \tilde{U}}\left[\frac{1}{n} \sum_{i=1}^{n} L_{i,1-\tilde{U}_i}^{W,\kappa} - (1 + C_1) L_{i,\tilde{U}_i}^{W,\kappa}\right] \\
&= \frac{1}{n} \sum_{i=1}^{n} \mathbb{E}_{L_{i,0}^{W}, L_{i,1}^{W}, \tilde{U}_i}\left[L_{i,1-\tilde{U}_i}^{W,\kappa} - (1 + C_1) L_{i,\tilde{U}_i}^{W,\kappa}\right] \\
&= \frac{1}{n} \sum_{i=1}^{n} \mathbb{E}_{L_{i,0}^{W}, L_{i,1}^{W}, \tilde{U}_i}\left[\left(1 + \frac{C_1}{2}\right)\left(L_{i,1-\tilde{U}_i}^{W,\kappa} - (1 + C_1) L_{i,\tilde{U}_i}^{W,\kappa}\right) - \frac{C_1}{2} L_{i,1-\tilde{U}_i}^{W,\kappa} - \frac{C_1}{2} L_{i,\tilde{U}_i}^{W,\kappa}\right] \\
&= \frac{1}{2n} \sum_{i=1}^{n}\left[\mathbb{E}_{L_{i,0}^{W}, \tilde{U}_i}\left[-(2 + C_1)(-1)^{\tilde{U}_i} L_{i,0}^{W,\kappa} - C_1 L_{i,0}^{W,\kappa}\right]\right. \\
&\quad \left. + \mathbb{E}_{L_{i,1}^{W}, \tilde{U}_i}\left[(2 + C_1)(-1)^{\tilde{U}_i} L_{i,1}^{W,\kappa} - C_1 L_{i,1}^{W,\kappa}\right]\right].
\end{aligned}
$$

Notice that $L_{i,0}^{W}$ and $L_{i,1}^{W}$ share the same marginal distribution, $\mathbb{E}[L_{i,0}^{W,\kappa}|\tilde{U}_i = 0] = \mathbb{E}[L_{i,1}^{W,\kappa}|\tilde{U}_i = 1]$, and $\mathbb{E}[L_{i,1}^{W,\kappa}|\tilde{U}_i = 0] = \mathbb{E}[L_{i,0}^{W,\kappa}|\tilde{U}_i = 1]$, we have

$$
\mathbb{E}_{W, \tilde{S}_s, \tilde{U}}\left[\frac{1}{n} \sum_{i=1}^{n} L_{i,1-\tilde{U}_i}^{W,\kappa} - (1 + C_1) L_{i,\tilde{U}_i}^{W,\kappa}\right] = \frac{1}{n} \sum_{i=1}^{n} \mathbb{E}_{L_{i,1}^{W}, \tilde{U}_i}\left[(2 + C_1)(-1)^{\tilde{U}_i} L_{i,1}^{W,\kappa} - C_1 L_{i,1}^{W,\kappa}\right].
$$

For any $i \in [1, n]$, by applying Lemma 3 on $C_2\left((2 + C_1)(-1)^{\tilde{U}_i} L_{i,1}^{W,\kappa} - C_1 L_{i,1}^{W,\kappa}\right)$, we have

$$
I(L_{i,1}^{W,\kappa}; \tilde{U}_i) \geq \mathbb{E}_{L_{i,1}^{W}, \tilde{U}_i}\left[C_2\left((2 + C_1)(-1)^{\tilde{U}_i} L_{i,1}^{W,\kappa} - C_1 L_{i,1}^{W,\kappa}\right)\right]
$$
$$
- \log \mathbb{E}_{\bar{L}_{i,1}^{W}, \tilde{U}_i}\left[e^{C_2\left((2+C_1)(-1)^{\tilde{U}_i} \bar{L}_{i,1}^{W,\kappa} - C_1 \bar{L}_{i,1}^{W,\kappa}\right)}\right]. \tag{49}
$$

Since $\bar{L}_{i,1}^{W,\kappa} \leq \kappa$, by selecting $C_1$ and $C_2$ as:

$$
C_1, C_2 \in \left\{C_2 \in \left(0, \frac{\log 2}{2\kappa}\right), C_1 \geq -\frac{\log\left(2 - e^{2C_2\kappa}\right)}{2C_2\kappa} - 1\right\},
$$

we will have

$$
\mathbb{E}_{\bar{L}_{i,1}^{W}, \tilde{U}_i}\left[e^{C_2\left((2+C_1)(-1)^{\tilde{U}_i} \bar{L}_{i,1}^{W,\kappa} - C_1 \bar{L}_{i,1}^{W,\kappa}\right)}\right] = \frac{\mathbb{E}_{\bar{L}_{i,1}^{W}}\left[e^{-2C_2(1+C_1)\bar{L}_{i,1}^{W,\kappa}} + e^{2C_2 \bar{L}_{i,1}^{W,\kappa}}\right]}{2} \leq 1. \tag{50}
$$

By substituting (50) into (49), we can get

$$
\mathbb{E}_{L_{i,1}^{W}, \tilde{U}_i}\left[C_2\left((2 + C_1)(-1)^{\tilde{U}_i} L_{i,1}^{W,\kappa} - C_1 L_{i,1}^{W,\kappa}\right)\right] \leq I(L_{i,1}^{W,\kappa}; \tilde{U}_i),
$$

By summing the inequality above over $i \in [1, n]$, we can prove that

$$
\mathbb{E}_{W, \tilde{S}_s, \tilde{U}}\left[\frac{1}{n}\sum_{i=1}^{n} L_{i,1-\tilde{U}_i}^{W,\kappa} - (1 + C_1) L_{i,\tilde{U}_i}^{W,\kappa}\right] \leq \sum_{i=1}^{n} \frac{I(L_{i,1}^{W,\kappa}; \tilde{U}_i)}{C_2 n}. \tag{51}
$$

Next, by reapplying the proof of Lemma 14 on $\tilde{R}^{W,-\kappa}$ instead of $\tilde{R}^W$, we can prove that

$$
\mathbb{E}_{W, \tilde{S}_s, \tilde{U}}\left[\frac{1}{n}\sum_{i=1}^{n} L_{i,1-\tilde{U}_i}^{W,-\kappa} - L_{i,\tilde{U}_i}^{W,-\kappa}\right] \leq \frac{2}{n}\sum_{i=1}^{n} \sqrt{2\sigma_\kappa^2 I(\Delta L_i^{W,-\kappa}; \tilde{U}_i)}, \tag{52}
$$

The proof is complete by substituting (51) and (52) into (48). $\qquad\square$

According to the Markov chain relationship $\tilde{U}_i \to L_{i,1}^{W} \to L_{i,1}^{W,\kappa}$ and by applying the data-processing inequality, we have $I(L_{i,1}^{W,\kappa}; \tilde{U}_i) \leq I(L_{i,1}^{W}; \tilde{U}_i)$. We also have $L_{i,\tilde{U}_i}^{W,\kappa} \leq L_{i,\tilde{U}_i}^{W}$. Therefore, Theorem 9 tightens Theorem 8 by reducing the $b^{W,\tilde{S}_s}$ factor to a manual threshold $\kappa$, at the cost of introducing an extra term that scales with $\sigma_\kappa$. When the losses are long-tailed, we will have $b^{W,\tilde{S}_s} \gg \sigma_\kappa$ and Theorem 9 can be significantly tighter than Theorem 8. Otherwise, we can simply take $\kappa \approx b^{W,\tilde{S}_s}$ and let Theorem 9 reduce to Theorem 8.

### E.3 COMPARISON WITH VARIANCE AND SHARPNESS BASED BOUNDS

Theorem 6 also provides alternative perspectives on fast-rate generalization bounds by leveraging the entropies of training and test losses. By comparison, the previous work (Wang & Mao, 2023) utilizes variance $V(\gamma)$ and sharpness $F(\gamma)$ related terms as replacements to the training risk, which are defined as:

$$
V(\gamma) = \mathbb{E}_{W, S}\left[\frac{1}{n}\sum_{i=1}^{n}(\ell(W, Z_i) - (1 + \gamma)L_S(W))^2\right],
$$

$$
F(\gamma) = \frac{1}{n}\sum_{i=1}^{n} \mathbb{E}_{W, Z_i}\left[\ell(W, Z_i) - (1 + \gamma)\mathbb{E}_{W|Z_i}[\ell(W, Z_i)]\right]^2.
$$

We show that our loss entropy measure serves as tightened versions of both the variance and sharpness-related terms. To see this, let $I \sim \text{Unif}([1, n])$ randomly selects a training sample $Z_I$ from $S$. We now measure the variance of $L_I = \ell(W, Z_I)$:

**Proposition 1.** *Assume that $\ell(\cdot, \cdot) \in [0, 1]$, then* $\text{Var}[L_I] \leq \min(V(0), F(0)) + \mathbb{E}_{W, S}[L_S(W)]$.

*Proof.* For the formula of the total variance, we have

$$\text{Var}_{W,S,I}[L_I] = \mathbb{E}_{W,S}[\text{Var}_I[L_I]] + \text{Var}_{W,S}[\mathbb{E}_I[L_I]]$$

$$= \mathbb{E}_{W,S}\left[\frac{1}{n}\sum_{i=1}^n (\ell(W, Z_i) - L_S(W))^2\right] + \text{Var}_{W,S}[L_S(W)]$$

$$= V(0) + \mathbb{E}_{W,S}[L_S^2(W)] - \mathbb{E}_{W,S}^2[L_S(W)]$$

$$\leq V(0) + \mathbb{E}_{W,S}[L_S(W)].$$

Similarly, we have

$$\text{Var}_{W,S,I}[L_I] = \mathbb{E}_S[\text{Var}_{W,I|S}[L_I]] + \text{Var}_S[\mathbb{E}_{W,I|S}[L_I]]$$

$$= \mathbb{E}_S\left[\mathbb{E}_{W,I|S}\big[\ell(W, Z_I) - \mathbb{E}_{W|S,I}[\ell(W, Z_I)]\big]^2\right] + \text{Var}_S[\mathbb{E}_{W,I|S}[L_I]]$$

$$\leq \mathbb{E}_S\left[\frac{1}{n}\sum_{i=1}^n \mathbb{E}_{W|Z_i}\big[\ell(W, Z_i) - \mathbb{E}_{W|Z_i}[\ell(W, Z_i)]\big]^2\right] + \text{Var}_S[\mathbb{E}_{W,I|S}[L_I]]$$

$$= \frac{1}{n}\sum_{i=1}^n \mathbb{E}_{W,Z_i}\big[\ell(W, Z_i) - \mathbb{E}_{W|Z_i}[\ell(W, Z_i)]\big]^2 + \text{Var}_S[\mathbb{E}_{W,I|S}[L_I]]$$

$$= F(0) + \mathbb{E}_S[\mathbb{E}_{W,I|S}^2[L_I]] - \mathbb{E}_S^2[\mathbb{E}_{W,I|S}[L_I]]$$

$$\leq F(0) + \mathbb{E}_S[\mathbb{E}_{W,I|S}[L_I]]$$

$$= F(0) + \mathbb{E}_{W,S}[L_S(W)].$$

$\square$

Here, $\mathbb{E}_{W,S}[L_S(W)]$ is the average training loss, which approaches zero in the interpolating regime. That is, the variance of $L_I$ is a natural lower bound for both the variance and sharpness measures. Furthermore, we show that the entropy of training losses is upper bounded by that of $L_I$:

$$\frac{1}{n}H(\tilde{R}_S^W) \leq \frac{1}{n}\sum_{i=1}^n H(L_{i,\tilde{U}_i}^W) = \sum_{i=1}^n \mathbb{P}(I = i)H(L_I|I = i) = H(L_I|I) \leq H(L_I).$$

Notice that $L_I$ is a Bernoulli random variable under the assumption that $\ell(\cdot, \cdot) \in \{0, 1\}$, which implies that the entropy of $L_I$ is determinated when we know its variance. Therefore, any upper bound for $\text{Var}[L_I]$ directly serves as an upper bound for $H(L_I)$. This conclusion can be extended to continuous cases, by noticing that the Gaussian distribution $N(0, \sigma^2)$ possesses the maximum entropy among all probability distributions whose variance is $\sigma^2$. On the contrary, the variance is unknown and can even be non-finite when only the entropy is provided (e.g. Cauchy distribution, student-$t$ distribution with $t < 2$, etc.). Moreover, the fast-rate bounds of (Wang & Mao, 2023) requires $\ell(\cdot, \cdot)$ to be 0-1 loss, while Theorem 6 can accommodate any $\ell(\cdot, \cdot) \in [0, 1]$. These observations further validate the superiority of our results.

## F  EXPERIMENT DETAILS AND ADDITIONAL RESULTS

In this paper, deep learning models are trained with an Intel Xeon CPU (2.10GHz, 48 cores), 256GB memory, and 4 Nvidia Tesla V100 GPUs (32GB).

### F.1  SYNTHETIC EXPERIMENTS

In our synthetic data experiments, the learning task is a 5-class classification on two-dimensional points generated with isotropic Gaussian distributions. The training set contains 50 samples, while the test set contains 250 samples. 216 models in total are trained according to different combinations of the following options: 4 MLP encoders ([256, 256, 128, 128], [128, 128, 64, 64], [64, 64, 32, 32], [32, 32, 16, 16]), 3 weight-decay rates (0, 0.01, 0.1), 3 dataset draws and 3 random seeds. The models are designed under the variational setting, where the encoder is trained to characterize a conditional distribution for the representation given the input via deterministic means and standard deviations. The parameterization trick is used for optimization. Models are trained for 300

| Metric | Spearman | Pearson | Kendall |
|---|---|---|---|
| Num. params. | -0.0576 | -0.0294 | -0.0402 |
| $\|W\|_F$ | -0.2172 | -0.0871 | -0.1374 |
| $\hat{I}(X;T^w)$ | 0.1816 | 0.2878 | 0.1280 |
| $\hat{I}(X;T^w|Y)$ | 0.1749 | 0.3167 | 0.1129 |
| $\check{I}(X;T^w)$ | 0.1648 | 0.3712 | 0.1223 |
| $\check{I}(X;T^w|Y)$ | 0.2293 | 0.3842 | 0.1515 |
| $\check{I}(S;W)$ | 0.0020 | 0.0211 | 0.0074 |
| $\check{I}(S;W) + \hat{I}(X;T^w)$ | 0.0178 | 0.0211 | 0.0178 |
| $\check{I}(S;W) + \hat{I}(X;T^w|Y)$ | 0.0163 | 0.0211 | 0.0167 |
| $\check{I}(S;W) + \check{I}(X;T^w)$ | 0.0135 | 0.0212 | 0.0162 |
| $\check{I}(S;W) + \check{I}(X;T^w|Y)$ | 0.0164 | 0.0211 | 0.0167 |
| $\tilde{I}(S;W) + \hat{I}(X;T^w)$ | 0.1104 | 0.1401 | 0.0794 |
| $\tilde{I}(S;W) + \hat{I}(X;T^w|Y)$ | 0.2253 | 0.3177 | 0.1567 |
| $\tilde{I}(S;W) + \check{I}(X;T^w)$ | 0.2684 | 0.3928 | 0.1912 |
| $\tilde{I}(S;W) + \check{I}(X;T^w|Y)$ | 0.3015 | 0.4130 | 0.2085 |
| $H(L^w)$ | 0.5767 | 0.5611 | 0.4037 |
| $H(L^w|Y)$ | **0.7088** | **0.6350** | **0.5251** |

Table 1: Correlation analysis between different metrics and the generalization gap. $\hat{I}$, $\check{I}$, $\tilde{I}$ represents different strategies for mutual information estimation.

| Metric | Spearman | Pearson | Kendall |
|---|---|---|---|
| Num. params. | 0.4944 | 0.2770 | 0.3985 |
| $\|W\|_F$ | 0.4944 | 0.2680 | 0.3985 |
| $\hat{I}(X;T^w)$ | 0.4799 | 0.6232 | 0.2941 |
| $\hat{I}(X;T^w|Y)$ | 0.4923 | 0.6185 | 0.3203 |
| $\check{I}(X;T^w)$ | 0.1496 | 0.0190 | 0.0196 |
| $\check{I}(X;T^w|Y)$ | 0.2198 | 0.0495 | 0.0980 |
| $\check{I}(S;W)$ | 0.6065 | 0.5692 | 0.4633 |
| $\check{I}(S;W) + \hat{I}(X;T^w)$ | 0.6140 | 0.6417 | 0.4248 |
| $\check{I}(S;W) + \hat{I}(X;T^w|Y)$ | 0.6202 | 0.6406 | 0.4510 |
| $\check{I}(S;W) + \check{I}(X;T^w)$ | 0.3808 | 0.1378 | 0.2549 |
| $\check{I}(S;W) + \check{I}(X;T^w|Y)$ | 0.4138 | 0.1648 | 0.2810 |
| $\tilde{I}(S;W) + \hat{I}(X;T^w)$ | 0.6223 | 0.5692 | 0.4510 |
| $\tilde{I}(S;W) + \hat{I}(X;T^w|Y)$ | 0.6223 | 0.5692 | 0.4510 |
| $\tilde{I}(S;W) + \check{I}(X;T^w)$ | 0.5666 | 0.5685 | 0.3987 |
| $\tilde{I}(S;W) + \check{I}(X;T^w|Y)$ | 0.5810 | 0.5707 | 0.4118 |
| $H(L^w)$ | 0.8782 | 0.8679 | 0.7647 |
| $H(L^w|Y)$ | **0.9030** | **0.8915** | **0.7778** |

Table 2: Correlation analysis between different metrics and the generalization gap with 20% label noise. $\hat{I}$, $\check{I}$, $\tilde{I}$ represents different strategies for mutual information estimation.

epochs with a learning rate of $0.01$. The conditional and unconditional loss entropies are estimated using a simple Gaussian kernel density estimator, where the kernel width is automatically selected by the well-known rule-of-thumb criterion. The code for this experiment is largely based on the implementation from Kawaguchi et al. (2023)[1].

Besides Pearson correlation analysis, we further present the estimated coefficients of Spearman and Kendall correlations in Table 1. Additionally, we conduct experiments with 20% label noise to examine the robustness of our bounds against noises in Table 2, i.e. 20% of the labels are replaced by

---

[1]https://github.com/xu-ji/information-bottleneck

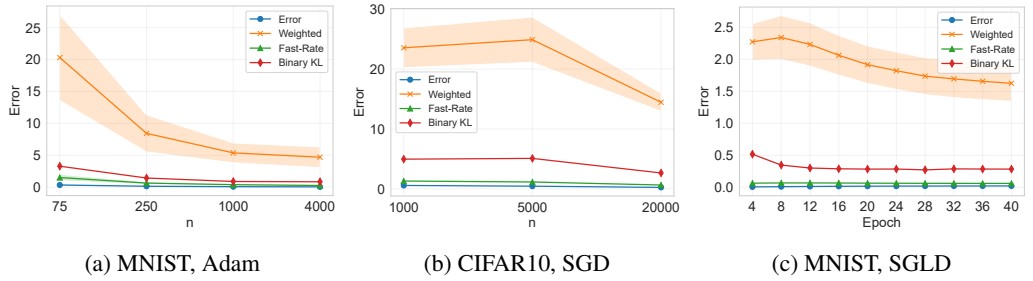

(a) MNIST, Adam    (b) CIFAR10, SGD    (c) MNIST, SGLD

Figure 3: Comparison of the generalization gap in 3 different deep-learning scenarios, along with theoretical upper bounds including the weighted bound (Theorem 5), the fast-rate bound (Theorem 6) and the binary KL bound (Theorem 7 in (Hellström & Durisi, 2022)).

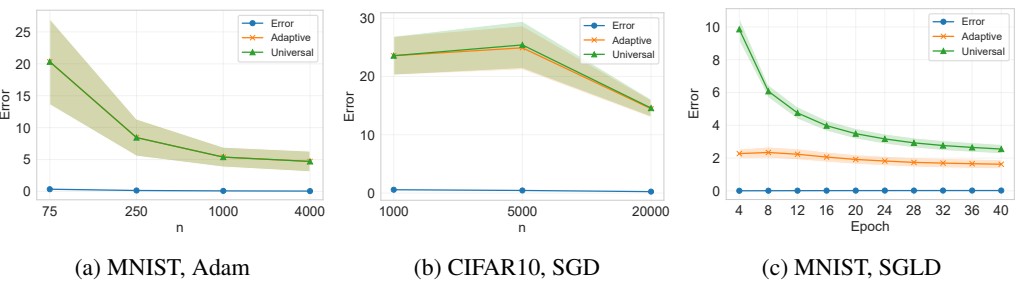

(a) MNIST, Adam    (b) CIFAR10, SGD    (c) MNIST, SGLD

Figure 4: Comparison of the generalization gap in 3 different deep-learning scenarios, along with the weighted generalization bound (Theorem 5) using adaptive $C_i$ values (Adaptive) or a universal value for all $C_i$ (Universal).

randomly generated ones. As can be seen in Table 1 and 2, the loss entropy metrics are consistently the better indicators of generalization.

## F.2   REAL-WORLD LEARNING TASKS

We conduct 3 real-world learning scenarios to evaluate different generalization bounds: 1) MNIST 4 vs 9 classification using Adam, 2) MNIST 4 vs 9 classification using SGLD, and 3) CIFAR-10 classification with fine-tuned ResNet-50. The loss function is cross-entropy.

In each experiment under the supersample (leave-one-out) setting, we draw $k_1$ samples of $\tilde{S}_s$ ($\tilde{S}_l$), each involving randomly sampling $2n$ ($n+1$) samples from the corresponding dataset. For each $\tilde{S}_s$ ($\tilde{S}_l$), we then draw $k_2$ samples of the training / test split variable $\tilde{U}$ ($U$), resulting in total of $k_1 \times k_2$ independent runs. The value of $k_1$, $k_2$ and experimental settings are kept the same as Harutyunyan

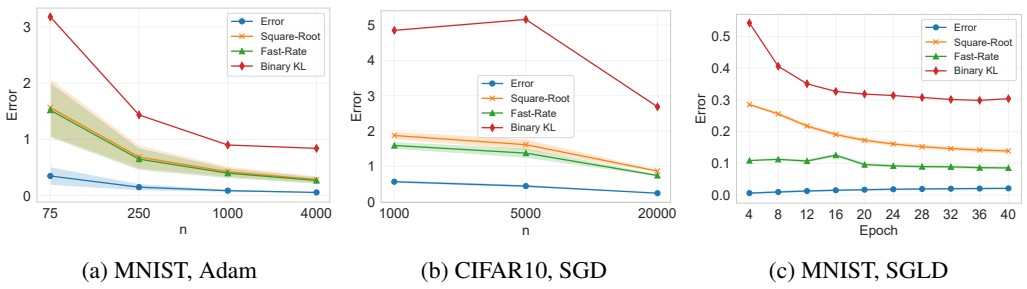

(a) MNIST, Adam    (b) CIFAR10, SGD    (c) MNIST, SGLD

Figure 5: Comparison of the generalization gap in 3 different deep-learning scenarios, along with theoretical upper bounds including the square-root bound (Theorem 4), the fast-rate bound (Theorem 6) and the binary KL bound (Theorem 7 in (Hellström & Durisi, 2022)) using bin size 0.6.

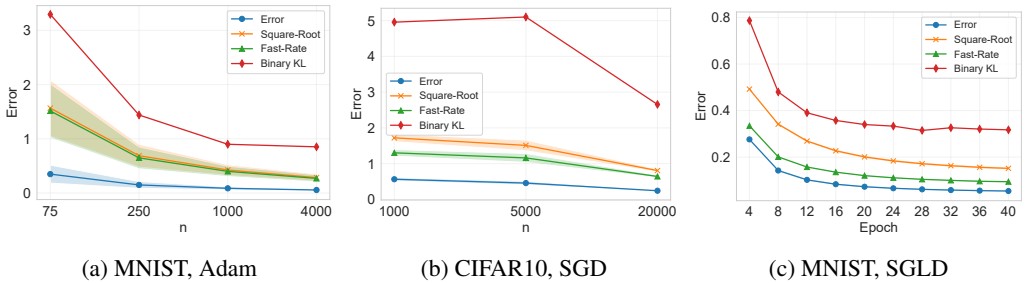

Figure 6: Comparison of the test risk in 3 different deep-learning scenarios, along with theoretical upper bounds for the test risk including the square-root bound (Theorem 4), the fast-rate bound (Theorem 6) and the binary KL bound (Theorem 7 in (Hellström & Durisi, 2022)).

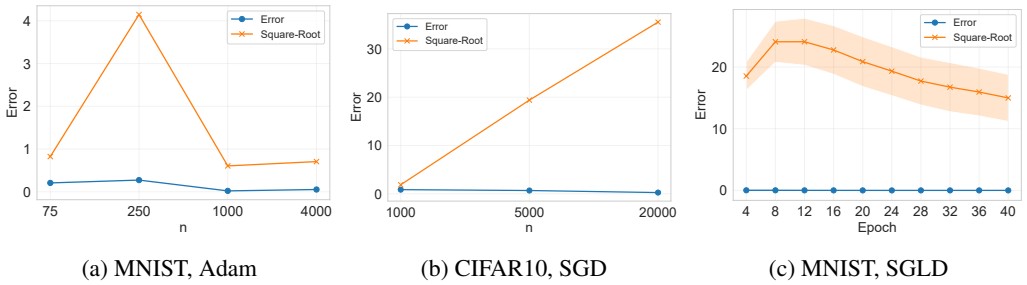

Figure 7: Comparison of the generalization gap and the theoretical upper bound (Theorem 3) under the leave-one-out setting.

et al. (2021)[2]. The discretization strategy discussed in Section G.3 is adopted to discretize each loss value with a bin size of $1.0$. A union bound of our main results and Proposition 3 is then taken to provide valid generalization bounds for continuous loss functions.

Although the RHS of Theorem 4 and 6 directly involve the test losses, the exact values of test losses are not required to evaluate the key coefficients $\tilde{C}_1^W$ or $\tilde{G}_3^W$ in these theorems (Note that we can directly take $\hat{L}_i^W, \hat{L}_i^{W,\kappa} = \kappa$ in Theorem 5 and 6). For any $w$, let $Z \sim \mu$ and $L = \ell(w, Z)$. Then it suffices to adopt a few validation samples to estimate $\mathbb{E}[L]$ and $\mathbb{E}[L^2]$. By applying Markov's inequality, we have:

$$
\mathbb{P}\left(\frac{1}{n}\sum_{i=1}^n (\Delta L_i^W)^2 \geq \epsilon\right) \leq \frac{\frac{1}{n}\sum_{i=1}^n \mathbb{E}[(\Delta L_i^W)^2]}{\epsilon}
$$
$$
= \frac{\frac{1}{n}\sum_{i=1}^n \mathbb{E}\left[(L_{i,\tilde{U}_i}^W)^2 + (L_{i,1-\tilde{U}_i}^W)^2 - 2L_{i,\tilde{U}_i}^W L_{i,1-\tilde{U}_i}^W\right]}{\epsilon}
$$
$$
= \frac{\frac{1}{n}\sum_{i=1}^n (L_{i,\tilde{U}_i}^W)^2 + \mathbb{E}[L^2] - 2\mathbb{E}[L](\frac{1}{n}\sum_{i=1}^n L_{i,\tilde{U}_i}^W)}{\epsilon}.
$$

Therefore, one can easily acquire a quantile estimation of $\tilde{C}_1^W$ or $\tilde{G}_3^W$ through a few validation samples, which is also necessary when training models in practice.

The hyper-parameters are automatically chosen by optimization using the Scipy package: Specifically, $\lambda$ is chosen by the L-BFGS-B algorithm since it is derivable; $\gamma$ and $\kappa$ are chosen by the Nelder-Mead optimization algorithm; and the binary KL measures in (Hellström & Durisi, 2022) are solved by the brentq algorithm. It is noteworthy the hyper-parameters should not be directly optimized over the RHS of our bounds, since they are assumed to be constants and should be independent of the draw of $W$, $\tilde{S}_s$ and $\tilde{U}$. Instead, note that these bounds can be expressed as $\mathbb{P}(\Delta \leq B) \geq 1-\delta$, where the validation error $\Delta$ and the upper bound $B$ are both random variables. By taking $b$ as the expec-

---

[2]https://github.com/hrayrhar/f-CMI/tree/master

tation of $B$ and then optimizing the hyper-parameters over $b$, we are able to fulfill the prerequisites of these bounds and keep the evaluation valid after optimization.

We select $\delta = 0.5$ to simulate the mean of generalization bounds. It is worth noting that the binary KL bound (Theorem 7 in (Hellström & Durisi, 2022)) is not directly tractable as it involves a high-dimensional stochastic KL divergence quantity:

**Theorem 10.** *(Theorem 7 in Hellström & Durisi, 2022) Assume that $\ell(\cdot, \cdot) \in [0, 1]$, then with probability at least $1 - \delta$ over the draw of $\tilde{S}_s$ and $\tilde{U}$,*

$$d\left( \mathbb{E}_{W|\tilde{S}_s, \tilde{U}}\left[ \frac{1}{n} \sum_{i=1}^n L_{i,\tilde{U}_i}^W \right] \, \middle\| \, \mathbb{E}_{W|\tilde{S}_s, \tilde{U}}\left[ \frac{1}{2n} \sum_{i=1}^n (L_{i,0}^W + L_{i,1}^W) \right] \right)$$

$$\leq \frac{\mathrm{KL}\left( P_{\tilde{R}^W|\tilde{S}_s, \tilde{U}} \, \middle\| \, P_{\tilde{R}^W|\tilde{S}_s} \right) + \log\left( \frac{2\sqrt{n}}{\delta} \right)}{n},$$

*where $d(p \,\|\, q) = p \log(p/q) + (1 - p) \log((1 - p)/(1 - q))$ is the binary KL divergence.*

In order to apply comparisons, we average the binary KL bound above over multiple draws of $\tilde{U}, \tilde{S}_s$. In this way, the stochastic KL divergence can be approximated by:

$$\mathbb{E}_{\tilde{U}, \tilde{S}_s}\left[ \mathrm{KL}\left( P_{\tilde{R}^W|\tilde{S}_s, \tilde{U}} \, \middle\| \, P_{\tilde{R}^W|\tilde{S}_s} \right) \right] = I(\tilde{R}^W; \tilde{U}|\tilde{S}_s).$$

Additionally, we will show that $\sum_{i=1}^n I(L_{i,0}^W, L_{i,1}^W; \tilde{U}_i)$ serves as a lower-bound approximation for $I(\tilde{R}^W; \tilde{U}|\tilde{S}_s)$:

**Proposition 2.** $\sum_{i=1}^n I(L_{i,0}^W, L_{i,1}^W; \tilde{U}_i) \leq I(\tilde{R}^W; \tilde{U}|\tilde{S}_s)$.

*Proof.* For convenience, we denote $\{\tilde{U}_i\}_{i=1}^n$ as $\tilde{U}_{1:n}$. Then

$$
\begin{aligned}
I(\tilde{R}^W; \tilde{U}|\tilde{S}_s) &= I(\tilde{R}^W; \tilde{U}|\tilde{S}_s) + I(\tilde{U}; \tilde{S}_s) \\
&= I(\tilde{R}^W; \tilde{U}) + I(\tilde{U}; \tilde{S}_s|\tilde{R}^W) \\
&\geq I(\tilde{R}^W; \tilde{U}) \\
&= I(\tilde{R}^W; \tilde{U}_1) + I(\tilde{R}^W; \tilde{U}_{2:n}|\tilde{U}_1) \\
&= I(\tilde{R}^W; \tilde{U}_1) + I(\tilde{R}^W; \tilde{U}_{2:n}) - I(\tilde{U}_{2:n}; \tilde{U}_1) + I(\tilde{U}_{2:n}; \tilde{U}_1|\tilde{R}^W) \\
&= I(\tilde{R}^W; \tilde{U}_1) + I(\tilde{R}^W; \tilde{U}_{2:n}) + I(\tilde{U}_{2:n}; \tilde{U}_1|\tilde{R}^W) \\
&\geq I(\tilde{R}^W; \tilde{U}_1) + I(\tilde{R}^W; \tilde{U}_{2:n}) \\
&\quad \cdots \\
&\geq \sum_{i=1}^n I(\tilde{R}^W; \tilde{U}_i) \\
&= \sum_{i=1}^n I(L_{i,0}^W, L_{i,1}^W; \tilde{U}_i) + I(\tilde{R}^W \setminus \{L_{i,0}^W, L_{i,1}^W\}; \tilde{U}_i|L_{i,0}^W, L_{i,1}^W) \\
&\geq \sum_{i=1}^n I(L_{i,0}^W, L_{i,1}^W; \tilde{U}_i).
\end{aligned}
$$

$\square$

We then evaluate the effectiveness of the proposed thresholding method. As shown in Figure 3, our fast-rate generalization bound (Theorem 6) is considerably tighter than the weighted generalization bound (Theorem 5). Without the thresholding technique, the fast-rate bounds fail to outperform the lower-bound approximation of the binary KL bound. Similarly, Figure 4 demonstrates the effectiveness of adaptively choosing coefficients $C_i$ for each individual training loss. It can be seen that this strategy is especially useful in the early stage of the training process, where the magnitude of training losses is comparable to test losses and remains to be minimized. This behavior is further

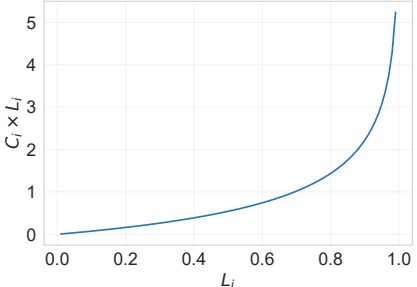

Figure 8: The influence of training loss $L$ using the adaptive $C_i$ strategy with $\kappa = 1$ in Theorem 5.

verified in Figure 8, that the adaptive $C_i$ strategy particularly lowers the bound when the training losses are close (but not equal) to the threshold. For well-trained networks where training losses approach zero, the improvement will be less significant.

To examine the impact of bin size on the visualization results, we additionally report the comparison between generalization bounds with a bin size of $0.6$ in Figure 5. In general, decreasing the bin size results in larger loss entropy values but lower discretization errors. Therefore, there exists a trade-off between these two quantities to acquire the tightest generalization bound. Moreover, we visualize the comparison between different bounds for the test risk instead of the generalization error in Figure 6. Note that each generalization bound directly implies a test risk bound by adding the training risk to both sides.

We further report the comparison between our leave-one-out generalization bound (Theorem 3) and the generalization gap in Figure 7. Although this bound does not capture the behavior of the generalization gap well for large $n$ values, we highlight that this is the first computationally tractable high-probability generalization bound under the leave-one-out setting.

## G    FURTHER DISCUSSIONS

### G.1    ADDITIONAL RELATED WORKS

High-probability generalization bounds in the literature could be categorized by the distribution they are taken over (Hellström et al., 2023):

- PAC-Bayesian bounds, which hold with high probability over the draw of the training dataset, but are averaged over the learning algorithm;
- Single-draw bounds, which hold with high probability over the draw of both the dataset and a single hypothesis;
- Mean-hypothesis bounds, high probability bounds of the average hypothesis output from the learning algorithm, given the dataset.

Exponential stochastic inequalities (ESI) were initially employed to derive generalization bounds in seminal works by (Zhang, 2006; Catoni, 2007). Subsequently, these bounds were formalized by (Koolen et al., 2016; Mhammedi et al., 2019; Grünwald & Mehta, 2020). The general form of PAC-Bayesian bounds has been established by (Germain et al., 2009; Bégin et al., 2014; Rivasplata et al., 2020), leading to various specific bounds that leverage the sub-gaussian nature of loss functions (Hellström & Durisi, 2020) and binary KL upper bounds for the population risk given the empirical risk with bounded loss functions (McAllester, 2013; Foong et al., 2021).

Data-dependent priors based on data splitting were introduced by (Ambroladze et al., 2006) and further extended by (Mhammedi et al., 2019; Dziugaite et al., 2021). Further investigations into data-dependent priors through the lens of differential privacy were conducted by (Dziugaite & Roy, 2018; Rivasplata et al., 2020). The exploration of distribution-dependent priors is discussed by Catoni (2007); Lever et al. (2013). It is also noteworthy that an essentially equivalent approach to the CMI framework was introduced in the PAC-Bayesian context much earlier by (Audibert, 2004;

Catoni, 2007) to mitigate the variance of PAC-Bayesian generalization bounds, under the name of "almost exchangeable priors" or "transductive learning".

Beyond PAC-Bayesian bounds, Catoni (2007) noted that analogous techniques can be employed to derive single-draw bounds from the posterior. Rivasplata et al. (2020) leverages the Radon-Nikodym derivative to formulate single-draw bounds beyond relative entropy, which are further extended by (Hellström & Durisi, 2021b;a). Esposito et al. (2021) linked single-draw generalization error with Rényi-, $f$-Divergences and Maximal Leakage. Xu & Raginsky (2017) adapted the monitor technique to transform average generalization bounds into single-draw counterparts.

### G.2 CONVERGENCE RATE OF LOO BOUNDS

In the interpolating regime where the training risk approaches zero, it is shown in Theorem II.1 of (Haghifam et al., 2022) that the LOO bound for the expected generalization error scales with $1/\log(n)$, while our Theorem 3 does not exhibit such property. However, it is worth noting that the dimensionality of the LOO-CMI term $I(R^W; U|\tilde{S}_l)$ is still proportional to $n$ even in the interpolating regime, so this does not necessarily lead to tighter or computationally tractable bounds when $n$ is large. In contrast, our loss entropy measure $H(R^W)$ reduces to the entropy of a single 1-dimensional variable, making the bound directly tractable.

### G.3 DISCRETIZATION OF CONTINUOUS LOSS

In practice, the loss function is usually continuous (e.g. cross-entropy). While the loss values can be treated as discrete variables by the fact that they are stored in digital computers with floating-point numbers, it is impractical to estimate the loss entropy with machine-precision binning size. Instead, we show that the loss values could be discretized with an arbitrary bin size and still acquire valid generalization bounds. Let $b > 0$ be the bin size and $\phi_b(x)$ be the rounding function of base $b$:

$$\phi_b(x) = b \times \arg\min_{i \in \mathbb{N}} |ib - x|.$$

Since the discretization error of the training risk is directly tractable, we only consider the discretization error of the test risk:

**Proposition 3.** *For any $w \in \mathcal{W}$ and $b > 0$, let $\{Z_i\}_{i=1}^n \sim \mu^n$ be i.i.d samples, $L_i = \ell(w, Z_i)$ and $\{D_i\}_{i=1}^n \sim \mathrm{Unif}([-\frac{b}{2}, \frac{b}{2}]^n)$ be i.i.d uniform variables. Then with probability at least $1 - \delta$, we have*

$$\frac{1}{n}\sum_{i=1}^n L_i - \frac{1}{n}\sum_{i=1}^n \phi_b(L_i + D_i) \le b\sqrt{\frac{2\log(\frac{1}{\delta})}{n}}.$$

*Proof.* Let $\hat{L}_i = L_i - \phi_b(L_i + D_i)$, then it is easy to verify that $\mathbb{E}_{D_i}[\hat{L}_i] = 0$ and $\hat{L}_i \in [-b, b]$, i.e. $\hat{L}_i$ is $b$-subgaussian. Since $\{L_i\}_{i=1}^n$ and $\{D_i\}_{i=1}^n$ are both i.i.d, we have that $\frac{1}{n}\sum_{i=1}^n \hat{L}_i$ is $\frac{b}{\sqrt{n}}$-subgaussian. Therefore,

$$\mathbb{P}\left(\frac{1}{n}\sum_{i=1}^n \hat{L}_i - \mathbb{E}\left[\frac{1}{n}\sum_{i=1}^n \hat{L}_i\right] \ge \epsilon\right) \le \exp\left(-\frac{n\epsilon^2}{2b^2}\right).$$

By taking $\delta$ as the RHS of the inequality above, we then have that with probability at least $1 - \delta$,

$$\frac{1}{n}\sum_{i=1}^n \hat{L}_i \le b\sqrt{\frac{2\log(\frac{1}{\delta})}{n}}.$$

The proof is complete. $\qquad\square$

Therefore, by perturbing each test loss with $D_i \sim \mathrm{Unif}([-\frac{b}{2}, \frac{b}{2}])$ and then rounding to the nearest bin, the loss values could be discretized without significant impact on the validation error. In this way, one can directly evaluate our main theorems with the discretized losses, and then take the union bound with Proposition 3 to acquire generalization upper bounds for continuous loss functions. Note that in our main Theorems 3 - 6, the losses are not required to be deterministic given the model $W$ and sample $Z$, so these bounds remain valid after introducing external randomness $D_i$ by taking the probability over the draw of $W$, $\tilde{S}_s$ (or $\tilde{S}_l$), $\tilde{U}$ (or $U$) and $D_i$.

## G.4 THE INTERPOLATING REGIME

It is common to observe that at the end of the training progress, the network overfits the training dataset and approaches 100% training accuracy. Such a phenomenon corresponds to the interpolating regime, which assumes that the network can always fit the given training dataset and achieve zero empirical risk. This assumption leads to certain simplifications and reveals many intriguing properties of information-theoretic generalization bounds (Hellström & Durisi, 2022; Wang & Mao, 2023). For example, the key information quantities in our main Theorems 3 - 6 could be simplified to the entropy of samplewise test loss. Furthermore, Theorem 5 achieves a convergence rate of $O(1/n)$ when the empirical risk approaches zero. It would be tempting to connect generalization analysis under over-parameterization frameworks (e.g. NTK (Chen et al., 2020) and Mean-field (Nitanda et al., 2021; Nishikawa et al., 2022; Aminian et al., 2023)) that investigate the interpolating regime, with information-theoretic generalization bounds.

