# OpenReview forum: "Rethinking Information-theoretic Generalization: Loss Entropy Induced PAC Bounds"
_ICLR.cc/2024/Conference — ICLR 2024 poster_

### Official Review · Reviewer_Fiyn · 2023-10-29

**Soundness:** 3 good
**Presentation:** 3 good
**Contribution:** 3 good
**Rating:** 8
**Confidence:** 4

**Summary:**

This paper propose a series of PAC information-theoretic generalization error bounds via entropy measures (Shannon entropy and Renyi entropy) of loss. Specifically, a data-independent case is first considered that gives a entropy related characterization of concentration. Then in the data-dependence case, PAC-generalization error bounds by entropy measure are proposed for both leave-one-out and supersample settings, and fast-rate bounds.

**Strengths:**

The proposed bounds based on loss entropy is advantagous since the loss is one-dimensional and easier to estimate. Many scenarios are considered, e.g., leave-one-out, supersample, and fast-rate. As far as I can tell, the proofs are sound. The theory and quantities in the bounds are well-explained. Experiemntal studies show that loss entropy-based bounds have the highest correlation with the error, and gives tighter numerical bounds in many datasets with different optimization algorithms.

**Weaknesses:**

There is no particular weakness of the paper.

**Questions:**

Should we assume the loss has finite alphabet so as to allow well defined Shannon entropy?

Minor:

First paragraph of Section 3, "We begin by enhancing existing upper bounds presented in (Kawaguchi et al., 2022; 2023)", should it be Kawaguchi et al., 2023 only?

First paragraph of Section 5, "chaining strategy" and "random subsets or individual" with the same ref Zhou et al. 2022.

---

> ### Author Response · Authors · 2023-11-15
> **Response to Reviewer Fiyn**
>
> Dear Reviewer Fiyn, thanks for your valuable comments! We address your raised questions as follows:
>
> **Should we assume the loss has finite alphabet so as to allow well-defined Shannon entropy?**
>
> Yes, the losses are assumed to be discrete and have finite cardinality. This assumption aligns well with digital computers employing floating-point numbers.
>
> **Other minor questions.**
>
> Thank you for the suggestions. These problems have been fixed in the updated manuscript.

---

> > ### Comment · Reviewer_Fiyn · 2023-11-23
> >
> > Thank you. I keep my positive rating of the paper.

---

### Official Review · Reviewer_YPBm · 2023-10-30

**Soundness:** 2 fair
**Presentation:** 3 good
**Contribution:** 4 excellent
**Rating:** 6
**Confidence:** 4

**Summary:**

In this paper, several new information-theoretic generalization bounds are derived. These bounds rely on the entropy of the losses of the model. While the proof techniques take inspiration from the information bottleneck approach of Kawaguchi et al, this is combined with the evaluated CMI perspective in the framework of Steinke and Zakynthinou. Both the leave-one-out and standard supersample settings are considered, and fast-rate bounds are derived. The usefulness of the bounds is demonstrated in several ways, both in terms of their correlation with the true error, and in terms of their numerical tightness.

**Strengths:**

The paper combines ideas from the work of Kawaguchi et al regarding typicality proofs with e-CMI approaches from the line of work starting with Steinke & Zakynthinou, and adds several new ideas on top of this. Information-theoretic generalization bounds have demonstrated the potential to be powerful tools for studying, e.g., deep nets--although their usefulness is still not entirely clear--so the advances are significant.

The resulting bounds are highly appealing from many perspectives, in that the bounds depend on information measures that are shown to be tighter than existing work in relevant scenarios; high-probability bounds are derived with samplewise metrics and a log-dependency on delta; the quantities are (comparatively) easy to estimate; and the bounds are numerically tight in the studied scenarios.

The experimental evaluation is extensive, and significant efforts are made to make fair comparisons to prior work (arguably, they are even unfair to the benefit of the prior work).

The paper contains many instructive comparisons to prior work and discussion of the presented bounds and proof techniques. While the results are not always easy to parse due to the heavy notation and the fact that they simply have a relatively complicated form, I appreciate the "progressive complexity" of the presentation: the results are first stated in terms of generic constants, to get an overall view of the result, but they are still explicitly defined immediately after, allowing the reader to see the full details.

**Weaknesses:**

Most of the weaknesses arise from the application to potentially unbounded losses. If the results are specialized to the 0-1 loss, all of the notable issues seem to disappear.

There seem to be some issues with the use of sub-Gaussianity in some proofs (see questions).

Regarding the numerical evaluation, the effect of the binning is not at all discussed, but seems like it can have a huge impact. Furthermore, there seem to be some issues with the optimization of the bounds, as well as the comparison with the binary KL bound. All of these are discussed in the questions below.

The bounds assume discrete loss functions, which enables the use of the entropy and the typicality arguments. Still, the bounds are applied to, e.g., the cross-entropy loss, which is continuous, with the motivation that all losses can be considered discrete due to floating-point numbers. I am not sure if this is convincing. Increasing the precision used would affect the cardinality of the loss, and potentially degrade the bounds. This does not seem like reasonable behavior, and would require a stronger motivation.

In the data-independent bounds, it is assumed that the features are generated from a function that depends on an $m$-dimensional nuisance variable, and that the model has finite sensitivity to the nuisances, even in a worst-case sense. This seems like quite a strong modelling assumption, but is not motivated or discussed in detail.

Theorem 3 is said to be most useful in the interpolating setting. The interpolating bound of Haghifam et al decays with log(n), unlike the presented bound. This merits a mention and discussion.

Several of the bounds on the generalization gap have an explicit dependence on the value of the test loss itself on the right-hand side. For bounded losses, this can simply be upper-bounded, but otherwise it seems quite puzzling.

The bounds depend on average quantity (entropy), and are hence not empirical. This may merit a disccussion.

PAC-Bayesian bounds are not discussed much, despite being closely related. They are particularly relevant since you point out that recent bounds are “primarily restricted to average-case scenarios”. Moreover, the supersample setting is equivalent to almost exchangeable priors (Audibert, “A better variance control for PAC-Bayesian classification,” 2004 and Catoni, 2007). It also seems relevant to compare to single-draw bounds from, e.g., Esposito et al., "Generalization Error Bounds Via Rényi-, f-Divergences and Maximal Leakage”, 2020.

Some other minor points are discussed in the questions below.

*Update*: All of these points have been mostly clarified or resolved. The sub-Gaussianity arguments are not super-clear but appear to be fine. The paper now includes a more well-motivated evaluation procedure with regards to the bins, as well as a study of their effect on the bounds. Related work and discussion points have been added. The appearance of the test-loss in the bounds has been shown to be avoidable in some cases, while for others it is argued that it can be reasonably estimated with validation data---although this definitely limits the applicability of the bounds and should be made very clear. I updated the score to a 6 from a 5 following the discussion.

**Questions:**

(See "*update*" at the end of weaknesses for summary of responses)

As noted in Theorem 3, the proofs require sub-Gaussianity under the _selection_ random variable, which has to hold for any fixed instance of the loss matrix/vector. This seems to get lost in the later derivations. First, in Lemma 14, it is assumed that $\ell(w,X)$ is sub-Gaussian. This seems to mean that it is sub-Gaussian when taking the expectation over $X$. However, in the proof, it is needed that $\ell(w,x_U)-\ell(w,x_{1- U}) $ is sub-Gaussian under $U$ for all $w$, $x_0$, and $x_1$ (or under the distribution where $X$ and $W$ are distributed jointly and $U$ is drawn independently). This does not follow from the sub-Gaussianity that is assumed. There are cases beyond bounded losses that one can consider (e.g., Steinke & Zakynthinou, Thm. 5.1 and Sec 5.4). Is my reading correct or did I miss something? The same issue occurs in Theorem 9.

Also, for, e.g., the equation preceding (29), it is written $P( \Delta(W, \tilde S_l, U) \geq t )$. This is misleading, as it appears that the probability holds over a joint draw of $(W, \tilde S_l, U)$, but in fact, only $U$ is random while $W$ and $\tilde S_l$ is fixed. Is this correct?

At the end of p.4: the supersample generalization gap approaches the true one, but the same is not true for the leave-one-out, as the test loss is still evaluated on only one sample. Does this limit the usefulness of the LOO results?

In the numerical evaluations, the cross-entropy loss is used, which is continuous. To compute bounds for MNIST etc, a binning with a bin size of 0.5 is used. The usefulness of the results, despite the assumption of discreteness, was motivated by the fact that floating-point numbers are used in practice. Certainly, machine precision is significantly smaller than 0.5. This specific choice is also not motivated. What is the effect when decreasing the binning size? Is there any reason to believe that  this is a sensible choice?

In the numerical evaluation, optimization is performed over several constants in the bounds. In the derivation, they are assumed fixed, and the bounds hold with a certain probability for this fixed value. In order to have a valid bound while optimizing, one needs to perform some kind of union bound, or other argument to guarantee the validity. Otherwise, it should be stated that the bounds are not actually valid due to this optimization, but are just illustrative. Did I miss something that motivated this optimization?

The bounds are said to be compared to the Binary KL bound of Hellstrom and Durisi, but Theorem 10 in the paper is not a binary KL bound. Furthermore, in HD Thm. 7, the information measure that appears is not a CMI, but a random KL divergence depending on the supersample and selection variable. Also, the binary KL bound does not imply a bound on the generalization gap, but instead, directly bounds the population loss given a training loss. Finally, the binary KL bound is inherently only valid for bounded losses, as the binary KL is undefined for inputs outside of $[0,1]$. Is the comparison actually to Thm. 10, i.e., the square-root bound of Hellstrom and Durisi?

The plots show the generalization gap, rather than the bound on the population loss itself. Arguably, the latter is more interesting, since a low generalization gap with a high training loss is not very useful. Is it possible to show the bound on the population loss?

***

I acknowledge that I may have missed something or misunderstood some of the arguments. If the issues above are addressed, especially regarding the soundness of the results and their evaluation, the relevant ratings will be significantly changed.

***

Minor points and typos:

While the somewhat unfortunate acronym LOO appears to be standard, the even more unfortunate acronym SS has, as far as I know, not been used before. Perhaps it is preferable to avoid it.

I am not sure the way the term “PAC” is used is really consistent with PAC learning. PAC bounds are typically uniform in some sense, whereas the bounds in this paper are tail bounds under the joint distribution of data and hypothesis.

Before Thm. 1: grammar issue

p.5, “The most notable improvement of…” here, a comparison is made to CMI bounds instead of e-CMI bounds, which are arguably more relevant; “This illustrates a novel trade-off” is it really a trade-off? Both are just preferred to be small.

For the notation of $L^{\kappa}$: perhaps it would be more intuitive to use $L^{<\kappa}$ and $L^{>\kappa}$?

In Lemma 12, it is written $\eta = ( …, … )$. Should this be $\eta \in (… , …)$?

---

> ### Author Response · Authors · 2023-11-15
> **Response to Reviewer YPBm (1/4)**
>
> Dear Reviewer YPBm, thank you for the thorough reading and constructive comments! They are really helpful for us to further improve our manuscript. In the following response, we will address your concerns one by one:
>
> **Most of the weaknesses arise from the application to potentially unbounded losses. If the results are specialized to the 0-1 loss, all of the notable issues seem to disappear.**
>
> Note that the applications of our bounds are not limited to classification but also regression tasks, where the 0-1 loss is no longer applicable. Even for classification tasks with 0-1 loss, the previous binary KL bound is still intractable due to its high-dimensional nature. This is the main benefit of our results compared to the previous bounds.
>
> **The bounds assume discrete loss functions, which enables the use of the entropy and the typicality arguments. Still, the bounds are applied to, e.g., the cross-entropy loss, which is continuous, with the motivation that all losses can be considered discrete due to floating-point numbers. I am not sure if this is convincing. Increasing the precision used would affect the cardinality of the loss, and potentially degrade the bounds. This does not seem like reasonable behavior, and would require a stronger motivation.**
>
> We highlight that the discrete assumption is common and adopted in previous works (e.g. Shwartz-Ziv et al., 2019, Kawaguchi et al., 2022; 2023). It allows the adoption of the well-defined discrete Shannon's entropy, which is always positive, and facilitates developing generalization bounds based on $1$-dimensional information measures. Otherwise, the differential entropy can be arbitrarily negative, so the generalization analysis will be restricted to mutual information measures, which are at least $2$-dimensional and therefore harder to estimate.
>
> In addition, such a discrete scheme enables a brand-new technique beyond the Donsker-Varadhan formula (Lemma 3) for decoupling random variables by separating the "typical subset" and then taking the union bound over them. The new technique, detailed in the paragraph preceding Section 3.2.1, is pivotal for our generalization analysis and enjoys broad extendability beyond leave-one-out and supersample settings.
>
> Lastly, many learning tasks naturally satisfy the discrete assumption, including semantic segmentation (with risk in pixels), discrete ordinal regression, generative models with hamming distance losses, and ranking losses over multiple objects. It is also possible to discretize the loss function before evaluation, e.g. round to 1 decimal. Such strategies are widely adopted in information-theoretic learning tasks (Biesiada et al., 2005) (Vinh et al., 2014), and only cause a mild impact on the generalization error.
>
> Shwartz-Ziv R, Painsky A, Tishby N. Representation compression and generalization in deep neural networks.
>
> Biesiada J, Duch W, Kachel A, Maczka K, Palucha S. Feature ranking methods based on information entropy with parzen windows.
>
> Vinh N, Chan J, Bailey J. Reconsidering mutual information based feature selection: A statistical significance view.
>
> **In the data-independent bounds, it is assumed that the features are generated from a function that depends on an $m$-dimensional nuisance variable, and that the model has finite sensitivity to the nuisances, even in a worst-case sense. This seems like quite a strong modeing assumption, but is not motivated or discussed in detail.**
>
> The nuisance variables are assumed to be the source of randomness in $X$ given $Y$. Their existence is well-motivated, e.g. for an animal classification task, the background, the position, or the behavior of animals do not affect the true classification label, and are therefore nuisance variables. It is also natural to assume that $m$ is finite and smaller than the dimensionality of $X$. For a given hypothesis $w$, one can determine the set of possible loss values $\mathcal{L}^w$, which is assumed to be finite. Then it is trivial to show that the sensitivity is finite by noticing $c^w \le \sup_{l_1,l_2 \in \mathcal{L}^w}|\log P(L^w = l_1) - \log P(L^w = l_2)|$. Since $m$ and $c^w$ are both constants irrelevant to $n$, these terms could be simply ignored for asymptotical analysis when $n \rightarrow \infty$, and allow us to mainly focus on $H(L^w)$ or $H(L^w|Y)$.
>
> **Theorem 3 is said to be most useful in the interpolating setting. The interpolating bound of Haghifam et al decays with log(n), unlike the presented bound. This merits a mention and discussion.**
>
> The generalization errors in (Haghifam et al., 2022) are expressed as expectations and thus are not directly comparable with our results. We add relevant discussions in Section G.2.

---

> ### Author Response · Authors · 2023-11-15
> **Response to Reviewer YPBm (2/4)**
>
> **Several of the bounds on the generalization gap have an explicit dependence on the value of the test loss itself on the right-hand side. For bounded losses, this can simply be upper-bounded, but otherwise it seems quite puzzling.**
>
> Note that evaluating these bounds only requires the losses to be finite but not necessarily bounded. Even for unbounded loss functions, the losses are still finite in practice, e.g. one always gets finite training and validation loss values using the cross-entropy loss. For any long-tailed loss distribution, we have $P(L_{i,0}^W < \infty) = 1$, so the right-hand side of these bounds is always finite in practice.
>
> **The bounds depend on average quantity (entropy), and are hence not empirical. This may merit a discussion.**
>
> Indeed, the underlying distributions of the losses are unknown in practice and can only be approximated by sampling. This requires repeating the training progress several times to acquire sufficient samples. The most widely used entropy estimators are kernel density estimation and binning methods, which are both adopted in this paper as we discussed in Section F.
>
> **PAC-Bayesian bounds are not discussed much, despite being closely related. They are particularly relevant since you point out that recent bounds are “primarily restricted to average-case scenarios”. Moreover, the supersample setting is equivalent to almost exchangeable priors (Audibert, “A better variance control for PAC-Bayesian classification,” 2004 and Catoni, 2007). It also seems relevant to compare to single-draw bounds from, e.g., Esposito et al., "Generalization Error Bounds Via Rényi-, f-Divergences and Maximal Leakage”, 2020.**
>
> We replenish discussions with these related works in the Appendix, see Section G.1.
>
> **As noted in Theorem 3, the proofs require sub-Gaussianity under the selection random variable, which has to hold for any fixed instance of the loss matrix/vector. This seems to get lost in the later derivations. First, in Lemma 14, it is assumed that $\ell(w,X)$ is sub-Gaussian. This seems to mean that it is sub-Gaussian when taking the expectation over $X$. However, in the proof, it is needed that $\ell(w,x_U) - \ell(w,x_{1-U})$ is sub-Gaussian under $U$ for all $w$, $x_0$, and $x_1$ (or under the distribution where $X$ and $W$ are distributed jointly and $U$ is drawn independently). This does not follow from the sub-Gaussianity that is assumed. There are cases beyond bounded losses that one can consider (e.g., Steinke \& Zakynthinou, Thm. 5.1 and Sec 5.4). Is my reading correct or did I miss something? The same issue occurs in Theorem 9.**
>
> In Theorem 3, the sub-gaussianity is only required for the single draw of $W$ and $\tilde{S}\_l$. In other words, $\Sigma_{R^W}$ is a random variable that is determined by $R^W = \\{L_1^W, \cdots, L_{n+1}^W\\}$. $\Sigma_{R^W}$ can be treated as a tightened version of $B^{W,\tilde{S}\_l}$, e.g. given two losses $R^W = \\{0, 2\\}$, we have $\Sigma_{R^W} = 1 < B^{W,\tilde{S}_l} = 2$.
>
> The typo in Lemma 14 and Theorem 9 has been fixed. We are actually assuming the sub-gaussianity of the loss variables $L_{i,0}^W$ and $L_{i,1}^W$. It is easy to prove that the sum of two (possibly dependent) $\sigma$-subgaussian variables is $2\sigma$-subgaussian by its definition.
>
> **Also, for, e.g., the equation preceding (29), it is written $P(\Delta(W,\tilde{S}_l,U) \ge t)$. This is misleading, as it appears that the probability holds over a joint draw of $(W,\tilde{S}_l,U)$, but in fact, only $U$ is random while $W$ and $\tilde{S}_l$ is fixed. Is this correct?**
>
> Yes. We have updated the proof of Theorem 3 and 4 for better clarity.
>
> **At the end of p.4: the supersample generalization gap approaches the true one, but the same is not true for the leave-one-out, as the test loss is still evaluated on only one sample. Does this limit the usefulness of the LOO results?**
>
> This is a natural limitation of the LOO setting that cannot be overcome from the perspective of generalization bounds. In the last paragraph of page 5, we discussed the possibility of using our techniques to explore alternative supersample settings of randomly selecting $n < m$ samples from $m$ supersamples. By letting $n \rightarrow \infty$ as $m \rightarrow \infty$, this limitation could be overcome in these new settings.

---

> ### Author Response · Authors · 2023-11-15
> **Response to Reviewer YPBm (3/4)**
>
> **In the numerical evaluations, the cross-entropy loss is used, which is continuous. To compute bounds for MNIST etc, a binning with a bin size of 0.5 is used. The usefulness of the results, despite the assumption of discreteness, was motivated by the fact that floating-point numbers are used in practice. Certainly, machine precision is significantly smaller than 0.5. This specific choice is also not motivated. What is the effect when decreasing the binning size? Is there any reason to believe that this is a sensible choice?**
>
> It is impractical to choose machine precision as the bin size, as we do not have enough samples to estimate the probability of each bin. Separating the data into $5 - 10$ bins is a common choice for entropy estimation (Yu et al., 2019) (Kawaguchi et al., 2023). In our empirical studies, the number of bins is roughly $15$, already larger than the common value $10$. We further report the visualization results with a bin size of $0.3$ in **Figure 5**. It can be seen that the bin size only has a mild impact on the final results. Considering that we only have $40$ samples for each loss for CIFAR10 experiments, there will not be sufficient samples for probability estimation once we further decrease the bin size.
>
> Yu S, Giraldo LG, Jenssen R, Principe JC. Multivariate Extension of Matrix-Based Renyi's $\alpha$-Order Entropy Functional.
>
> **In the numerical evaluation, optimization is performed over several constants in the bounds. In the derivation, they are assumed fixed, and the bounds hold with a certain probability for this fixed value. In order to have a valid bound while optimizing, one needs to perform some kind of union bound, or other argument to guarantee the validity. Otherwise, it should be stated that the bounds are not actually valid due to this optimization, but are just illustrative. Did I miss something that motivated this optimization?**
>
> In this paper, our bounds are presented in the form of $P(\Delta \le B) \ge 1 - \delta$, where $\Delta$ is the validation error and $B$ is the upper bound involving the constants $\kappa$, $\gamma$, etc. Let $b = \mathbb{E}[B]$ be the expectation of $B$ taken over $W$, $\tilde{S}_s$, and $\tilde{U}$. We then optimize the constants by minimizing $b$, which is a constant independent of the randomness in our bounds. Such an optimization strategy is sufficient to guarantee the validity of our results.
>
> **The bounds are said to be compared to the Binary KL bound of Hellstrom and Durisi, but Theorem 10 in the paper is not a binary KL bound. Furthermore, in HD Thm. 7, the information measure that appears is not a CMI, but a random KL divergence depending on the supersample and selection variable. Also, the binary KL bound does not imply a bound on the generalization gap, but instead, directly bounds the population loss given a training loss. Finally, the binary KL bound is inherently only valid for bounded losses, as the binary KL is undefined for inputs outside of $[0,1]$. Is the comparison actually to Thm. 10, i.e., the square-root bound of Hellstrom and Durisi?**
>
> Sorry for the confusion, and we have corrected Theorem 10 to represent the binary KL bound. Since the stochastic KL divergence is hard to estimate, we conduct the comparison by averaging both bounds over multiple draws of $\tilde{S}_s$ and $\tilde{U}$. In this way, the averaged stochastic KL divergence approaches the CMI quantity, which is then estimated through a lower bound approximation. To accommodate unbounded losses, we empirically estimate the largest loss $B^{W,\tilde{S}_s}$ and adopt the scaled loss $\ell(W,\cdot)/B^{W,\tilde{S}_s}$ as suggested by (Hellstrom and Durisi et al.).
>
> **The plots show the generalization gap, rather than the bound on the population loss itself. Arguably, the latter is more interesting, since a low generalization gap with a high training loss is not very useful. Is it possible to show the bound on the population loss?**
>
> As the population loss is defined in terms of expectation, our results are not directly applicable to bound the population loss. Fortunately, in the supersample setting, the population loss can be approximated by the average test loss when $n$ is large enough. We additionally visualize the comparison between different upper bounds for the test risk in **Figure 6**.

---

> ### Author Response · Authors · 2023-11-15
> **Response to Reviewer YPBm (4/4)**
>
> **I am not sure the way the term “PAC” is used is really consistent with PAC learning. PAC bounds are typically uniform in some sense, whereas the bounds in this paper are tail bounds under the joint distribution of data and hypothesis.**
>
> PAC bounds typically indicate generalization bounds that hold for a certain class of hypotheses, where the probability is taken over the draw of the training dataset. Generally speaking, Theorem 1 and 2 can be regarded as a special case of PAC bounds, where the hypothesis class is taken as a single $w$. Upon further literature review, we agree that the terminologies of "high-probability bounds" or "single-draw bounds" are more suitable to characterize our theoretical results. As defined in (Hellstrom et al., 2023), single-draw bounds are generalization bounds that hold with high probability over the draw of both the dataset and the hypothesis. We have updated our main text accordingly to enhance clarity.
>
> Hellstrom F, Durisi G, Guedj B, Raginsky M. Generalization Bounds: Perspectives from Information Theory and PAC-Bayes.
>
> **p.5, “The most notable improvement of…” here, a comparison is made to CMI bounds instead of e-CMI bounds, which are arguably more relevant; “This illustrates a novel trade-off” is it really a trade-off? Both are just preferred to be small.**
>
> As indicated by the second equation on page 5, the loss entropy term $H(R^W)$ is also likely to be tighter than the e-CMI quantity $I(R^W;U|\tilde{S}_l)$. During training, only the training loss entropy can be explicitly minimized. To balance the minimization of training and test loss entropies, one needs to adopt certain regularization (e.g. weight decay, dropout), which may negatively affect training loss entropy, implying a trade-off between the two.

---

> ### Comment · Reviewer_YPBm · 2023-11-15
> **Thank you; remaining concerns**
>
> Thank you for your response. It addresses many of the points I raised. Below are some points where I still have concerns.
>
> ***
>
> Regarding the assumption of discreteness and the binning:
>
> I understand that the discrete assumption enables you to derive bounds with attractive properties, and that it has been used in prior works. Neither of these motivate applying it to continuous settings like regression, which needs further justification.
>
> For the bins, I understand that it is impractical to choose machine precision as the bin size. Thus, it is misleading to say that the discrete approach “aligns well with digital computers employing floating-point numbers”—as you point out, this is still too impractical to be well-aligned.
>
> Regarding Figure 5: Thank you for producing this additional figure. I do not agree that this demonstrates a mild impact. For MNIST, SGLD, for instance, the loss-entropy bounds increase by around a factor of 2. I suspect that this increase would continue until you reached machine precision — intuitively, one would expect some losses very close to 0, while the rest would almost invariably be in their own bin, right?
>
> I think there are some valid conclusions that you can draw based on the binned bounds, though, but they only apply to the actual binned loss (which can be used to get upper bounds on the underlying loss). So, if you use a bin size of $0.5$ and get a bound of $0.6$, you can conclude that the actual loss is at most $1.0$.
>
> ***
>
> Regarding the appearance of the test loss on the RHS of bounds: I understand that the maximum observed test loss is finite in practice. This does not change the fact that _you are bounding the test loss in terms of the test loss_. If the test loss is assumed to be known, you already have a very tight bound (an equality, in fact). Arguably, similar critiques can be raised against any e-CMI-type bound, but a dependence on information measures including the test loss can be more reasonably controlled than just the actual value of the test loss itself. This raises questions about the purpose of the bound.
>
> ***
>
> Regarding the sub-Gaussianity assumptions: it is still not clear under what distribution the losses are sub-Gaussian. Here, $L^W_{i,0}=\ell(W, Z_{i,0})$. Do you mean that it is sub-Gaussian under the joint distribution of $W, Z_{i,0}$? Do you mean that, for a fixed $w$, it is sub-Gaussian under the expectation of $Z_{i,0}$? If it is the first one, then this is assuming sub-Gaussianity under the _joint_ distribution of the hypothesis and training data, which is significantly different from the standard assumption, and if it is the second one, then this is not sufficient for the rest of the argument.
>
> ***
>
> Regarding $P(\Delta(W,\tilde{S}_l,U) \ge t)$: the notation you use still has the same issue as I commented about. If you either use lower-case letters to indicate fixed instances, or use $P_U$ to notate the probability under the randomness of $U$, or potentially write that it is the probability given $(W,\tilde{S)$,  it would be clearer.
>
> ***
>
> Regarding the discussion of related work in Appendix G.1: I appreciate that you tried to cover all related work, but the suggestion mainly regarded giving some qualifications for the statement “these bounds are primarily restricted to average-case scenarios” when discussing information-theoretic generalization bounds.

---

> > ### Author Response · Authors · 2023-11-17
> > **Response to Remaining Concerns (1/2)**
> >
> > Dear Reviewer YPBm, thanks for the prompt response! We will address your remaining concerns as follows:
> >
> > **I understand that the discrete assumption enables you to derive bounds with attractive properties, and that it has been used in prior works. Neither of these motivate applying it to continuous settings like regression, which needs further justification.**
> >
> > **I think there are some valid conclusions that you can draw based on the binned bounds, though, but they only apply to the actual binned loss (which can be used to get upper bounds on the underlying loss). So, if you use a bin size of $0.5$ and get a bound of $0.6$, you can conclude that the actual loss is at most $1.0$.**
> >
> > In Section G.3, we further discuss an applicable discretization strategy for continuous losses. It is shown that the discretization error of the test risk can be upper-bounded by $O(b/\sqrt{n})$, where $b$ is the bin size. This observation is also verified by our empirical results: in the MNIST (Adam) experiment, let $\Delta$ and $\hat{\Delta}$ be the validation error before and after discretization with bin size $0.3$.
> >
> > | $n$  | $\mathbb{E}\|\Delta\|$ | $\mathbb{E}\|\Delta-\hat{\Delta}\|$ | $\frac{\mathbb{E}\|\Delta-\hat{\Delta}\|}{\mathbb{E}\|\Delta\|}$ |
> > |------|:----------------------:|:-----------------------------------:|:----------------------------------------------------------------:|
> > | 75   | 0.3483                 | 0.0049                              | 1.41\%                                                           |
> > | 250  | 0.1488                 | 0.0018                              | 1.22\%                                                           |
> > | 1000 | 0.0863                 | 0.0006                              | 0.68\%                                                           |
> > | 4000 | 0.0553                 | 0.0002                              | 0.42\%                                                           |
> >
> > As can be seen, the discretization error $\mathbb{E}\|\Delta-\hat{\Delta}\|$ is much smaller than the actual bin size and scales with $1/\sqrt{n}$. This discretization error bound can then be combined with our main theorems to yield valid generalization bounds for continuous loss functions.
> >
> > **For the bins, I understand that it is impractical to choose machine precision as the bin size. Thus, it is misleading to say that the discrete approach “aligns well with digital computers employing floating-point numbers”—as you point out, this is still too impractical to be well-aligned.**
> >
> > **Regarding Figure 5: Thank you for producing this additional figure. I do not agree that this demonstrates a mild impact. For MNIST, SGLD, for instance, the loss-entropy bounds increase by around a factor of 2. I suspect that this increase would continue until you reached machine precision — intuitively, one would expect some losses very close to 0, while the rest would almost invariably be in their own bin, right?**
> >
> > We have modified our statements regarding floating-point numbers. We agree that the impact of bin size on the loss entropy is not negligible, but a machine-precision bin size is not necessary to acquire valid generalization bounds for continuous losses, as we discussed above. This impact can also be further minimized by increasing the number of samples.
> >
> > **Regarding the appearance of the test loss on the RHS of bounds: I understand that the maximum observed test loss is finite in practice. This does not change the fact that you are bounding the test loss in terms of the test loss. If the test loss is assumed to be known, you already have a very tight bound (an equality, in fact). Arguably, similar critiques can be raised against any e-CMI-type bound, but a dependence on information measures including the test loss can be more reasonably controlled than just the actual value of the test loss itself. This raises questions about the purpose of the bound.**
> >
> > To evaluate our bounds, the exact values of test losses are not required. Instead, we only require a quantile estimation for the coefficients $\Sigma_{R^W}$ in Theorem 3, $\tilde{C}_1^W$ in Theorem 4, or $\tilde{G}_3^W$ in Theorem 6 (Note that we can directly take $\hat{L}_i^W, \hat{L}_i^{W,\kappa} = \kappa$ in Theorem 5 and 6 when test losses are not available). These quantities only reveal the rough magnitude of test losses, and should be much easier to estimate than those information measures - not only because they are $1$-dimensional, but also the fact that we just need a quantile estimation, rather than the entire probability distribution required by estimating information measures. In general, these quantities would not be the bottleneck to evaluate our bounds when test losses are not available.

---

> > ### Author Response · Authors · 2023-11-17
> > **Response to Remaining Concerns (2/2)**
> >
> > **Regarding the sub-Gaussianity assumptions: it is still not clear under what distribution the losses are sub-Gaussian. Here, $L_{i,0}^W = \ell(W, Z_{i,0})$. Do you mean that it is sub-Gaussian under the joint distribution of $W, Z_{i,0}$? Do you mean that, for a fixed $w$, it is sub-Gaussian under the expectation of $Z_{i,0}$? If it is the first one, then this is assuming sub-Gaussianity under the joint distribution of the hypothesis and training data, which is significantly different from the standard assumption, and if it is the second one, then this is not sufficient for the rest of the argument.**
> >
> > We mean the first case, that $L_{i,0}^W$ is $\sigma$-subgaussian under the joint distribution of $W, Z_{i,0}$. We agree that this is different from the standard definition. However, $\sigma$ can still be directly estimated from given samples, and is also strictly tighter than the upper bound of the loss function. These conditions are sufficient for us to get tighter upper-bound estimates for the expected generalization error.
> >
> > **Regarding $P(\Delta(W,\tilde{S}_l,U) \ge t)$: the notation you use still has the same issue as I commented about. If you either use lower-case letters to indicate fixed instances, or use $P_U$ to notate the probability under the randomness of $U$, it would be clearer.**
> >
> > Thanks for the suggestion. We are now using $P_U$ to indicate the distribution that is taken over.
> >
> > **Regarding the discussion of related work in Appendix G.1: I appreciate that you tried to cover all related work, but the suggestion mainly regarded giving some qualifications for the statement “these bounds are primarily restricted to average-case scenarios” when discussing information-theoretic generalization bounds.**
> >
> > Thank you for the suggestion. We have qualified these statements to indicate the fact that most computationally tractable generalization bounds are restricted to average-case scenarios.

---

> ### Comment · Reviewer_YPBm · 2023-11-17
> **Sub-Gaussianity; appearance of test loss; discretization**
>
> Thank you for your reply.
>
> ***
>
> Sub-Gaussianity: Okay, I think the steps are essentially fine, although I think the argument sort of implicitly assumes zero-mean? However, when you take the difference of the two terms I think this does not matter. The assumption of sub-Gaussianity under the joint distribution of the data and hypothesis is a bit strange, and I am unsure if this really takes you much further than essentially bounded losses (e.g., sub-Gaussian under $P_Z$ for all $w$ does not imply your condition).
>
> ***
>
> Regarding the appearance of the test-loss: okay, for some cases the exact value is not needed. But I don't see how you get away \$\tilde C_1^W$ in Theorem 4 -- how would you estimate/bound the sum of squared test-train-loss gaps without knowing the test loss?
>
> ***
>
> Regarding the discretization: the technique you present in App. G.3 appears to be neat and to work. However, your figures appear to be the same. Unless I misunderstand, G.3 does not mean that your results as-presented are valid -- you would need to estimate both training and test loss using the perturb-discrete approach, and also use these perturb-discretized losses to estimate the loss entropy. Could you clarify this point?

---

> > ### Author Response · Authors · 2023-11-18
> > **Response to Remaining Questions**
> >
> > Dear Reviewer YPBm, thank you again for the prompt response! We address the remaining questions as follows:
> >
> > **Sub-Gaussianity: Okay, I think the steps are essentially fine, although I think the argument sort of implicitly assumes zero-mean? However, when you take the difference of the two terms I think this does not matter. The assumption of sub-Gaussianity under the joint distribution of the data and hypothesis is a bit strange, and I am unsure if this really takes you much further than essentially bounded losses (e.g., sub-Gaussian under $P_Z$ for all $w$ does not imply your condition).**
> >
> > Actually, we do not require the loss difference to be zero-mean, as $\mathbb{E}[(-1)^{U_i}] = 0$ is sufficient to guarantee that $\mathbb{E}[(-1)^{U_i}\Delta\bar{L}_i^W] = 0$. It is noteworthy that even for bounded losses, the subgaussian coefficient $\sigma$ can still be tighter than the loss upper bound. For example, assume that $P(L = 0) = 0.9$, $P(L = 1) = 0.1$ when using the 0-1 loss. We then have $\sigma \approx 0.43 < 0.5$.
> >
> > **Regarding the appearance of the test-loss: okay, for some cases the exact value is not needed. But I don't see how you get away $\tilde C_1^W$ in Theorem 4 -- how would you estimate/bound the sum of squared test-train-loss gaps without knowing the test loss?**
> >
> > For any $w$, let $Z \sim \mu$ and $L = \ell(w, Z)$. We agree that estimating $\tilde C_1^W$ is intractable without any prior knowledge about $L$. However, it suffices to adopt a few validation samples to estimate $\mathbb{E}[L]$ and $\mathbb{E}[L^2]$. To see this, let $L_{i,0}$ and $L_{i,1}$ with $i \in [1,n]$ be the training and test losses respectively, then by applying Markov's inequality:
> >
> > $$ P\left(\frac{1}{n}\sum_{i=1}^n (\Delta L_i)^2 \ge \epsilon\right) \le \frac{\frac{1}{n}\sum_{i=1}^n \mathbb{E}[(\Delta L_i)^2]}{\epsilon} = \frac{\frac{1}{n}\sum_{i=1}^n \mathbb{E}[L_{i,0}^2 + L_{i,1}^2 - 2L_{i,0}L_{i,1}]}{\epsilon} = \frac{\frac{1}{n}\sum_{i=1}^n L_{i,0}^2 + \mathbb{E}[L^2] - 2\mathbb{E}[L] (\frac{1}{n}\sum_{i=1}^n L_{i,0})}{\epsilon}.$$
> >
> > Therefore, one can easily acquire a quantile estimation of $\tilde C_1^W$ through a few validation samples, which is also necessary when training models in practice.
> >
> > **Regarding the discretization: the technique you present in App. G.3 appears to be neat and to work. However, your figures appear to be the same. Unless I misunderstand, G.3 does not mean that your results as-presented are valid -- you would need to estimate both training and test loss using the perturb-discrete approach, and also use these perturb-discretized losses to estimate the loss entropy. Could you clarify this point?**
> >
> > Indeed, the evaluation of our bounds utilizing the new discretization strategy is different from the original one. As suggested, we have updated Figures 2 - 6 to reflect this change. The loss values are discretized to estimate the main bounds, which are then combined with Proposition 3 to upper bound the continuous validation error.

---

> > > ### Comment · Reviewer_YPBm · 2023-11-18
> > > **Final response**
> > >
> > > *Sub-Gaussianity*: It appears that the sub-Gaussianity under $P_U$, $P_{WZ}$, or $P_UP_{WZ}$ is mixed somewhat confusingly. I understand that $\sigma$ can be tighter, but obtaining tighter bounds requires knowledge about the joint distribution of the hypothesis and data.
> > >
> > > *The test-loss*: so in order to compute your high-probability bound on the test loss, you need to use a validation set to estimate the average test loss and its variance. I think this would be an important point to emphasize.
> > >
> > > *Discretization*: Thank you for updating the figures. It appears as if the study of the effect of bin size has disappeared, though.

---

> > > > ### Author Response · Authors · 2023-11-18
> > > > **Response to Remaining Concerns**
> > > >
> > > > Dear Reviewer YPBm, we address the remaining concerns as follows:
> > > >
> > > > **Sub-Gaussianity: It appears that the sub-Gaussianity under $P_U$, $P_{WZ}$, or $P_UP_{WZ}$ is mixed somewhat confusingly. I understand that $\sigma$ can be tighter, but obtaining tighter bounds requires knowledge about the joint distribution of the hypothesis and data.**
> > > >
> > > > It should be noted that when applying Lemma 2 in the proof of Lemma 14, the expectation is taken over $P_{\tilde{U}\_i} P_{\Delta L_i^W}$ instead of $P_{\tilde{U}}P_{W,\tilde{S}\_s}$. In Lemma 14, $L_{i,0}^W$ and $L_{i,1}^W$ are assumed to be $\sigma$-subgaussian under the joint distribution of $W$, $\tilde{S}\_s$, and $\tilde{U}$. This also means that $L_{i,0}^W$ and $L_{i,1}^W$ are $\sigma$-subgaussian under their marginal distributions $P_{L_{i,0}^W}$ and $P_{L_{i,1}^W}$. We then know that $\Delta L_i^W$ is $2\sigma$-subgaussian under $P_{\Delta L_i^W}$. Since $\Delta L_i^W$ is symmetric, $(-1)^{\tilde{U}\_i} \Delta \bar{L}\_i^W$ shares the same distribution as $\Delta L_i^W$, and is thus $2\sigma$-subgaussian under $P_{\tilde{U}\_i} P_{\Delta L_i^W}$.
> > > >
> > > > Meanwhile, the current experimental setting as adopted in (Wang \& Mao, 2023) is sufficient to estimate $\sigma$, as multiple draws of $W$, $\tilde{S}\_s$, and $\tilde{U}$ are taken under their joint distribution to acquire samples for each $L_{i,0}^W$ and $L_{i,1}^W$.
> > > >
> > > > **The test-loss: so in order to compute your high-probability bound on the test loss, you need to use a validation set to estimate the average test loss and its variance. I think this would be an important point to emphasize.**
> > > >
> > > > Thanks for the suggestion. We have added relevant discussions in Appendix F.2.
> > > >
> > > > **Discretization: Thank you for updating the figures. It appears as if the study of the effect of bin size has disappeared, though.**
> > > >
> > > > We add back the study of the effect of bin sizes, as seen in Figure 5. We also discuss the trade-off between loss entropies and the discretization error raised by the choice of bin sizes in Appendix F.2.

---

### Official Review · Reviewer_gBmk · 2023-10-31

**Soundness:** 2 fair
**Presentation:** 3 good
**Contribution:** 2 fair
**Rating:** 6
**Confidence:** 4

**Summary:**

In this paper, the authors introduce novel upper bounds for generalization error or validation error in the context of high-dimensional applications, particularly deep models.

**Strengths:**

The paper provides some new PAC upper bounds which can be used in deep models.

**Weaknesses:**

Weaknesses and Questions:

-  In Theorems 1, 2, 3, and 4, it is essential to specify the distribution under which the inequality holds.

- The paper states that "the model $w$ is deterministic, which is commonly the case for modern deep learning models," but notes that $w$ is not truly deterministic due to initialization.

- In the discussion preceding Section 3.2, the authors mention that their data-independent bounds provide insights into understanding the generalization of the MEE criterion. However, for binary classification, margin-based loss functions, as discussed in [1], cannot be represented as a function of $E^w$.

- The effect of the learning algorithm on Theorems 1 and 2 is unclear. It seems that the model, $w$, remains fixed and independent of the training set, limiting the practicality of these upper bounds.

- The proof of Lemma 13 lacks clarity. More elaboration is needed, especially in the last line of the proof.

- The assumption in Theorem 5 that $\kappa \geq B^{W,\tilde{S}}$ guarantees a valid $C_i$ due to the term $\log(2-e^{2\eta \hat{L}_i^W})$. However, Theorem 6 discusses the case where $\kappa < B^{W,\tilde{S}}$. Explain how $C_i$ remains valid in Theorem 6.

- If the loss function is bounded, then $B^{W,\tilde{S}}$ is also bounded. Can the authors provide an example where the loss function is unbounded, but $B^{W,\tilde{S}}$ remains bounded?

- Please verify the equation between eq. (41) and (42) in the appendix.

- The discussion after Theorem 5 is unclear. Define what an "interpolating regime" is, and clarify whether the empirical risk is considered a random variable in Theorem 5.

- The novelty of this work is limited.

Suggestions:

- Include a table summarizing notations in the Appendix.

References:

[1]: Bartlett, Peter L., Michael I. Jordan, and Jon D. McAuliffe. "Convexity, classification, and risk bounds." Journal of the American Statistical Association 101.473 (2006): 138-156.

----


After Rebuttal, I increased my score to 6.

**Questions:**

See the weaknesses section.

---

> ### Author Response · Authors · 2023-11-15
> **Response to Reviewer gBmk (1/2)**
>
> Dear Reviewer gBmk, thanks for your detailed and constructive comments! We will address your concerns in the following response:
>
> **In Theorems 1, 2, 3, and 4, it is essential to specify the distribution under which the inequality holds.**
>
> As we stated in Section 2, for Theorem 1 and 2, the probability is taken over $S$; for the following Theorems 3 - 6, the probability is taken over $W$, $\tilde{S}_l$ (or $\tilde{S}_s$), and $U$ (or $\tilde{U}$). We have updated the expressions of these Theorems to reflect this.
>
> **The paper states that "the model $w$ is deterministic, which is commonly the case for modern deep learning models," but notes that $w$ is not truly deterministic due to initialization.**
>
> Sorry for the confusion, but we do not presume that the model itself is deterministic. Instead, we mean that the forward propagation of the neural network does not involve external randomness, such that the output $L^w$ can be determined given the sample $Z$, and we have $H(L^w|Z) = 0$ for any given model $w$. This is commonly seen in most CNN architectures.
>
> **In the discussion preceding Section 3.2, the authors mention that their data-independent bounds provide insights into understanding the generalization of the MEE criterion. However, for binary classification, margin-based loss functions, as discussed in [1], cannot be represented as a function of $E^w$.**
>
> Following the notations in [1], let $L^w = \phi(Y f(X))$ be a margin-based loss function and $E^w = Y - f(X)$. Here, $L^w$ can still be represented as a deterministic function of $E^w$ given $Y$: $L^w = \phi(Y (Y - E^w))$, so we have $H(L^w|Y) \le H(E^w|Y)$ by the data-processing inequality. The same conclusion then follows by the fact that $H(E^w|Y) \le H(E^w)$.
>
> **The effect of the learning algorithm on Theorems 1 and 2 is unclear. It seems that the model, $w$, remains fixed and independent of the training set, limiting the practicality of these upper bounds.**
>
> Theorem 1 and 2 are intended to be **data-independent**, where the model $w$ is assumed to be fixed and independent of $S$. As we discussed in the last paragraph of Section 3.1, these bounds are still applicable to pre-training or validation tasks. Note that the subsequent Section 3.2 provides **data-dependent** bounds, where the model $W$ is no longer fixed and is learned from $\tilde{S}_s$, enabling a wider range of applications in practice.
>
> **The proof of Lemma 13 lacks clarity. More elaboration is needed, especially in the last line of the proof.**
>
> We include more details in the proof of Lemma 13 for better clarity.
>
> **The assumption in Theorem 5 that $\kappa > B^{W,\tilde{S}}$ guarantees a valid $C_i$ due to the term $\log(2-e^{2\eta \hat{L}_i^W})$. However, Theorem 6 discusses the case where $\kappa < B^{W,\tilde{S}}$. Explain how $C_i$ remains valid in Theorem 6.**
>
> Note that in Theorem 6, $\hat{L}_i^W$ is replaced by $\hat{L}_i^{W,\kappa}$ in the expression of $C_i$, where $\hat{L}_i^{W,\kappa} \le \kappa$ is guaranteed. Therefore, $C_i$ remains valid.
>
> **If the loss function is bounded, then $B^{W,\tilde{S}}$ is also bounded. Can the authors provide an example where the loss function is unbounded, but $B^{W,\tilde{S}}$ remains bounded?**
>
> To evaluate our bounds, $B^{W,\tilde{S}}$ is only required to be finite, and boundedness is not necessary. For example, although the cross-entropy loss is usually unbounded, we always get finite loss values during the training process, resulting in finite $B^{W,\tilde{S}}$. In fact, for any long-tailed loss distribution, we have $P(L_i^W<\infty) = 1$, so $B^{W,\tilde{S}}$ is guaranteed to be finite with probability $1$.
>
> **Please verify the equation between eq. (41) and (42) in the appendix.**
>
> Thanks for pointing out the typos in our proof. These typos have been fixed and do not affect our subsequent analysis.
>
> **The discussion after Theorem 5 is unclear. Define what an "interpolating regime" is, and clarify whether the empirical risk is considered a random variable in Theorem 5.**
>
> Interpolating regime is the cases where the model achieves $0$ training risk (Hellstrom \& Durisi, 2022) (Wang \& Mao, 2023), i.e. $L_{i,\tilde{U}_i}^W = 0$ for all $i \in [1,n]$. The empirical risk is considered a random variable in this paper, where the randomness comes from $W$, $\tilde{S}_l$ (or $\tilde{S}_s$), and $U$ (or $\tilde{U}$).

---

> > ### Comment · Reviewer_gBmk · 2023-11-20
> > **Thanks for rebuttal**
> >
> > Thank the authors for their response. It addresses many of the points I raised. Below are some points where I still have concerns.
> >
> > ---
> >
> > > Interpolating regime is the cases where the model achieves training risk (Hellstrom & Durisi, 2022) (Wang & Mao, 2023), i.e. $L_{i,\hat{U}_i}^{W}=0$ for all $i\in[1,n]$. The empirical risk is considered a random variable in this paper, where the randomness comes ....
> >
> > From a theoretical perspective, *you can not assume that a random variable is zero.* You should prove that if $n \to \infty$ then the empirical risk would converge zero. You should also show what is the rate of this convergence. Then, you can combine it with your first term in Theorem 5 and finally conclude that your convergence is $O(1/n)$. Otherwise, I believe that it is a strong assumption.
> >
> > >To evaluate our bounds, $B^{W,\tilde{S}}$ is only required to be finite, and boundedness is not necessary. For example, although the cross-entropy loss is usually unbounded, we always get finite loss values during the training process, ....
> >
> > Could we have a loss function which is not sub-gaussian and then we get finite $B^{W,\tilde{S}}$?
> >
> > ----
> > **follow-up questions:**
> > I would be so grateful if the authors could answer the following questions:
> >
> > - What is your activation function in your experiments?
> >
> > - In the proof of Theorem 6, the authors apply union bound, however, the final statement is $1-\delta$, I checked the proof, I think that it should be $1-2\delta$ for $0<\delta\leq 0.5$.
> >
> > - In Theorem 6, we have a third term which is not of order $O(1/n)$ necessarily. Why the upper bound in Theorem 6 is categorized as fast rate upper bound?
> >
> > - what is the connection between the definition of sub-exponential, sub-weibull or sub-gaussian and $B^{W,\tilde{S}}$?

---

> > > ### Author Response · Authors · 2023-11-20
> > > **Response to Remaining Concerns**
> > >
> > > Dear Reviewer gBmk, thanks for the feedback! We will address your remaining concerns as follows:
> > >
> > > **From a theoretical perspective, you can not assume that a random variable is zero. You should prove that if $n \rightarrow \infty$ then the empirical risk would converge zero. You should also show what is the rate of this convergence. Then, you can combine it with your first term in Theorem 5 and finally conclude that your convergence is $O(1/n)$. Otherwise, I believe that it is a strong assumption.**
> > >
> > > Actually, this assumption is widely adopted in the literature. Let $\hat{L}$ be the empirical risk, it is assumed that $\mathbb{E}[\hat{L}] = 0$ as seen in (Hellstrom \& Durisi, 2022) (Wang \& Mao, 2023). Since $\hat{L} \ge 0$, this is equivalent to assuming that $P(\hat{L} = 0) = 1$. In other words, it is assumed that the network can always fit the given training dataset. Such a phenomenon can also be seen in practice, where the training accuracy approaches 100\% at the end of training.
> > >
> > > We agree that our claim regarding the $O(1/n)$ convergence rate is a bit strong, and have added certain qualifications to these statements. In general, the interpolating regime is an interesting special case that is worth some discussion. It is only used to intuitively show the advantage of Theorem 5 and 6 when the empirical risk is close to $0$, and is not adopted in our experiments when evaluating these bounds.
> > >
> > > **Could we have a loss function which is not sub-gaussian and then we get finite $B^{W,\tilde{S}}$?**
> > >
> > > It is important to note that the sub-gaussianity of losses is not only dependent on the loss function, but more importantly the distribution of $W$ and $\tilde{S}$. For example, let the error $E = Y - f(W, X)$ be sub-gaussian and $L = E^2$, where $f(W, \cdot)$ is the network parameterized by $W$, then the loss $L$ is no longer sub-gaussian but sub-exponential. As discussed in our previous response, $B^{W,\tilde{S}}$ remains finite as long as for each loss $L_i$, $P(L_i<\infty) = 1$. This condition is satisfied by any tailed distributions, including both light-tailed (e.g. sub-gaussian) and heavy-tailed (e.g. sub-exponential) ones. One can then take the union bound over each $L_i$ to show that $B^{W,\tilde{S}}$ is finite with probability $1$.
> > >
> > > **What is your activation function in your experiments?**
> > >
> > > We adopt ReLU as the activation function in our experiments.
> > >
> > > **In the proof of Theorem 6, the authors apply union bound, however, the final statement is $1 - \delta$, I checked the proof, I think that it should be $1 - 2\delta$ for $0<\delta\leq 0.5$.**
> > >
> > > Given any random variables $A, B$ and functions $f, g$, by taking the union bound, $P(A \le f(\frac{1}{\delta})) \ge 1 - \delta$ and $P(B \le g(\frac{1}{\delta})) \ge 1 - \delta$ together imply that $P(A + B \le f(\frac{1}{\delta}) + g(\frac{1}{\delta})) \ge 1 - 2\delta$, or equivalently, $P(A + B \le f(\frac{2}{\delta}) + g(\frac{2}{\delta})) \ge 1 - \delta$. As can be seen, the value of $\delta$ is already halved by taking the union bound: $G_2^W = \frac{1}{\lambda}\log(\frac{1}{\delta}) + \log(\frac{4}{\delta})$ in Theorem 5, while $\tilde{G}\_2^W = \frac{1}{\lambda}\log(\frac{2}{\delta}) + \log(\frac{8}{\delta})$ in Theorem 6.
> > >
> > > **In Theorem 6, we have a third term which is not of order $O(1/n)$ necessarily. Why the upper bound in Theorem 6 is categorized as fast rate upper bound?**
> > >
> > > We named Theorem 6 "Fast-Rate" because it is a direct follow-up of the fast-rate bound Theorem 5, and its main advantage compared to previous square-root bounds is brought by the fast convergence rate of the second term. We hope that such arrangements help readers to understand the difference between these bounds.
> > >
> > > **what is the connection between the definition of sub-exponential, sub-weibull or sub-gaussian and $B^{W,\tilde{S}}$?**
> > >
> > > The concepts of sub-exponential, sub-Weibull, and sub-gaussian are used to characterize the distributions of random variables, where sub-Weibull is a generalization of sub-exponential and sub-gaussian. If the losses are sub-Weibull, such a property can be used to provide concentration bounds for the losses (Li \& Liu, 2022), and also for $B^{W,\tilde{S}}$ by further taking the union bound. Generally, one can expect smaller $B^{W,\tilde{S}}$ when the losses are light-tailed (e.g. sub-gaussian), and the opposite when heavy-tailed (e.g. sub-exponential).
> > >
> > > Li S, Liu Y. High probability guarantees for nonconvex stochastic gradient descent with heavy tails.

---

> > > > ### Comment · Reviewer_gBmk · 2023-11-20
> > > > **Thanks for response.**
> > > >
> > > > Thanks for the response.
> > > >
> > > > The over-parameterization regime is an example of an interpolating regime. However, the authors did not mention the works related to generalization error analysis in the over-parameterization regime. For example, people studied the generalization error in some frameworks related to over-parameterization regimes, including, NTK [1] or Mean-field [2,3,4]. Note that in these works, the authors did not assume that the training loss (or empirical risk) is zero for all learning algorithms. As the main claim for this paper is related to providing results for deep learning models, I think the authors should also compare their results to these works in over-parameterization regime for better positioning their paper.
> > > >
> > > > Finally, I increased my score to 6.
> > > >
> > > > ---
> > > >
> > > > References:
> > > >
> > > > [1]:Zixiang Chen, Yuan Cao, Quanquan Gu, and Tong Zhang. A generalized neural tangent kernel
> > > > analysis for two-layer neural networks. Advances in Neural Information Processing Systems, 33:
> > > > 13363–13373, 2020.
> > > >
> > > > [2]: Naoki Nishikawa, Taiji Suzuki, Atsushi Nitanda, and Denny Wu. Two-layer neural network on
> > > > infinite dimensional data: global optimization guarantee in the mean-field regime. In Advances in
> > > > Neural Information Processing Systems, 2022
> > > >
> > > > [3]: Atsushi Nitanda, Denny Wu, and Taiji Suzuki. Particle dual averaging: Optimization of mean field
> > > > neural network with global convergence rate analysis. Advances in Neural Information Processing
> > > > Systems, 34:19608–19621, 2021.
> > > >
> > > > [4]: Aminian, Gholamali, Samuel N. Cohen, and Łukasz Szpruch. "Mean-field Analysis of Generalization Errors." arXiv preprint arXiv:2306.11623 (2023).

---

> > > > > ### Author Response · Authors · 2023-11-21
> > > > > **Thanks for the valuable comments**
> > > > >
> > > > > Dear Reviewer gBmk, thanks for the valuable suggestions! For completeness, we include further discussions about the over-parameterization and interpolating regimes, as seen in Appendix G.4.

---

> ### Author Response · Authors · 2023-11-15
> **Response to Reviewer gBmk (2/2)**
>
> **The novelty of this work is limited.**
>
> As discussed in the Introduction, we briefly summarize our contributions in this paper as follows:
>
> - We improve previous data-independent generalization bounds in (Kawaguchi et al., 2023) by introducing loss entropy. Our bounds are proven to be strictly tighter, and also provide the first information-theoretic understanding of the MEE criterion, showing that the generalization error scales with $O(\sqrt{H(E^w)/n})$.
>
> - We explore a novel technique beyond the traditional Donsker-Varadhan formula (Lemma 3) for decoupling random variables, which is the core of information-theoretic generalization analysis (see the paragraph preceding Section 3.2.1). Benefitting from this, we are able to reduce the dimensionality of the key information quantities in information-theoretic generalization bounds to $1$-dimensional, yielding computationally tractable high-probability generalization bounds.
>
> - By further exploring the thresholding strategy, we alleviate the stringent bounded loss assumption in previous high-probability and fast-rate generalization bounds, achieving significantly tighter bounds as verified by our empirical experiments.

---

### Official Review · Reviewer_xf1n · 2023-11-01

**Soundness:** 3 good
**Presentation:** 3 good
**Contribution:** 3 good
**Rating:** 6
**Confidence:** 3

**Summary:**

This paper proposes a series of PAC information-theoretic generalization bounds based on loss entropy. The novel bounds are tighter than the previous bounds. The authors conducted several experiments that show the superiority of their bounds.

**Strengths:**

- The paper is well-written and organized.
- Deep learning generalization analysis through information-theoretic tools has received a lot of attention lately. This paper makes a sound contribution to this line of work.
- The theory is sound. I skimmed through most of the proofs (I did not go through all of them in detail)  but the proofs are well-structured, easy to follow, and sound.
- I like the fact that the paper considers not only one but two settings: the Leave-One-Out Setting and the supersample setting. This makes the work more complete.

**Weaknesses:**

- The major concern with this work is that the analysis relies on $b^w$ and $B^{w, S}$, which represent the maximum of the loss. I think this makes the bounds sensitive to noise. In fact, it is sufficient that one sample has a relatively high loss to make the whole bound loose. The authors do not perform any experiments or analysis to show how robust their bounds are to noise.

**Questions:**

- Compared to the bounds in (Kawaguchi et al., 2023), the introduced bounds in this paper, e.g., Theorem 2, do not have any explicit dependency on the input variable $ X$. Interestingly, It only depends on the label $Y$. Can the authors comment on this point?

- Related to my first question, as the bounds depend directly on $Y$, it would interesting to see how sensitive they are to label noise. Do the authors have any comments on this? I think in this case, it might be interesting to conduct an additional experiment with label noise to evaluate how robust the bounds are.

---

> ### Author Response · Authors · 2023-11-15
> **Response to Reviewer xf1n**
>
> Dear Reviewer xf1n, thanks for your valuable suggestions! We address the raised questions as follows:
>
> **The major concern with this work is that the analysis relies on $b^w$ and $B^{w,S}$, which represent the maximum of the loss. I think this makes the bounds sensitive to noise. In fact, it is sufficient that one sample has a relatively high loss to make the whole bound loose. The authors do not perform any experiments or analysis to show how robust their bounds are to noise.**
>
> The quantities $b^w$ and $B^{w,S}$ are only considered in our data-independent bounds (**Theorem 1, 2**), which are mainly for qualitative analysis purposes. In the following data-dependent analysis, our main results (**Theorem 3, 4, 6**) overcome this limitation by tightening the $b^w$ and $B^{w,S}$ terms with the subgaussian norm or $L_2$ norm of loss differences, which are naturally more robust against noises. We refer the reviewers to **Figure 3** for a comparison between these robust bounds (Theorem 6) and the bound explicitly relying on $B^{W,\tilde{S}_s}$ (Theorem 5). In fact, this is one of the major contributions of our work: while the previous binary KL bound is easily loosened by the largest loss, our bounds are insensitive to extreme loss values and greatly improve over the binary KL bound (see **Figure 2**).
>
> **Compared to the bounds in (Kawaguchi et al., 2023), the introduced bounds in this paper, e.g., Theorem 2, do not have any explicit dependency on the input variable $X$. Interestingly, It only depends on the label $Y$. Can the authors comment on this point?**
>
> The data-independent bounds are implicitly dependent on $X$ through $L^w$: recall that $L^w = \ell(w, Z)$, where $Z$ denotes a pair of $X$ and $Y$. This is actually one of our main contributions: the loss entropy measure $H(L^w|Y)$ is strictly tighter than the original term $I(X;T^w|Y)$ in (Kawaguchi et al., 2023), as we remarked below Theorem 2.
>
> **Related to my first question, as the bounds depend directly on $Y$, it would interesting to see how sensitive they are to label noise. Do the authors have any comments on this? I think in this case, it might be interesting to conduct an additional experiment with label noise to evaluate how robust the bounds are.**
>
> Our data-independent bounds (Theorem 1, 2) are incapable of quantitative analysis since they involve the properties of the underlying nuisance variables ($c^w$ and $m$). Instead, we conduct a correlation analysis following (Kawaguchi et al., 2023) and show that $H(L^w)$ and $H(L^w|Y)$ can better reflect the behavior of the generalization error. As suggested by the reviewer, we conducted additional experiments with 20\% label noise in **Table 2**. It can be seen that our loss entropy metrics are consistently the better indicators of generalization.

---

> > ### Comment · Reviewer_xf1n · 2023-11-23
> > **Thanks for response**
> >
> > Thank you for the detailed answer and the clarifications. I do not have any further questions. I have read all the other reviews too. The main limitation here is the derived bound can not be computed in practice (only correlation analysis is possible), which 'limits' the impact/contribution. From a theoretical perspective, the contribution is strong. This is why I will keep my positive score.

---

### Comment · Area_Chair_YRvE · 2023-11-10
**Authors-Reviewers discussion starts today, ends on Nov 22**

Dear authors and reviewers,

@Authors: please make sure you make the most of this phase, as you have the opportunity to clarify any misunderstanding from reviewers on your work. Please write rebuttals to reviews where appropriate, and the earlier the better as the current phase ends on Nov 22, so you might want to leave a few days to reviewers to acknowledge your rebuttal. After this date, you will no longer be able to engage with reviewers. I will lead a discussion with reviewers to reach a consensus decision and make a recommendation for your submission.

@Reviewers: please make sure you read other reviews, and the authors' rebuttals when they write one. Please update your reviews where appropriate, and explain so to authors if you decide to change your score (positively or negatively). Please do your best to engage with authors during this critical phase of the reviewing process.

This phase ends on November 22nd.

Your AC

---

### Meta-Review · Area_Chair_YRvE · 2023-12-05

**Metareview:**

This meta-review is a reflection of the reviews, rebuttals, discussions with reviewers and/or authors, and calibration with my senior area chair. This paper contributes novel and tighter PAC information-theoretic generalisation bounds. There is a consensus among reviewers to praise the significance of the strong theoretical contributions, although some reviewers have pointed out a limited anticipated practical impact.

**Justification For Why Not Higher Score:**

Strong theoretical contributions, although with some limited practical impact.

**Justification For Why Not Lower Score:**

Strong theoretical contributions, although with some limited practical impact.

---

### Decision · Program_Chairs · 2024-01-16

Accept (poster)